# ADVERSARIAL ATTACK ROBUST DATASET PRUNING

## ABSTRACT

Dataset pruning, while effective for reducing training data size, often leads to models vulnerable to adversarial attacks. This paper introduces a novel approach to creating adversarially robust coresets. We first theoretically analyze how existing pruning methods result in non-smooth loss surfaces, increasing susceptibility to attacks. To address this, we propose two key innovations: 1) a Frequency-Selective Excitation Network (FSE-Net) that dynamically selects important frequency components, smoothing the loss surface while reducing storage requirements, and 2) a "Joint-entropy" score for selecting stable and informative samples. Our method significantly outperforms state-of-the-art pruning algorithms across various adversarial attacks and pruning ratios. On CIFAR-10, our approach achieves up to 58.19% accuracy under AutoAttack with an 80% pruning ratio, compared to 42.98% for previous methods. Moreover, our frequency pruning technique improves robustness even on full datasets, demonstrating its potential for enhancing model security while reducing computational costs.

## 1 INTRODUCTION

Dataset pruning aims to select a small subset of training data that can be used to efficiently train future models while maintaining high accuracy. A common approach to coreset selection involves assigning importance score to each example and selecting the most important ones (Ash et al., 2019)

Current state-of-the-art (SOTA) methods face challenges in that the model trained on the coreset often has low adversarial robustness. For instance, on CIFAR-10, a SOTA method CCS (Zheng et al., 2022) achieves 86.81% accuracy with a 90% pruning ratio, but this drops to just 37.86% when subjected to AutoAttack (Croce & Hein, 2020). This significant accuracy decline remains unexplained and poses a serious obstacle to further advancements in dataset pruning.

Traditional algorithms enhance robustness through adversarial training, which iteratively introduces perturbations to the training set. This significantly increases training costs, making it impractical for edge devices with limited resources (Bai et al., 2021). These devices typically use pruned datasets for training, rendering the overhead of adversarial training unsuitable.

We present theoretical and empirical insights into low adversarial robustness in existing models and introduce a novel coreset selection framework. Our analysis reveals how current coreset selection methods lead to non-smooth local minimum geometry (Definition 1), reducing adversarial robustness. We propose two algorithms to address this problem: 1) We designed a neural network to select important frequency components. This improves adversarial robustness by reducing logit entropy with extra benefits to reduce data storage which is

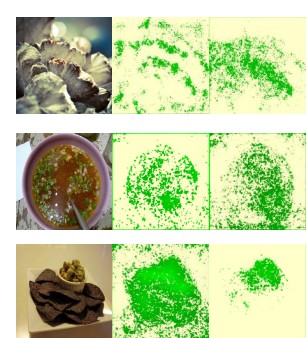

Figure 1: Sensitivity maps using SmoothGrad that highlight key components (green points) influencing model predictions. From left to right: the original image, the sensitivity map for the model trained with a 50% frequency pruning ratio, and the model trained on the original dataset. All the original figures come from Imagenet-1K.

valuable for memory-limited edge devices and 2) for the training processing action analysis, we introduced a data importance score based on entropy variation during training, helping to select a stable coreset that maintains performance and further boosts adversarial robustness. In experiments,

we can apply method 1) to the entire dataset or combine methods 1) and 2) to generate a coreset with stronger adversarial robustness. All lemmas and theorems are rigorously proven in the Appendix..

The main contribution of our paper is: 1) To the best of our knowledge, this is the first work to address adversarial robustness in the context of dataset pruning. 2) We proposed a learnable frequency pruning algorithm that enhances adversarial robustness while reducing training data storage requirements. 3) We introduced a data importance score, based on analyzing variations in model logit entropy throughout the training process, to select a coreset that enhances the model's robustness against adversarial attacks. 4) We conducted extensive experiments across various datasets and adversarial attacks to demonstrate the efficiency of our algorithm.

## 2 RELATED WORKS

### 2.1 DATASET PRUNING

Dataset Pruning, also known as Coreset Selection, aims to shrink the dataset scale by selecting important samples according to some predefined criteria. Entropy (Coleman et al., 2019) explores the uncertainty and decision boundary with the predicted outputs. GraNd/EL2N (Paul et al., 2021) calculates the importance of a sample with its gradient magnitude. Forgetting (Toneva et al., 2018) defines forgetting events as an accuracy drop at consecutive epochs, and hard samples with the most forgetting events are important. AUM (Pleiss et al., 2020) identifies data by computing the Area Under the Margin, the difference between the true label logits and the largest other logits. CCS (Zheng et al., 2022) extends previous methods by pruning hard samples and using stratified sampling to achieve good coverage of data distributions at a large pruning ratio. While these algorithms propose various methods to enhance coreset performance, none consider the adversarial robustness of the model when trained on the coreset selected by these methods.

### 2.2 ADVERSARIAL ATTACK

Adversarial attacks manipulate machine learning models by introducing subtle perturbations to input data, causing incorrect predictions. FGSM (Goodfellow et al., 2014) generates adversarial examples using the gradient of a model's loss function. PGD (Madry, 2017) extends FGSM by iteratively applying small perturbations. AutoAttack (Croce & Hein, 2020) combines multiple methods for automatic evaluation without manual tuning. C&W attack (Madry, 2017) finds the smallest perturbation causing misclassification. Despite extensive research in deep learning, no existing algorithms specifically evaluate the impact of adversarial attacks on models trained with pruned datasets.

## 3 METHODOLOGY

### 3.1 THEORY ANALYSIS

Drawing from the findings in Stutz et al. (2021) and Liu et al. (2020), which establish a correlation between loss landscape flatness and adversarial robustness, we posit that enhancing a model's resilience to adversarial attacks necessitates the smoothing of its *local minimum geometry* shown in Definition 1.

**Definition 1** (Smooth Local Minimum Geometry). *Local minimum geometry stands for the geometric characteristics of the loss landscape in the immediate vicinity of the converged solution. For a model with parameters $\theta$ and loss function $\mathcal{L}$, a smoother local minimum geometry at the converged solution $\theta^*$ implies that for a given perturbation $\varepsilon$, where $\|\varepsilon\| \leq \delta$ for some small $\delta > 0$, the change in loss $\Delta\mathcal{L} = \mathcal{L}(\theta^* + \varepsilon) - \mathcal{L}(\theta^*)$ is statistically smaller compared to models with less smooth geometries.*

Dataset pruning aims to construct a coreset $S = \{(x_m, y_m)\}_{m=1}^{M}$, where $S \subset D$. The objective of dataset pruning is to identify a coreset such that a model trained on $S$ closely approximates the performance of a model trained on the full dataset $D$. This can be formulated as follows:

$$\mathbb{E}_{(x,y)\sim\mathcal{S},\theta_S\sim\mathcal{P}(\theta_S)}\left[\nabla_{\theta_S}\mathcal{L}(f_{\theta_S}(x),y)\right] \approx \mathbb{E}_{(x,y)\sim\mathcal{D},\theta_D\sim\mathcal{P}(\theta_D)}\left[\nabla_{\theta_D}\mathcal{L}(f_{\theta_D}(x),y)\right] \quad (1)$$

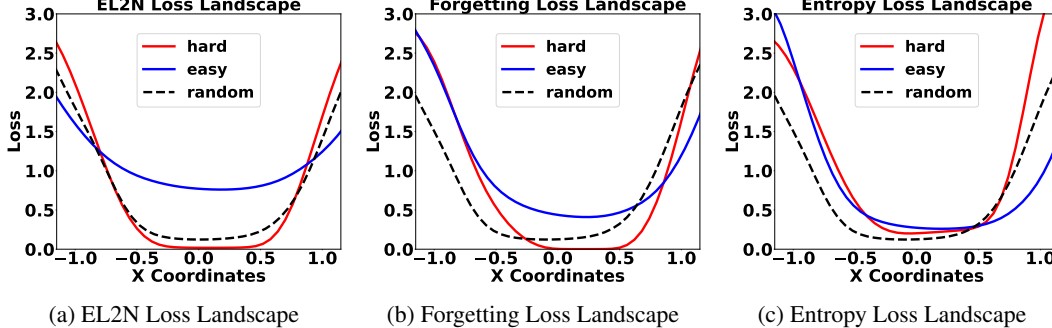

(a) EL2N Loss Landscape     (b) Forgetting Loss Landscape     (c) Entropy Loss Landscape

Figure 2: We compare training ResNet-18 on the top 50% hardest (red lines), top 50% easiest (blue lines), and a random 50% (black lines) of CIFAR-10 images. Local minimum geometry is visualized based on (a) EL2N, (b) Forgetting, and (c) Entropy scores. The hardest images result in the least smooth geometry. The figures follow the same setup as Li et al. (2018).

where $\nabla_\theta$ is the gradient operator with respect to the model parameters $\theta$, $\mathcal{L}(\cdot)$ is the loss function, $\theta_S$ and $\theta_D$ are the parameters of the models trained on $S$ and $D$ respectively, $f_{\theta_S}$ represents the model trained on subset $S$, and $f_{\theta_D}$ represents the model trained on the full dataset $D$, $y$ is true label to input $x$ (He et al., 2023). The traditional coreset selection method prioritizes selecting subsets that are difficult for the model to learn which are called "hard samples". Hard samples are characterized by producing larger gradients during training, leading to lower prediction confidence and requiring more substantial weight updates (Paul et al., 2021).

**Theorem 1** (Hard Samples and Local Minimum Geometry). *Formally, we compare the local minimum geometry at the converged solution $\theta^*$ for hard samples $x_h$ and randomly sampled data points $x_r$. For a given perturbation $\varepsilon$, where $\|\varepsilon\| \leq \delta$ for some small $\delta > 0$, hard samples are more likely to induce less smooth geometries, which can be characterized as:*

$$\mathbb{E}_{x_h}[\Delta\mathcal{L}_h] > \eta \cdot \mathbb{E}_{x_r}[\Delta\mathcal{L}_r] \tag{2}$$

*where $\Delta\mathcal{L}_h = |\mathcal{L}(\theta^* + \varepsilon, x_h) - \mathcal{L}(\theta^*, x_h)|$ and $\Delta\mathcal{L}_r = |\mathcal{L}(\theta^* + \varepsilon, x_r) - \mathcal{L}(\theta^*, x_r)|$. $\eta > 1$ is a threshold constant. The expectation $\mathbb{E}_{x_h}$ is taken over the distribution of hard samples, while $\mathbb{E}_{x_r}$ is taken over the randomly selected data distribution.*

Theorem 1 shows that achieving a smooth local minimum geometry with a coreset requires more than traditional pruning methods. Focusing only on the hardest samples leads to non-smooth local minimum geometry as demonstrated in Fig. 2a, Fig. 2b and Fig. 2c.

However, a smoother local minimum geometry does not always improve performance. Excessive smoothness can significantly degrade model capacity, resulting in poor generalization (Mei et al., 2022), so we need to find a balance point between smooth local minimum geometry and the capacity of the model. We can now formulate our goal to find a coreset as follows:

$$\min_{S \subset D} \quad \mathbb{E}_{(x,y) \sim S}\left[\max_{\|\delta\| \leq \epsilon} \Delta\mathcal{L}(f_{\theta_S} + \delta, x, y)\right]$$

$$\text{s.t.} \quad |C(f_{\theta_S}) - C(f_{\theta_D})| \leq \eta_c$$

where $\eta_c$ stands for a threshold value and $C(f_\theta)$ stands for model capacity, this formulation indicates the goal of selecting a coreset that minimizes the impact of perturbations on the local minimum geometry while maintaining a model capacity similar to that of the model trained on the full dataset.

Our algorithm can be summarized as follows: 1) We apply a learnable frequency pruning technique to preprocess the original dataset, targeting the inherent frequency characteristics of each sample for static, sample-level optimization (see Section 3.2). 2) We evaluate the importance of each sample by capturing the training dynamics and using this information to calculate a "Joint-Entropy" score for each sample (see Section 3.3).

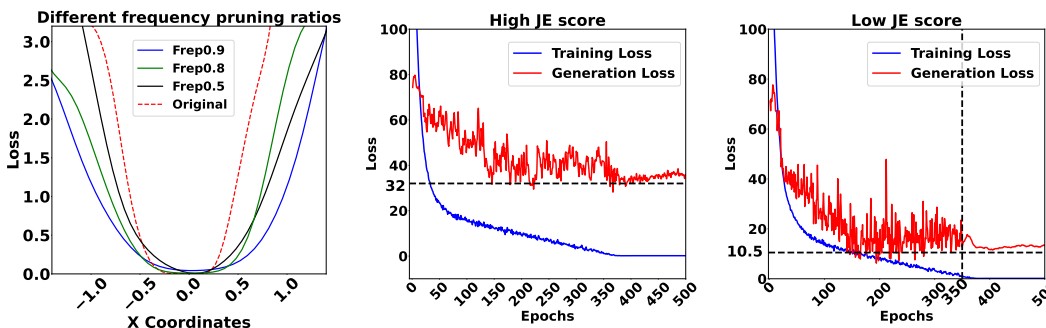

(a) Different Frequency pruning ratios

(b) Model trained by images with high JE score

(c) Model trained by images with low JE score

Figure 3: (a) shows the local minimum geometry of ResNet-18 trained on the CIFAR-10 dataset with different frequency pruning levels: 90% (blue), 80% (green), 50% (black), and no pruning (red). The results demonstrate that frequency pruning smooths the local minimum geometry. Figures (b) and (c) display the training loss (blue) and generation error (red) over 500 epochs for a ResNet-18 model trained on a CIFAR-10 coreset and the model was attacked by AutoAttack. In (b), the model is trained on the top 50% of images with the highest JE scores, while in (c) the bottom 50% with the lowest JE scores.

## 3.2 Energy-based Learnable Frequency component selection

The motivation of our algorithm is that (Zhang & Zhu, 2019) demonstrates adversarial training can improve the model's adversarial robustness by shifting the model's focus from texture and color to shape and silhouette features. Frequency pruning removes textural details while preserving key shape features, helping the model focus more on shape, as shown in Fig. 1. We proposed that a carefully designed frequency pruning algorithm could potentially achieve comparable results to adversarial training, offering a resource-conservation approach to improve model robustness. Our approach adaptively selects important frequency components for each image, aiming to both smooth the model's local minimum geometry and maintain the model's capacity.

**Lemma 1** (Relationship between Frequency Alterations and local minimum geometry Smoothness). *Let $x \in \mathcal{X}$ denote an original image and $\tilde{x} \in \mathcal{X}$ denote the image after frequency pruning. Let $f_\theta$ be the model with parameters $\theta$. Set $H(\cdot)$ as the entropy function and $f_\theta(\tilde{x})$ represents the logits output by the model for input $\tilde{x}$. Let $p_i$ be the predicted probability for class $i$, computed from the logits using the softmax function:*

$$p_i = \frac{\exp(f_\theta(\tilde{x})_i)}{\sum_{j=1}^{K} \exp(f_\theta(\tilde{x})_j)} \tag{3}$$

*where $f_\theta(\tilde{x})_i$ is the $i$-th element of the logits vector $f_\theta(\tilde{x})$, and $K$ is the number of classes. The entropy of the model's output is then defined as:*

$$H(f_\theta(\tilde{x})) = -\sum_{i=1}^{K} p_i \log p_i \tag{4}$$

*We propose that the relationship between the entropy and the gradient norm can be expressed as:*

$$H(f_\theta(\tilde{x})) \propto \|\nabla_\theta \mathcal{L}(f_\theta(x), y)\| \tag{5}$$

*where $H(f_\theta(\tilde{x}))$ is the entropy of the model's output probabilities and $\|\nabla_\theta \mathcal{L}(f_\theta(x), y)\|$ is the norm of the gradient of the loss with respect to the model parameters $\theta$. Based on this relationship we suggest that lower entropy of the output probabilities leads to a smoother local minimum geometry.*

According to Lemma 1, we know that we can smooth the local minimum geometry by reducing the entropy of the model's logits (referred to as "logit entropy"). We introduce Frequency-Selective

Excitation Network (FSE-Net), a trainable model for dynamic frequency component selection. The decision-making process has three steps: 1) Compression: Global average pooling captures frequency information. 2) Motivation: A fully connected layer learns nonlinear relationships and generates importance weights. 3) Recalibration: Sigmoid-normalized weights are multiplied with the original frequency components to assign importance scores to each component. (The network structure is provided in Appendix E) and its loss function is:

$$L(\theta) = H(f_\theta(x_f)) - \lambda \left( \frac{1}{|\mathcal{D}|} \sum_{(\tilde{x}, y) \in \mathcal{D}} \mathbf{1}_{\arg\max f_\theta(x_f) = y} \right) \tag{6}$$

We aim to minimize the following loss function using gradient descent, where $x_f$ is the image after frequency pruning (applying FSE-Net to prune frequency components and then convert back to the spatial domain), and $\lambda$ is a hyperparameter. Additionally, the second term of the loss function is crucial to preserve the main features of the image to maintain the model's capacity. Let $\hat{X}_{i,j}$ be the $(i, j)$-th coefficient of the Discrete Cosine Transform (DCT) of an image $X$. We use DCT, not DFT/FFT because DCT coefficients are real, while DFT coefficients include imaginary parts, making them harder for FSE-Net to learn (Xu et al., 2020).

**Theorem 2** (Biased Learning in DCT Frequency Selection). *Let $\mathcal{F} = f_1, \ldots, f_n$ be the set of frequency components obtained after applying Discrete Cosine Transform (DCT) to an input signal, with corresponding energies $E = E_1, \ldots, E_n$. Let $\mathcal{F}_H$ and $\mathcal{F}_L$ denote the sets of high-energy and low-energy components respectively. Given a selection process $S : \mathcal{F} \to [0,1]^n$ and a loss function $L(S(\mathcal{F}))$. Considering the inherent energy disparity in DCT coefficients where:*

$$\min_{f_i \in \mathcal{F}_H, f_j \in \mathcal{F}_L} \frac{E_i}{E_j} \gg 1 \tag{7}$$

*The learning process is prone to exhibit a significant bias towards high-energy frequency components, ultimately resulting in limited representational capacity and reduced effectiveness in capturing the full spectrum of frequency information.*

Theorem 2 shows that models tend to focus on high-energy frequency features and ignore low-energy ones, resulting in suboptimal outcomes (Allen-Zhu et al., 2019). To mitigate this issue, we fix the selection of high-energy components and focus our learnable selection process on low-energy frequency components. Define the energy of each frequency component as $E(i, j) = |\hat{X}_{i,j}|^2$. Let $E^{(1)} \geq E^{(2)} \geq \cdots \geq E^{(n)}$ be the sorted energies of all frequency components, Let $E^{(k)}$ represent the energy of the $k$-th highest frequency component. The frequency component selection mechanism of FSE-Net, denoted as $F_{sel}$, can be modeled as:

$$F_{sel}(X_c, E_c; \theta_F, k, k_{total}) = \begin{cases} 1 & \text{if } E_c \geq E^{(k)} \text{ or } (E_c < E^{(k)} \text{ and } g(X_c; \theta_F) \geq s_{k_{total}-k}) \\ 0 & \text{otherwise} \end{cases} \tag{8}$$

where $\theta_F$ represents the set of learnable parameters from FSE-Net, $E_c$ denotes the energy of the frequency component $X_c$, and $g(X_c; \theta_F)$ is implemented as a neural network with sigmoid activation in the final layer. It takes $X_c$ as input and outputs an importance score. $s_{k_{total}-k}$ is the $(k_{total}-k)$-th highest score among the components with $E_c < E^{(k)}$ where $k_{total}$ is the total number of frequency components we want to preserve.

The selection process directly chooses the top $k$ frequency components with the highest energy, then selects the top $k_{total} - k$ components with the highest importance scores from the remaining lower-energy components. The frequency component $X_c$ is retained when $F_{sel}(X_c, E_c; \theta_F, k, k_{total}) = 1$. This mechanism ensures that FSE-Net can thoroughly learn both high-energy and low-frequency features. The number of $k$ chosen through ablation experiments in Fig. 4c. Fig. 3a illustrates how our frequency pruning algorithm smooths the model's local minimum geometry, leading to enhanced adversarial robustness. The algorithm flow is shown in Appendix A.

### 3.3 Enhance Coreset robustness by reward score

In this section, we propose a novel coreset selection algorithm that assesses the impact of images on shaping the decision surface throughout training. Drawing from reinforcement learning (Kaelbling et al., 1996) and Markov decision processes (Hordijk & Kallenberg, 1979), we assign each image a reward based on its actions during training and calculate the accumulated reward as its final score.

To track the temporal dynamics of model parameter changes, we define $H(f_\theta(\tilde{x}))_t$ as the logit entropy at epoch $t$. Our method aims to balance exploration in the early stages and exploitation in the later stages of training, as suggested by (Petzka & Sminchisescu, 2021). In the early training stage, we encourage the model to explore a larger parameter space to capture more features and escape local minima (Soloperto et al., 2023). Later in training, we aim to reduce gradient variance, signaling stable optimization toward the global minimum. To quantify this balance, we introduce a local variance function $V(t, w)$:

$$V(t, w) = \text{Var}\left(\left\{H(f_\theta(\tilde{x}))_i \mid \max(0, t - \frac{w}{2}) \le i < \min(T, t + \frac{w}{2} + 1)\right\}\right) \quad (9)$$

Where $T$ is the total number of epochs and $w$ is the window size for local variance calculation.

To encourage initial exploration followed by convergence, we design a reward function $R(t, V(t, w))$:

$$R(t, V(t, w)) = \begin{cases} -V(t, w) & \text{if } t < \tau T \\ +V(t, w) & \text{if } t \ge \tau T \end{cases} \quad (10)$$

Where $\tau \in (0, 1)$ determines the transition point between the exploration and exploitation phases. This reward function assigns different rewards to images based on their behavior at each stage, ensuring alignment with our optimization goals. In the early stage ($t < \tau T$), negative rewards for low variance encourage the exploration of a larger parameter space. In the later stage ($t \ge \tau T$), positive rewards for high variance promote convergence to smooth local minima, we set $\tau = 2/3$ through ablation experiments. The overall optimization objective is captured by a cumulative discounted reward $S$:

$$S = \sum_{t=0}^{T-1} \gamma^t R(t, V(t, w)) + \gamma^T R_T \quad (11)$$

Here, the image with a lower score is considered more important, $\gamma \in (0, 1)$ is a discount factor prioritizing more recent rewards, we set $\gamma = 0.99$ similar to the setting in many tasks in Reinforcement Learning (Yoshida et al., 2013). $R(t, V(t, w))$ represents the reward at time step $t$. The term $R_T$ is a terminal reward defined as:

$$R_T = \text{Var}(\{H(f_\theta(\tilde{x}))_0, H(f_\theta(\tilde{x}))_1, ..., H(f_\theta(\tilde{x}))_{T-1}\}) \quad (12)$$

The terminal reward $R_T$ is based on the model's entropy variance across all epochs, accounting for the stability of the entire training optimization. We ranked the CIFAR-10 images by their scores in ascending order. Fig.3c shows the model trained on the top 50% of images (those with the lowest scores), while Fig.3b shows the model trained on the bottom 50% of images (those with the highest scores). We observe two key differences between these models: 1) Generation Loss Behavior: In the model trained on the top 50% of images, the generation loss exhibits greater fluctuations before epoch 350 (about two-thirds of total epochs), followed by a smoother trajectory. This suggests an initial exploration of a larger parameter space before converging to a smooth global minimum. 2) Final Performance: The model trained on the top 50% achieves a lower final generation loss compared to the other model. This indicates better overall performance and improved generalization capability. To enhance coreset diversity, we employ a stratified sampling algorithm as proposed by Zheng et al. (2022). This method involves ranking images based on their scores in ascending order, followed by the application of stratified sampling to select the final coreset.

| Pruning Algorithm | Attack | Prune Rate | | | | | | | | | |
|---|---|---|---|---|---|---|---|---|---|---|---|
| | | CIFAR-10 | | | | | CIFAR-100 | | | | |
| | | 90% | 80% | 70% | 60% | 50% | 90% | 80% | 70% | 60% | 50% |
| Random | AA | 15.27 | 21.55 | 20.85 | 22.33 | 21.53 | 12.27 | 16.55 | 16.88 | 17.03 | 17.57 |
| | PGD-20 | 16.27 | 20.55 | 19.85 | 21.33 | 20.53 | 11.97 | 15.59 | 16.78 | 17.23 | 16.57 |
| | C&W | 16.39 | 20.45 | 18.83 | 20.53 | 20.77 | 12.97 | 14.59 | 15.78 | 16.23 | 14.57 |
| Entropy | AA | 21.65 | 21.27 | 30.92 | 23.28 | 20.74 | 11.56 | 14.33 | 17.98 | 15.33 | 17.71 |
| | PGD-20 | 20.68 | 20.87 | 20.92 | 21.28 | 22.74 | 12.16 | 14.03 | 16.18 | 15.13 | 17.01 |
| | C&W | 20.44 | 20.78 | 20.21 | 21.17 | 22.83 | 12.44 | 12.35 | 17.18 | 16.37 | 17.51 |
| CCSFEM | AA | 37.86 | 40.98 | 41.02 | 40.18 | 41.28 | 15.11 | 16.85 | 18.05 | 18.19 | 17.92 |
| | PGD-20 | 38.97 | 40.11 | 39.99 | 39.76 | 41.91 | 13.98 | 15.92 | 13.09 | 14.08 | 17.91 |
| | C&W | 38.99 | 40.33 | 40.02 | 39.96 | 42.05 | 12.17 | 16.15 | 17.91 | 18.60 | 17.27 |
| Ours-JE | AA | 40.16 | 44.32 | 42.97 | 41.38 | 40.65 | 16.15 | 17.12 | 21.37 | 20.09 | 18.65 |
| | PGD-20 | 39.16 | 39.32 | 41.07 | 40.88 | 42.95 | 14.76 | 16.88 | 17.01 | 16.89 | 18.05 |
| | C&W | 39.06 | 39.72 | 41.37 | 40.96 | 43.05 | 12.99 | 17.32 | 18.07 | 19.88 | 18.95 |
| Ours-LF | AA | **55.7** | **58.19** | **53.46** | **51.14** | **51.04** | **20.99** | **24.94** | **25.07** | **25.31** | **23.41** |
| | PGD-20 | **56.18** | **56.05** | **51.9** | **50.61** | **50.07** | **23.72** | **21.64** | **21.36** | **24.39** | **23.75** |
| | C&W | **56.42** | **56.21** | **54.14** | **54.38** | **55.32** | **22.37** | **24.23** | **25.48** | **26.39** | **28.69** |
| Ours-JELF | AA | 46.54 | 47.35 | 48.89 | 49.72 | 48.18 | 20.39 | 20.53 | 22.47 | 22.94 | 22.85 |
| | PGD-20 | 47.24 | 48.59 | 49.64 | 50.25 | 50.01 | 18.09 | 19.71 | 20.98 | 21.85 | 22.48 |
| | C&W | 47.61 | 48.12 | 49.01 | 48.19 | 48.66 | 18.99 | 20.01 | 21.54 | 22.03 | 21.99 |

Table 1: We assess CIFAR-10 and CIFAR-100 performance under various adversarial attacks and dataset pruning ratios. On CIFAR-10, accuracy is 43.86% for AutoAttack, 42.83% for PGD-20, and 43.65% for C&W. On CIFAR-100, accuracy is 18.51% under AutoAttack, 18.59% under PGD-20, and 19.57% under C&W. "CCSFEM" uses forgetting, EL2N, and AUM scores with CCS to compute the mean accuracy. "Ours-JE" applies the joint-entropy score with CCS sampling, "Ours-LF" uses Learnable Frequency Pruning on the total dataset, and "Ours-JELF" combines Learnable Frequency Pruning (preserving 50% of frequency components) with joint-entropy based coreset selection using CCS sampling.

## 4 EXPERIMENTS

### 4.1 EXPERIMENTAL SETUP

We use three datasets: CIFAR-10, CIFAR-100, and ImageNet-1K. All attacks were constrained by the $\ell_\infty$ norm with a perturbation budget of $\epsilon = \frac{8}{255}$. For CIFAR-10 and CIFAR-100, we trained ResNet-18 models from scratch. We applied three different attack algorithms on the entire test sets: AutoAttack (AA) (Croce & Hein, 2020), PGD-20 (Madry, 2017), and C&W attack (Carlini & Wagner, 2017). For PGD-20, we used 20 iterations with a step size of $\epsilon = \frac{2}{255}$. For C&W, we used 100 iterations with a learning rate of 0.01. AutoAttack was applied with its default settings. For ImageNet-1K, we trained a ResNet-34 model and evaluated the robustness using AutoAttack on 1000 randomly selected points from the validation set. All datasets were normalized before feeding into the models, and standard data augmentations were applied.

### 4.2 BASELINES

Since our work tackles a less-studied problem of high adversarial-robustness dataset pruning with no known clear solution, it is important to set an adequate baseline for comparison. We compare our approaches with six original dataset pruning algorithms: 1) **Random**: Uniform random sampling. 2) **Entropy**: Selects highest entropy examples. 3) **Forgetting**: Chooses examples with highest Forgetting scores. 4) **EL2N**: Selects based on highest EL2N scores. 5) **AUM**: Chooses examples with highest Area Under the Margin scores. 6) **CCS**: Uses stratified sampling across importance scores. These algorithms test the result without adversarial attack, providing a comparison baseline for our work.

In this experiment, we apply various pruning methods. For "Random", "Entropy", "CCSEFM", "Ours-JE", and "Ours-JELF", we employ sample-wise pruning, where the pruning rate represents the percentage of images removed from the dataset. For "Ours-LF", we use frequency-domain

pruning, where the pruning rate indicates the percentage of frequency components removed from each image.

|  | Original Adversarial Training | | Sample Adversarial Training | | pre-trained Adversarial Training | |
|---|---|---|---|---|---|---|
|  | 50% coreset | Original Dataset | 50% coreset | Original Dataset | 50% coreset | Original Dataset |
| **TDFAT** | Acc:45.41 Time: 59.58s | Acc: 48.66 Time: 119.69s | Acc:30.05 Time: 18.53s | Acc: 38.72 Time: 39.69s | Acc:35.41 Time: 10.15s | Acc: 40.66 Time: 17.66s |
| **CURC** | Acc: 44.92 Time:74.12s | Acc: 52.48 Time: 149.52s | Acc: 31.21 Time:17.87s | Acc: 36.05 Time:38.52s | Acc:38.92 Time:9.31s | Acc:41.48 Time: 17.92s |
| **RATTE** | Acc: 45.98 Time: 64.33s | Acc: 48.2 Time: 131.35s | Acc: 34.05 Time: 17.23s | Acc: 37.15 Time: 38.35s | Acc:36.23 Time: 10.21s | Acc:41.03 Time: 17.85s |
| **FATSC** | Acc: 17.21 Time: 59.82s | Acc: 23.68 Time: 122.71s | Acc: 20.82 Time: 17.12s | Acc: 23.77 Time: 39.71s | Acc:12.22 Time: 9.32s | Acc:16.93 Time:18.87s |
| **ours-LF** | Acc:48.18 Time:9.51s | Acc:51.04 Time:17.64s | Acc:50.07 Time:9.51s | Acc:51.04 Time:17.64s | Acc:50.01 Time:9.69s | Acc:48.18 Time:17.64s |

Table 2: We compare recent adversarial training (AT) algorithms with our Learnable Frequency Pruning method. "Time" refers to the average time required per epoch to train ResNet-18 on the same batch size and GPU. Adversarial robustness was evaluated against AutoAttack using both the full CIFAR-10 training set and a 50% coreset selected by the "Ours-JE" strategy. "Original Adversarial Training" applies standard AT on the entire dataset, while "Sample Adversarial Training" applies adversarial perturbations to a random subset of images each epoch, leaving the rest unchanged to match our method's training cost. Finally, " pre-trained Adversarial Training" uses a pre-trained ResNet-18 model with high adversarial robustness to generate adversarial perturbations without further optimization during training, ensuring no additional Time. We train datasets of the same size for an equal number of epochs under identical conditions.

## 4.3 PERFORMANCE COMPARISON

Table **??** shows the performance of our algorithm on CIFAR-10 and CIFAR-100 under different adversarial attacks. We observe the following key findings: 1) The state-of-the-art (SOTA) dataset pruning algorithm demonstrates low accuracy under adversarial attacks (e.g., only 41.28% with a 50% pruning ratio under AutoAttack in CIFAR-10). 2) Without frequency pruning, the "ours-JE" approach significantly outperforms SOTA algorithms across all pruning ratios and adversarial attacks. 3) On the original dataset, "Ours-LF" achieves better performance than without pruning, indicating that it not only enhances robustness against adversarial attacks but also reduces storage costs. 4) On pruned datasets, "Ours-JELF" outperforms SOTA pruning methods, highlighting the ability of our approach to improve the adversarial robustness of the model in dataset pruning scenarios. Table 3 presents similar findings, demonstrating the effectiveness of our algorithm in enhancing adversarial robustness on the model trained on the ImageNet-1K dataset.

## 4.4 ABLATION EXPERIMENT

To demonstrate the effectiveness of our Learnable Frequency Pruning algorithm in enhancing model robustness, we compared it with recent adversarial training methods: TDFAT (Tong et al., 2024), RATTE (Jin et al., 2023), FATSC (Zhao et al., 2023), and CURC (Gowda et al., 2024). These methods, which involve iterative optimization, significantly increase training costs. To ensure a fair comparison, we adjusted the number of perturbed images to match the training time of these methods with our algorithm, and we also used a pre-trained ResNet-18 model with high adversarial robustness to generate perturbations without further optimization. As shown in Table 2, while adversarial training on the full dataset provides better performance, it greatly increases training costs. When the cost is reduced to match our algorithm, the performance of adversarial training drops significantly, making it less suitable for resource-limited environments. We also evaluated the impact of different values of $\tau T$ (Fig. 4a), $\lambda$ (Fig. 4b), and $k$ (Fig. 4c), along with various coreset selection strategies combined with Learnable Frequency Pruning (Fig. 5). The best results from our experiments are presented.

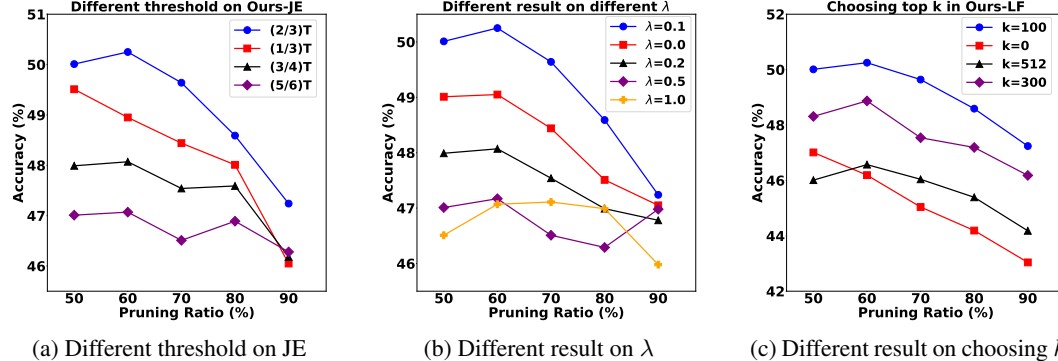

(a) Different threshold on JE     (b) Different result on $\lambda$     (c) Different result on choosing $k$

Figure 4: (a) evaluates the time threshold $\tau T$ from Section 3.3, adjusting $\tau T$ to select coresets and prune 50% of frequency components, let PGD-20 as an adversarial attack. Setting the threshold to $(2/3)T$ yields the best result. (b) examines the effect of adjusting $\lambda$ in Section 3.2, which controls the loss function in Learnable Frequency Pruning. Using "Ours-JE" for coreset selection and pruning 50% of frequency components, PGD-20 as an adversarial attack, we find that $\lambda = 0.1$ yields the best result. (c) compares different values of $k$ from Section 3.2 using "Ours-JE" for coreset selection and pruning 50% of frequency components, let PGD-20 as an adversarial attack. The best performance is achieved with $k = 100$, using PGD-20 as the adversarial attack.

|  | Pruning Ratio | | | | |
|---|---|---|---|---|---|
|  | 90% | 80% | 70% | 60% | 50% |
| Random | 15.87 | 20.51 | 20.15 | 18.39 | 18.58 |
| Entropy | 16.87 | 18.06 | 21.02 | 18.11 | 17.28 |
| CCSFEM | 15.86 | 16.96 | 18.02 | 17.11 | 19.28 |
| ours-JE | 19.16 | 21.51 | 22.17 | 21.18 | 20.96 |
| ours-LF | **25.6** | **27.19** | **26.46** | **27.44** | **27.14** |
| ours-JELF | **23.54** | **26.15** | **25.03** | **25.22** | **24.95** |

Table 3: Performance on Imagenet-1K using different pruning strategies. The original dataset accuracy is 20.16% under AutoAttack. "Ours-JE" refers to coreset selection using the joint-entropy score with CCS sample strategy, while "Ours-LF" applies Learnable Frequency Pruning. "Ours-JELF" combines Learnable Frequency Pruning (preserving 50% of frequency components) with coreset selection using the joint-entropy score with the CCS sample strategy.

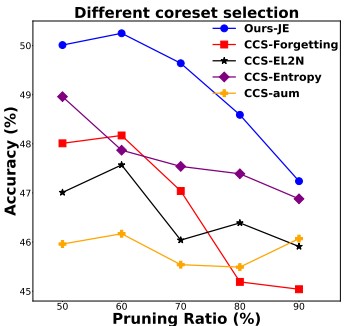

Figure 5: Compares different coreset selection algorithms on CIFAR-10, followed by Learnable Frequency Pruning with a 50% pruning ratio, PGD-20 as an adversarial attack. "Ours-JE" achieves the best performance.

## 5    CONCLUSION, LIMITATION, AND FUTURE WORK

We introduce Adversarial Attack Robust Dataset Pruning, a method that enhances model robustness against adversarial attacks on pruned datasets. Our approach improves robustness in two ways: First, we use a Learnable Frequency Pruning algorithm to smooth the model's local minimum geometry, increasing robustness without additional training costs and reducing storage needs. Second, we propose a "joint-entropy" data importance score for better coreset selection. Experiments show our method surpasses existing pruning strategies in adversarial robustness across various datasets and attacks. This work is the first to address adversarial robustness in dataset pruning.

We recognize several limitations and areas for future work. First, our algorithm does not consider the link between image distribution and adversarial robustness, which could be explored further. Improving the sampling method beyond the traditional CCS algorithm may enhance results. Additionally, while we focus on adversarial robustness in dataset pruning, the research could extend to areas like Dataset Distillation and Neural Network pruning to improve the model's adversarial robustness.

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

# A ALGORITHM FLOW

Algorithm 1 shows the algorithm flow of FSE-Net, Figure 6 shows the whole process of how to combine FSE-Net and the coreset selection to get the final coreset.

---

**Algorithm 1** Frequency Selection with FSE-Net for Improved Robustness

---

**Require:** Images $\mathcal{X}$, Labels $\mathcal{Y}$, Model $f_\theta$, Parameters $k$, $k_{total}$, Learning rate $\eta$, Hyperparameter $\lambda$, The $(i, j)$-th DCT coefficient $\hat{X}_{i,j}$.

1: $X_f \leftarrow \text{DCT}(\mathcal{X})$
2: Calculate energy for each frequency component: $E(i, j) = |\hat{X}_{i,j}|^2$
3: Sort frequency components based on energy $E$ and select top-$k$ components
4: Initialize FSE-Net parameters $\theta_F^{(0)}$ for selecting remaining frequency components
5: **for** $t = 0$ to $T - 1$ **do**
6: $\quad$ Define $F_{sel}(X_c, E_c; \theta_F^{(t)}, k, k_{total})$:
7: $\quad\quad$ if $E_c \geq E^{(k)}$ or $(E_c < E^{(k)}$ and $g(X_c; \theta_F^{(t)}) \geq s_{k_{total}-k})$:
8: $\quad\quad\quad$ return 1
9: $\quad\quad$ else:
10: $\quad\quad\quad$ return 0
11: $\quad$ $\tilde{x} \leftarrow \text{IDCT}(X_f \odot F_{sel})$
12: $\quad$ Compute logits: $f_\theta(\tilde{x}) \leftarrow f_\theta(\tilde{x})$
13: $\quad$ Compute probabilities: $p_i = \frac{\exp(f_\theta(\tilde{x})_i)}{\sum_{j=1}^{K} \exp(f_\theta(\tilde{x})_j)}$
14: $\quad$ Compute entropy: $H(f_\theta(\tilde{x})) = -\sum_{i=1}^{K} p_i \log p_i$
15: $\quad$ $L(\theta_F^{(t)}) \leftarrow H(f_\theta(\tilde{x})) - \lambda \left( \frac{1}{|\mathcal{D}|} \sum_{(\tilde{x},y)\in\mathcal{D}} \mathbf{1}_{\arg\max f_\theta(\tilde{x})=y} \right)$
16: $\quad$ $\theta_F^{(t+1)} \leftarrow \theta_F^{(t)} - \eta \nabla_{\theta_F} L(\theta_F^{(t)})$
17: **end for**
18: Define final selection function $F_{sel}^{final}$ using $\theta_F^{(T)}$
19: $\tilde{\mathcal{X}} \leftarrow \{\text{IDCT}(X_f \odot F_{sel}^{final}) | X_f \in \text{DCT}(\mathcal{X})\}$
20: **Return** $\tilde{\mathcal{X}}, \mathcal{Y}$

---

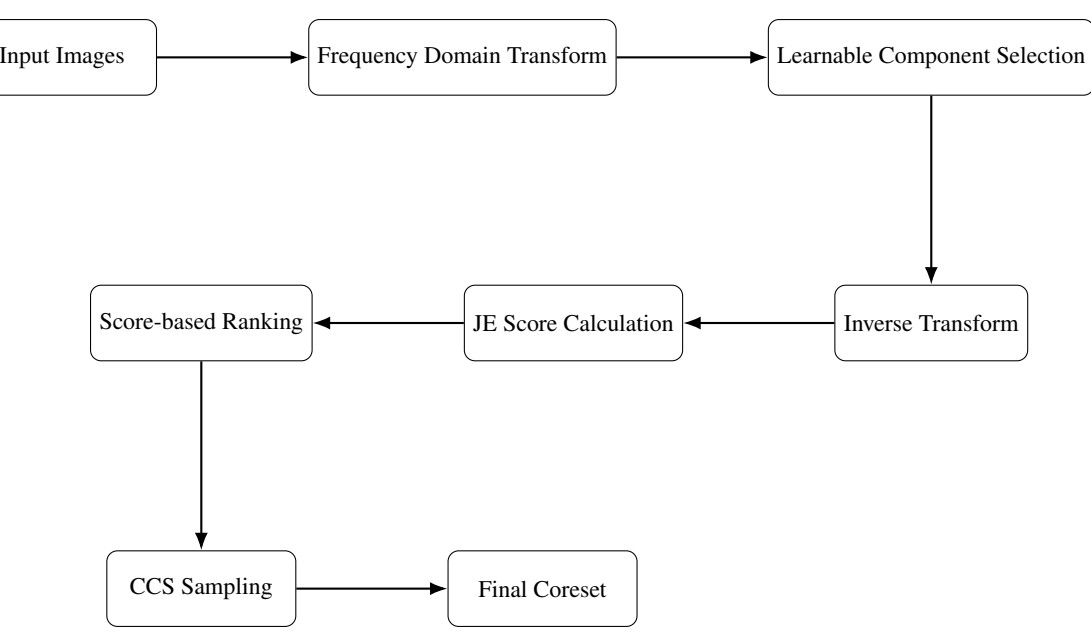

Figure 6: Flowchart of the coreset generation process.

## B  Experiment Setting

In this section, we will show the details of our algorithm. For the Learnable frequency pruning experiments, we set (Pruning ratio, Total Number of preserved frequency components, top k, $\lambda$) to show the setting details. For CIFAR-10 and dataset, our settings are (90%, 102, 30, 0.05) (80%, 204, 50, 0.01) (70%, 308, 80, 0.08) (60%, 410, 100, 0.1) (50%, 512, 100, 0.1) (30%, 717, 100, 0.05), the Learnable pruning ratio will be optimized for 300 epochs and we set the batch size 128. We trained We use ResNet18 (He et al., 2016) as the network architecture for CIFAR-10. We train the whole dataset with 200 epochs with a 256 batch size. We use the SGD optimizer (0.9 momentum and 0.0002 weight decay) with a 0.1 initial learning rate. For the "ours-JE" algorithm, we calculated the logit entropy every five epochs, we set the time threshold as $(2/3)T$ and the size of the window to 3, we used the setting of CCS same as the Original paper's setting (Zheng et al., 2022).

For CIFAR-100, our settings are (90%, 102, 30, 0.1) (80%, 204, 50, 0.08) (70%, 308, 80, 0.13) (60%, 410, 100, 0.15) (50%, 512, 100, 0.13) (30%, 717, 100, 0.09), the Learnable pruning ratio will be optimized for 300 epochs and we set the batch size 128. We trained We use ResNet18 (He et al., 2016) as the network architecture for CIFAR-10. We train the whole dataset with 200 epochs with a 256 batch size. We use the SGD optimizer (0.9 momentum and 0.0002 weight decay) with a 0.1 initial learning rate. For the "ours-JE" algorithm, we calculated the logit entropy every five epochs, we set the time threshold as $(2/3)T$ and the size of the window to 3, we used the setting of CCS same as the Original paper's setting (Zheng et al., 2022).

For Imagenet-1K, because the input images are sized 224 * 224 *3, For every channel, our settings are (90%, 5018, 600, 0.05) (80%, 10036, 1000, 0.02) (70%, 15053, 1000, 0.05) (60%, 20071, 1000 ,0.1) (50%, 25090, 1000, 0.1) (30%, 35123, 1000,0.1) We use ResNet34 to train the dataset and We use the SGD optimizer (0.9 momentum and 0.0001 weight decay) with a 0.1 initial learning rate. The learning rate scheduler is the cosine annealing learning rate scheduler. For the "ours-JE" algorithm, we calculated the logit entropy every two epochs, we set the time threshold as $(2/3)T$ and the size of the window to 3, and we used the setting of CCS same as the Original paper's setting (Zheng et al., 2022)

## C  Proof

### C.1  Proof of Therom 1

**Theorem 1** (Hard Samples and Local Minimum Geometry). *Formally, for a model with parameters $\theta$ and loss function $\mathcal{L}$, we compare the local minimum geometry at the converged solution $\theta^*$ for hard samples $x_h$ and randomly sampled data points $x_r$. For a given perturbation $\varepsilon$, where $\|\varepsilon\| \leq \delta$ for some small $\delta > 0$, hard samples are more likely to induce less smooth geometries, which can be characterized as:*

$$\mathbb{E}_{x_h}[\Delta\mathcal{L}_h] > \eta \cdot \mathbb{E}_{x_r}[\Delta\mathcal{L}_r] \tag{13}$$

*where $\Delta\mathcal{L}_h = |\mathcal{L}(\theta^* + \varepsilon, x_h) - \mathcal{L}(\theta^*, x_h)|$, $\Delta\mathcal{L}_r = |\mathcal{L}(\theta^* + \varepsilon, x_r) - \mathcal{L}(\theta^*, x_r)|$, and $\eta > 1$ is a threshold constant. The expectation $\mathbb{E}_{x_h}$ is taken over the distribution of hard samples, while $\mathbb{E}_{x_r}$ is taken over the random sample data distribution.*

Proof:

We begin by defining hard samples. For a hard sample $x_h$ and a randomly sampled data point $x_r$, we assume:

$$\frac{\|\nabla_{w^*}\mathcal{L}(w^*, x_h)\|}{\mathbb{E}_{x_r}[\|\nabla_{w^*}\mathcal{L}(w^*, x_r)\|]} \geq \gamma \tag{14}$$

where $\gamma > 1$ is a threshold constant and $w^*$ represents the converged model parameters.

To establish the relationship between gradients and the local geometry of the loss surface, we introduce the concept of directional derivatives. For any sample $x$ and unit vector $u$, the directional derivative of the loss function is defined as:

$$D_u\mathcal{L}(w^*, x) = \lim_{t \to 0} \frac{\mathcal{L}(w^* + tu, x) - \mathcal{L}(w^*, x)}{t} \tag{15}$$

This directional derivative is related to the gradient through the following equation:

$$D_u\mathcal{L}(w^*, x) = \nabla_{w^*}\mathcal{L}(w^*, x) \cdot u \tag{16}$$

Now, consider a small perturbation $\varepsilon$ applied to the parameters $w^*$. We define a unit vector $u = \frac{\varepsilon}{\|\varepsilon\|}$ in the direction of this perturbation. Using this, we can approximate the change in loss due to the perturbation:

$$\Delta\mathcal{L} \approx |\nabla_{w^*}\mathcal{L}(w^*, x) \cdot \varepsilon| = \|\varepsilon\||D_u\mathcal{L}(w^*, x)| \tag{17}$$

Applying the expectation operator to both sides:

$$\mathbb{E}_x[\Delta\mathcal{L}] \approx \|\varepsilon\|\mathbb{E}_x[|D_u\mathcal{L}(w^*, x)|] \tag{18}$$
$$\leq \|\varepsilon\|\mathbb{E}_x[\|\nabla_{w^*}\mathcal{L}(w^*, x)\|] \quad \text{(by Cauchy-Schwarz inequality)} \tag{19}$$

From our initial assumption, we can state:

$$\mathbb{E}_{x_h}[\|\nabla_{w^*}\mathcal{L}(w^*, x_h)\|] \geq \gamma \cdot \mathbb{E}_{x_r}[\|\nabla_{w^*}\mathcal{L}(w^*, x_r)\|] \tag{20}$$

Combining these results, we obtain:

$$\mathbb{E}_{x_h}[\Delta\mathcal{L}_h] \approx \|\varepsilon\|\mathbb{E}_{x_h}[\|\nabla_{w^*}\mathcal{L}(w^*, x_h)\|] \tag{21}$$
$$\geq \|\varepsilon\|\gamma \cdot \mathbb{E}_{x_r}[\|\nabla_{w^*}\mathcal{L}(w^*, x_r)\|] \tag{22}$$
$$\approx \gamma \cdot \mathbb{E}_{x_r}[\Delta\mathcal{L}_r] \tag{23}$$

Since $\gamma > 1$, we can choose $\eta = \gamma - \epsilon$ for some small $\epsilon > 0$, ensuring $\eta > 1$. This allows us to conclude:

$$\mathbb{E}_{x_h}[\Delta\mathcal{L}_h] > \eta \cdot \mathbb{E}_{x_r}[\Delta\mathcal{L}_r] \tag{24}$$

This result holds for sufficiently small $\|\varepsilon\|$, where our approximations remain accurate.

Thus, we have demonstrated that for sufficiently small perturbations, the expected change in loss for hard samples is more than $\eta$ times the expected change for randomly sampled data points, where $\eta > 1$. This indicates that hard samples induce less smooth geometries in the vicinity of local minima, consistent with the statement of the theorem.

This proof highlights the unique characteristics of hard samples relative to the entire data distribution (as represented by random sampling), rather than just in comparison to easy samples. It emphasizes the importance of hard samples in the model training process and their impact on the geometry of local minima.

## C.2 PROOF OF LEMMA 1

**Lemma 1** (Relationship between Frequency Alterations and local minimum geometry Smoothness). *Let $x \in \mathcal{X}$ denote an original image and $\tilde{x} \in \mathcal{X}$ denote the image after frequency pruning. Let $f_\theta$ be the model with parameters $\theta$. Set $H(\cdot)$ as the entropy function and $f_\theta(\tilde{x})$ represents the logits output by the model for input $\tilde{x}$. Let $p_i$ be the predicted probability for class $i$, computed from the logits using the softmax function:*

$$p_i = \frac{\exp(f_\theta(\tilde{x})_i)}{\sum_{j=1}^{K} \exp(f_\theta(\tilde{x})_j)} \tag{25}$$

*where $f_\theta(\tilde{x})_i$ is the $i$-th element of the logits vector $f_\theta(\tilde{x})$, and $K$ is the number of classes. The entropy of the model's output is then defined as:*

$$H(f_\theta(\tilde{x})) = -\sum_{i=1}^{K} p_i \log p_i \tag{26}$$

*We propose that the relationship between the entropy and the gradient norm can be expressed as:*

$$H(f_\theta(\tilde{x})) \propto \|\nabla_\theta \mathcal{L}(f_\theta(x), y)\| \tag{27}$$

*where $H(f_\theta(\tilde{x}))$ is the entropy of the model's output probabilities and $\|\nabla_\theta \mathcal{L}(f_\theta(x), y)\|$ is the norm of the gradient of the loss with respect to the model parameters $\theta$. Based on this relationship we suggest that lower entropy of the output probabilities leads to a smoother local minimum geometry.*

Proof:

To prove Lemma 1, we begin with the cross-entropy loss function:

$$\mathcal{L}(f_\theta(x), y) = -\sum_{i=1}^{K} y_i \log p_i \tag{28}$$

where $y_i$ is the one-hot encoded true label, and $p_i$ is the predicted probability for class $i$.

The gradient of this loss with respect to the logits $f_\theta(\tilde{x})_j$ is:

$$\frac{\partial \mathcal{L}}{\partial f_\theta(\tilde{x})_j} = p_j - y_j \tag{29}$$

Now, consider the entropy of the model's output:

$$H(f_\theta(\tilde{x})) = -\sum_{i=1}^{K} p_i \log p_i \tag{30}$$

We observe that when the model is very confident (low entropy), one $p_i$ will be close to 1 and the rest close to 0. In this case, both the entropy and the gradient norm will be small. Conversely, when the model is uncertain (high entropy), the $p_i$ values will be more evenly distributed, resulting in larger values for both the entropy and the gradient norm.

To illustrate this more formally, let's consider the extreme cases:

1. Maximum certainty: One $p_i = 1$, rest are 0
   - Entropy: $H = 0$
   - Gradient: $\|\nabla_{f_\theta(\tilde{x})} \mathcal{L}\| = 0$ (assuming correct prediction)
2. Maximum uncertainty: All $p_i = \frac{1}{K}$
   - Entropy: $H = \log K$ (maximum)
   - Gradient: $\|\nabla_{f_\theta(\tilde{x})} \mathcal{L}\| = \sqrt{\sum_{j=1}^{K} (\frac{1}{K} - y_j)^2}$ (maximum)

These extreme cases demonstrate that as entropy increases, so does the gradient norm.

Furthermore, we can express the gradient norm as:

$$\|\nabla_{f_\theta(\tilde{x})}\mathcal{L}\|^2 = \sum_{j=1}^{K}(p_j - y_j)^2 = \sum_{j=1}^{K}p_j^2 - 2p_y + 1 \qquad (31)$$

Note that $\sum_{j=1}^{K}p_j^2$ is minimized when all $p_j$ are equal (high entropy) and maximized when one $p_j$ is 1 and the rest are 0 (low entropy), which aligns with the behavior of the entropy.

Finally, by the chain rule, $\nabla_\theta\mathcal{L} = \frac{\partial f_\theta(\tilde{x})}{\partial\theta}\nabla_{f_\theta(\tilde{x})}\mathcal{L}$. Assuming $\frac{\partial f_\theta(\tilde{x})}{\partial\theta}$ is bounded, we can conclude:

$$H(f_\theta(\tilde{x})) \propto \|\nabla_\theta\mathcal{L}(f_\theta(x), y)\| \qquad (32)$$

This establishes a proportional relationship between the entropy of the model's output and the norm of the gradient of the loss with respect to the model parameters. Given that the gradient norm represents the rate of change of the loss function at a specific point, a higher gradient norm indicates a steeper loss function surface. Consequently, the geometry of the local minimum becomes more precipitous.

Conversely, lower entropy of the output probabilities leads to smaller gradient norms, resulting in a smoother local minimum geometry. This proves the relationship proposed in Lemma 1.

## C.3   PROOF OF THEOREM 2

**Theorem 2** (Biased Learning in DCT Frequency Selection). *Let $\mathcal{F} = f_1, \ldots, f_n$ be the set of frequency components obtained after applying Discrete Cosine Transform (DCT) to an input signal, with corresponding energies $E = E_1, \ldots, E_n$. Let $\mathcal{F}_H$ and $\mathcal{F}_L$ denote the sets of high-energy and low-energy components respectively. Given a selection process $S : \mathcal{F} \to [0, 1]^n$ and a loss function $L(S(\mathcal{F}))$, and considering the inherent energy disparity in DCT coefficients where:*

$$\min_{f_i \in \mathcal{F}_H, f_j \in \mathcal{F}_L} \frac{E_i}{E_j} \gg 1 \qquad (33)$$

*The learning process is prone to exhibit a significant bias towards high-energy frequency components, ultimately resulting in limited representational capacity and reduced effectiveness in capturing the full spectrum of frequency information.*

Proof:

We will prove this theorem by demonstrating that the gradient of the loss function with respect to the selection process is biased towards high-energy components, leading to their preferential selection.

Let $S(\mathcal{F}) = [s_1, ..., s_n]$ where $s_i \in [0, 1]$ represents the selection probability for frequency component $f_i$. The loss function $L(S(\mathcal{F}))$ can be expressed as a function of these selection probabilities.

Consider the gradient of the loss function with respect to the selection probabilities:

$$\nabla_S L = \left[\frac{\partial L}{\partial s_1}, ..., \frac{\partial L}{\partial s_n}\right] \qquad (34)$$

Now, let's examine the impact of selecting a frequency component on the reconstructed signal. The contribution of a frequency component $f_i$ to the reconstructed signal is proportional to its energy $E_i$. Therefore, we can express the partial derivative of the loss with respect to $s_i$ as:

$$\frac{\partial L}{\partial s_i} \propto E_i \cdot g_i \qquad (35)$$

where $g_i$ is some function of the frequency component that depends on the specific loss function used.

Given the energy disparity stated in the theorem:

$$\min_{f_i \in \mathcal{F}_H, f_j \in \mathcal{F}_L} \frac{E_i}{E_j} \gg 1 \tag{36}$$

We can conclude that for any pair of components $f_i \in \mathcal{F}_H$ and $f_j \in \mathcal{F}_L$:

$$\left| \frac{\partial L}{\partial s_i} \right| \gg \left| \frac{\partial L}{\partial s_j} \right| \tag{37}$$

This inequality holds true unless the function $g_i$ heavily penalizes high-energy components, which is unlikely in most practical loss functions designed for signal reconstruction or classification tasks.

As a result, during the optimization process, the selection probabilities for high-energy components will be updated more aggressively compared to low-energy components:

$$\Delta s_i \gg \Delta s_j, \quad \forall f_i \in \mathcal{F}_H, f_j \in \mathcal{F}_L \tag{38}$$

Over multiple iterations, this leads to:

$$s_i \gg s_j, \quad \forall f_i \in \mathcal{F}_H, f_j \in \mathcal{F}_L \tag{39}$$

This bias in the selection process results in the preferential selection of high-energy components, while low-energy components are largely ignored or underrepresented.

The consequence of this biased selection is twofold:

1. Limited Representational Capacity: By predominantly selecting high-energy components, the model fails to capture the fine details and nuances often represented by low-energy components. This limits the model's ability to represent complex patterns in the data.

2. Reduced Effectiveness: The model's focus on high-energy components may lead to overfitting dominant features while missing subtle but potentially important information in the low-energy spectrum. This can result in reduced generalization capability and overall effectiveness of the model.

Therefore, we have proven that the learning process in DCT frequency selection, given the inherent energy disparity in DCT coefficients, is prone to exhibit a significant bias towards high-energy frequency components. This bias ultimately results in limited representational capacity and reduced effectiveness in capturing the full spectrum of frequency information.

## C.4 ADVERSARIAL ROBUSTNESS AND LOSS LANDSCAPE

### C.4.1 ADVERSARIAL ROBUSTNESS MEASURE

To rigorously establish the relationship between a smooth (flat) loss landscape and higher adversarial robustness, we begin by defining the adversarial robustness measure in terms of the "Expected Distortion Rate (EDR)" of the loss function with respect to input perturbations. By connecting this metric to the gradient norm of the loss and demonstrating how smoother loss landscapes yield smaller gradient norms, we can show that a flatter landscape reduces such distortion. This ultimately supports the conclusion that smoother loss landscapes enhance adversarial robustness by limiting variations in loss under input perturbations.

To measure adversarial robustness, we define the *Expected Distortion Rate (EDR)*. This measure captures the sensitivity of the model's loss function to adversarial perturbations in the input space. The mathematical definition is:

$$\text{EDR}_\theta = \mathbb{E}_{x \sim \mathcal{X}} \left[ |L(\theta, x + \delta_x, y) - L(\theta, x, y)| \right],$$

where:

- $\mathcal{X}$ is a compact subset of $\mathbb{R}^n$, representing the input space.
- $\theta$ denotes the model parameters.

- $L(\cdot)$ is the loss function of the model.
- $y$ is the ground truth label corresponding to the input $x$.
- The perturbation from different kinds of adversarial attacks can be measured in $L_1$, $L_2$, or $L_\infty$ norms. We specifically use the $L_2$ norm, where $\|\delta_x\|_2 \leq \epsilon$  for $\epsilon > 0$.

A smaller EDR$_\theta$ indicates better robustness, as it implies that adversarial attacks induce minimal changes in the loss.

### C.4.2  How to Describe the Smoothness of the Loss Landscape

Based on the loss landscape visualization depicted in Figure 2a, 2b and 2c, we observe that the geometric characteristics of the loss landscape are predominantly determined by two fundamental components. The smoothness of the loss landscape is a critical factor in determining the robustness of a model. This section explains how to characterize smoothness using first-order (gradient) and second-order (Hessian) properties.

The local behavior of the loss function $L(\theta, x, y)$ around a point $x_0$ can be described as follows:

1. The gradient $\nabla_x L(\theta, x_0, y_0)$ determines the direction and rate of steepest ascent in the loss surface.
2. The Hessian $\nabla_x^2 L(\theta, x_0, y_0)$ quantifies the curvature of the loss surface, describing how the gradient changes in different directions.

To achieve a smoother loss landscape, it is important to minimize both the magnitude of the gradient and the spectral norm of the Hessian matrix. Prior research has proposed diverse metrics to characterize the smoothness of loss landscapes and their correlation with model generalization, including Volume $\varepsilon$-Flatness (Hochreiter & Schmidhuber, 1997), Hessian-based measures (Dinh et al., 2017) and gradient-based analysis (Zhang et al., 2023). Our work adopts a more comprehensive approach by jointly analyzing both gradient and curvature characteristics across extended regions of the loss surface. This broader perspective is particularly vital for understanding adversarial robustness, as adversarial perturbations can push model predictions far from local minima, where the geometric properties of non-minimal regions become crucial determinants of model behavior.

To formally quantify smoothness, we use the concept of *Lipschitz smoothness*. A function $f : \mathbb{R}^n \to \mathbb{R}$ is $\beta$-smooth if:
$$\|\nabla f(x) - \nabla f(y)\|_2 \leq \beta \|x - y\|_2, \quad \forall x, y \in \mathbb{R}^n,$$
where $\beta$ is the Lipschitz constant of the gradient. For the loss function $L(\theta, x, y)$, this implies that the spectral norm of the Hessian is bounded as:
$$\|\nabla_x^2 L(\theta, x_0, y_0)\|_2 \leq \beta.$$

### C.4.3  Relating the EDR to the Gradient and Hessian

We now connect the smoothness of the loss landscape to the adversarial robustness measure (EDR). The analysis is divided into two cases based on the magnitude of the perturbation $\delta_x$.

When $\delta_x$ is small, we can approximate the loss function using the second-order Taylor expansion:

$$L(\theta, x + \delta_x, y) = L(\theta, x, y) + \nabla_x L(\theta, x, y)^\top \delta_x + \frac{1}{2} \delta_x^\top \nabla_x^2 L(\theta, x + \xi\delta_x, y)\delta_x + O(\|\delta_x\|^3),$$

where $\xi \in [0, 1]$, $L(\theta, x, y)$ is the original loss value, representing the base value before perturbation. The term $\nabla_x L(\theta, x, y)^\top \delta_x$ is the first-order approximation, which is the inner product of the gradient and the perturbation. The term $\frac{1}{2} \delta_x^\top \nabla_x^2 L(\theta, x + \xi\delta_x, y)\delta_x$ is the second-order approximation, capturing the local curvature of the loss landscape. Finally, $O(\|\delta_x\|^3)$ contains all terms of order 3 and higher.

$$|L(\theta, x + \delta_x, y) - L(\theta, x, y)| \leq \|\nabla_x L(\theta, x, y)\|_q \|\delta_x\|_p + \frac{\beta}{2}\|\delta_x\|_2^2.$$

Taking the expectation over the data distribution $x \sim \mathcal{X}$, the EDR can be bounded as:

$$\text{EDR}_\theta \leq \mathbb{E}_{x \sim \mathcal{X}} \left[ \|\nabla_x L(\theta, x, y)\|_q \|\delta_x\|_p + \frac{\beta}{2}\|\delta_x\|_2^2 \right].$$

where $\|\cdot\|_p$ and $\|\cdot\|_q$ are dual norms satisfying $\frac{1}{p} + \frac{1}{q} = 1$. In this proof, we choose $p = q = 2$, which makes the analysis of the gradient and Hessian easier (from the Cauchy-Schwarz inequality).

This bound characterizes how the expected distortion depends on both the average gradient magnitude and the curvature of the loss landscape across the data distribution.

For larger perturbations where higher-order terms are non-negligible, we use integral approximation:

$$L(\theta, x + \delta_x, y) \approx L(\theta, x, y) + \int_x^{x+\delta_x} \nabla_x L(\theta, z, y) dz.$$

Applying the Mean Value Inequality for vector-valued functions:

$$|L(\theta, x + \delta_x, y) - L(\theta, x, y)| \leq \sup_{z \in \text{conv}(x, x+\delta_x)} \|\nabla_x L(\theta, z, y)\|_q \|\delta_x\|_p,$$

where $\text{conv}(x, x + \delta_x)$ represents the convex hull between $x$ and $x + \delta_x$. Thus, the EDR can be bounded as:

$$\text{EDR}_\theta \leq \mathbb{E}_{x \sim \mathcal{X}} \left[ \sup_{z \in B(x, \epsilon)} \|\nabla_x L(\theta, z, y)\|_q \|\delta_x\|_p \right].$$

### C.4.4 FINAL BOUND

From both Taylor expansion and integral approximation analyses, we can establish the relationship between loss landscape smoothness and model adversarial robustness:

The Expected Distortion Rate (EDR) can be bounded as:

$$\text{EDR}_\theta \leq \mathbb{E}_{x \sim \mathcal{X}} \left[ \|\nabla_x L(\theta, x, y)\|_q \|\delta_x\|_p + \frac{\beta}{2} \|\delta_x\|_2^2 \right],$$

where the first-order effect is controlled by the gradient magnitude, and the second-order effect is governed by the Hessian bound $\beta$. Smaller gradients and a smaller $\beta$ result in tighter bounds on loss changes.

We can define:

$$T = \mathbb{E}_{x \sim \mathcal{X}} \left[ \|\nabla_x L(\theta, x, y)\|_q \|\delta_x\|_p + \frac{\beta}{2} \|\delta_x\|_2^2 \right].$$

Alternatively, the EDR can also be bounded as:

$$\text{EDR}_\theta \leq \mathbb{E}_{x \sim \mathcal{X}} \left[ \sup_{z \in B(x, \epsilon)} \|\nabla_x L(\theta, z, y)\|_q \|\delta_x\|_p \right],$$

which we can define as:

$$I = \mathbb{E}_{x \sim \mathcal{X}} \left[ \sup_{z \in B(x, \epsilon)} \|\nabla_x L(\theta, z, y)\|_q \|\delta_x\|_p \right].$$

This indicates that loss changes are controlled by the gradient magnitude, with smoother regions (characterized by smaller gradients) leading to smaller distortions. Robustness is inherently dependent on the smoothness properties of the function.

Finally, given a perturbation magnitude $\epsilon = \|\delta_x\|$ and a threshold $\epsilon_0$, we can establish a comprehensive bound:

$$\text{EDR}_\theta \leq \begin{cases} \min\{T, I\}, & \text{if } \epsilon \leq \epsilon_0, \\ I, & \text{if } \epsilon > \epsilon_0. \end{cases}$$

Here, $\epsilon_0$ marks the critical threshold where Taylor expansion remains valid. For small perturbations ($\epsilon \leq \epsilon_0$), both bounds hold, and we can leverage the tighter one. For large perturbations ($\epsilon > \epsilon_0$), only the integral approximation bound remains valid.

## D  MEMORY AND TIME LOSS

Our learnable frequency pruning algorithm offers the additional advantage of reducing dataset storage costs. By pruning certain frequency components while preserving others, storing the coreset in the frequency domain significantly lowers storage requirements, as many frequency components are removed. If we want to use this method, we will need an extra time cost because we need to transform the image from the frequency domain to spatial domain. Now we will discuss the time cost of IDCT processing.

### D.1  COMPUTATIONAL COMPLEXITY

The computational complexity of 2D-DCT is $O(N^2 \log N)$ and the computational complexity of 2D-IDCT is also $O(N^2 \log N)$, N stands for the width and height of the image.

### D.2  COMPUTATIONAL COST

In this section we will show the time cost detail of our algorithm, Table 4(a) shows the time cost of Using DCT on CIFAR-10, CIFAR-100, and Imagenet-1K. We can find that the time cost of GPU is far less than using CPU, and the time cost of CPU is also really small which shows that on some edge devices which do not have GPU to run deep learning algorithms, we can run DCT using CPU. The reason why the imagenet-1K time cost is far higher than CIFAR-10 and CIFAR-100 is that the imagenet-1K trainset is really large containing 1281167 images and every image size of $224 \times 224$. The storage of imagenet-1k is about 138GB which is hard to deploy on a single-edge device, so in practical using we always choose to separate this dataset into many different subsets and deploy them on different devices. In this project, we use the GPU model NVIDIA A100-SXM4-40GB

Table 4(b) highlights the time cost comparison for applying IDCT on various datasets using CPU and GPU implementations. The data clearly demonstrates that our algorithm achieves remarkable efficiency in performing IDCT, with minimal time consumption across all tested datasets. This observation underscores the computational feasibility of our approach, as the IDCT step does not introduce significant overhead to the overall pruning process.

The frequency pruning can reduce the storage of the dataset. By applying DCT and leveraging sparsity in the frequency domain, we store only significant non-zero coefficients using an optimized sparse storage format. Each non-zero coefficient requires 6 bytes: 4 bytes for the float32 value and 2 bytes for packed indices. Since the image size is 32×32, we can efficiently encode both row and column indices using 5 bits each, combining them into a single 16-bit integer.

|  | Time of using CPU | Time of using GPU |
|---|---|---|
| CIFAR-10 | 15.57s | 3.43s |
| CIFAR-100 | 15.43s | 3.49s |
| Imagenet-1k | 5381.93s | 548.06s |

(a) DCT Time Comparison

|  | Time of using CPU | Time of using GPU |
|---|---|---|
| CIFAR-10 | 15.13s | 0.95s |
| CIFAR-100 | 15.09s | 1.55s |
| Imagenet-1k | 673.38s | 164.74s |

(b) IDCT Time Comparison

Table 4: Time comparison for different datasets using DCT and IDCT on CPU and GPU.

In the practical experiments, we only care about the time cost of IDCT if we store the frequency components, we need to use it to process the dataset. By applying DCT and leveraging sparsity in the frequency domain, we store only significant non-zero coefficients using an optimized sparse storage format. Each non-zero coefficient requires 6 bytes: 4 bytes for the float32 value and 2 bytes for packed indices. Since the image size is 32×32, we can efficiently encode both row and column indices using 5 bits each, combining them into a single 16-bit integer. The results of the practical storage compression ratio on CIFAR-10 are shown in Table 5.

| Pruning Ratio | Elements per Image | Storage per Image (bytes) | Total Storage | Percentage of Original |
|---|---|---|---|---|
| 50% | 1,536 | 9,216 | 220 MB | 75% |
| 70% | 922 | 5,532 | 132 MB | 45% |
| 80% | 614 | 3,684 | 88 MB | 30% |
| 90% | 307 | 1,842 | 44 MB | 15% |

Table 5: Storage requirements for different pruning ratios on CIFAR-10.

# E  DETAILED STRUCTURE OF FSE-NET

---

**Model Structure:**

---

SEBlock(
    (avg_pool): AdaptiveAvgPool2d(output_size=1)
    (fc): Sequential(
        (0): Linear(in_features=N, out_features=N/16, bias=False)
        (1): ReLU(inplace=True)
        (2): Linear(in_features=N/16, out_features=N, bias=False)
        (3): Sigmoid()
    )
)

---

Figure 7: Detailed architecture of FSE-Net, where N represents the input channel dimension.

Figure 7 presents the architectural overview of our proposed FSE-Net. The network incorporates a feature selective enhancement mechanism that adaptively models channel-wise feature interdependencies. The selective attention mechanism helps the network focus on the most discriminative features, thereby improving the overall performance of the model.

# F  TESTING ON DIFFERENT ADVERSARIAL ATTACKS

Tables 6 and 7 provide a comprehensive comparison of our algorithm's performance against various types of adversarial attacks, specifically $l_2$-AA (adversarial attacks constrained in the $l_2$ norm), $l_0$-AA (attacks constrained in the $l_0$ norm, targeting sparsity), and s-AA (structured adversarial attacks). These tables highlight the robustness of our approach by demonstrating superior accuracy and resilience under these diverse adversarial scenarios. Table 8 demonstrates the effectiveness of our algorithm against adaptive attack.

| | $l_2$-**AA** | | | | | $l_1$-**AA** | | | | |
|---|---|---|---|---|---|---|---|---|---|---|
| $p$ | **50%** | **60%** | **70%** | **80%** | **90%** | **50%** | **60%** | **70%** | **80%** | **90%** |
| **Random** | 18.87 ±0.12 | 23.51 ±0.32 | 23.15 ±0.27 | 21.39 ±0.20 | 21.58 ±0.29 | 15.87 ±0.62 | 20.51 ±0.75 | 20.15 ±0.68 | 18.39 ±0.55 | 18.58 ±0.83 |
| **Entropy** | 24.65 ±0.45 | 24.27 ±0.31 | 33.92 ±0.18 | 26.28 ±0.41 | 23.74 ±0.25 | 21.65 ±0.78 | 21.27 ±0.67 | 30.92 ±0.85 | 23.28 ±0.72 | 20.74 ±0.63 |
| **CSFEM** | 40.86 ±0.34 | 44.98 ±0.38 | 44.02 ±0.21 | 43.18 ±0.44 | 44.28 ±0.26 | 37.86 ±0.83 | 40.98 ±0.58 | 41.02 ±0.79 | 40.18 ±0.87 | 41.28 ±0.68 |
| **ours-JE** | 43.16 ±0.19 | 47.32 ±0.29 | 46.97 ±0.14 | 44.38 ±0.27 | 43.65 ±0.43 | 40.16 ±0.72 | 44.32 ±0.84 | 42.97 ±0.61 | 41.38 ±0.76 | 40.65 ±0.89 |
| **ours-LF** | 58.70 ±0.23 | 62.19 ±0.46 | 56.46 ±0.36 | 55.14 ±0.31 | 54.04 ±0.20 | 55.70 ±0.86 | 58.19 ±0.65 | 53.46 ±0.77 | 51.14 ±0.59 | 51.04 ±0.81 |
| **ours-JELF** | 49.83 ±0.22 | 50.32 ±0.41 | 51.19 ±0.25 | 52.72 ±0.37 | 51.21 ±0.24 | 46.83 ±0.69 | 47.32 ±0.88 | 48.19 ±0.66 | 49.72 ±0.74 | 48.21 ±0.57 |

Table 6: Performance of CIFAR-10 dataset on ResNet-18 under $l_2$-AA (Croce & Hein, 2020) and $l_1$-AA (Croce & Hein, 2021) attacks. We run every experiment five times and report their mean and standard deviation.

| Method | 90% | 80% | 70% | 60% | 50% |
|---|---|---|---|---|---|
| **Random** | 7.87±0.62 | 9.51±0.75 | 9.15±0.68 | 7.39±0.55 | 7.58±0.83 |
| **Entropy** | 10.65±0.78 | 10.27±0.67 | 19.92±0.85 | 12.28±0.72 | 9.74±0.63 |
| **CSFEM** | 26.86±0.83 | 29.98±0.58 | 30.02±0.79 | 29.18±0.87 | 30.28±0.68 |
| **ours-JE** | 29.16±0.72 | 33.32±0.84 | 31.97±0.61 | 30.38±0.76 | 29.65±0.89 |
| **ours-LF** | 44.70±0.86 | 45.19±0.65 | 42.46±0.77 | 40.14±0.59 | 40.04±0.81 |
| **ours-JELF** | 35.83±0.69 | 36.32±0.88 | 37.19±0.66 | 38.72±0.74 | 37.21±0.57 |

Table 7: Performance of CIFAR-10 dataset on ResNet-18 under s-AA (Zhong et al., 2024), we run every experiment five times and get their mean and standard deviation.

| Method | 90% | 80% | 70% | 60% | 50% |
|---|---|---|---|---|---|
| **Random** | 13.97±0.62 | 18.61±0.75 | 18.25±0.68 | 16.49±0.55 | 16.68±0.83 |
| **Entropy** | 19.85±0.78 | 19.47±0.67 | 29.12±0.85 | 21.48±0.72 | 18.94±0.63 |
| **CSFEM** | 35.96±0.83 | 39.08±0.58 | 39.12±0.79 | 38.28±0.87 | 35.38±0.68 |
| **ours-JE** | 38.36±0.72 | 42.52±0.84 | 41.17±0.61 | 39.58±0.76 | 38.85±0.89 |
| **ours-LF** | 55.50±0.86 | 57.99±0.65 | 53.26±0.77 | 50.94±0.59 | 50.84±0.81 |
| **ours-JELF** | 46.63±0.69 | 47.12±0.88 | 47.99±0.66 | 49.52±0.74 | 48.01±0.57 |

Table 8: Performance of CIFAR-10 dataset on ResNet-18 under $l_1$-APGD (Croce & Hein, 2021), we run every experiment five times and get their mean and standard deviation.

# G    DETAILED EXPERIMENT RESULT OF OUR BASELINE EXPERIMENTS

Table 9 presents comprehensive experimental results, where each configuration was repeated five times to ensure statistical reliability. We report both the mean accuracy and standard deviation (shown as ±) to demonstrate the consistency and robustness of our method across multiple runs. Tables 11a and 11b present ablation studies on two key hyperparameters: learning rate and number of iterations, which guided our selection of optimal values for the proposed method. Table 10a, Table 10b and Table 10c demonstrate the transferability of our method across different lightweight architectures (ShuffleNet, MobileNet-v2, and EfficientNet-B0). Our LF-based approach maintains consistent superior robustness on both networks against various attacks, showing strong generalization capability across different model architectures compared to baseline methods.

| Pruning Algorithm | Attack | Prune Rate | | | | | | | | | |
|---|---|---|---|---|---|---|---|---|---|---|---|
| | | CIFAR-10 | | | | | CIFAR-100 | | | | |
| | | 90% | 80% | 70% | 60% | 50% | 90% | 80% | 70% | 60% | 50% |
| Random | AA | 15.27±0.63 | 21.55±0.75 | 20.85±0.58 | 22.33±0.67 | 21.53±0.72 | 12.27±0.55 | 16.55±0.68 | 16.88±0.71 | 17.03±0.64 | 17.57±0.59 |
| | PGD-20 | 16.27±0.65 | 20.55±0.73 | 19.85±0.62 | 21.33±0.69 | 20.53±0.77 | 11.97±0.58 | 15.59±0.66 | 16.78±0.74 | 17.23±0.61 | 16.57±0.57 |
| | C&W | 16.39±0.68 | 20.45±0.71 | 18.83±0.64 | 20.53±0.76 | 20.77±0.79 | 12.97±0.54 | 14.59±0.69 | 15.78±0.72 | 16.23±0.63 | 14.57±0.56 |
| Entropy | AA | 21.65±0.67 | 21.27±0.74 | 30.92±0.59 | 23.28±0.65 | 20.74±0.73 | 11.56±0.57 | 14.33±0.70 | 17.98±0.75 | 15.33±0.62 | 17.71±0.58 |
| | PGD-20 | 20.68±0.64 | 20.87±0.72 | 20.92±0.61 | 21.28±0.68 | 22.74±0.76 | 12.16±0.56 | 14.03±0.67 | 16.18±0.73 | 15.13±0.65 | 17.01±0.60 |
| | C&W | 20.44±0.66 | 20.78±0.70 | 20.21±0.63 | 21.17±0.75 | 22.83±0.78 | 12.44±0.53 | 12.35±0.71 | 17.18±0.74 | 16.37±0.64 | 17.51±0.55 |
| CCSFEM | AA | 37.86±0.69 | 40.98±0.73 | 41.02±0.60 | 40.18±0.66 | 41.28±0.74 | 15.11±0.59 | 16.85±0.72 | 18.05±0.76 | 18.19±0.63 | 17.92±0.57 |
| | PGD-20 | 38.97±0.63 | 40.11±0.71 | 39.99±0.62 | 39.76±0.67 | 41.91±0.75 | 13.98±0.55 | 15.92±0.68 | 13.09±0.72 | 14.08±0.66 | 17.91±0.61 |
| | C&W | 38.99±0.65 | 40.33±0.69 | 40.02±0.64 | 39.96±0.74 | 42.05±0.77 | 12.17±0.52 | 16.15±0.73 | 17.91±0.75 | 18.60±0.65 | 17.27±0.54 |
| Ours-JE | AA | 40.16±0.68 | 44.32±0.72 | 42.97±0.61 | 41.38±0.65 | 40.65±0.73 | 16.15±0.58 | 17.12±0.71 | 21.37±0.77 | 20.09±0.64 | 18.65±0.56 |
| | PGD-20 | 39.16±0.62 | 39.32±0.70 | 41.07±0.63 | 40.88±0.66 | 42.95±0.74 | 14.76±0.54 | 16.88±0.69 | 17.01±0.71 | 16.89±0.67 | 18.05±0.62 |
| | C&W | 39.06±0.64 | 39.72±0.68 | 41.37±0.65 | 40.96±0.73 | 43.05±0.76 | 12.99±0.51 | 17.32±0.72 | 18.07±0.76 | 19.88±0.66 | 18.95±0.53 |
| Ours-LF | AA | **55.7±0.67** | **58.19±0.71** | **53.46±0.62** | **51.14±0.64** | **51.04±0.72** | 20.99±0.57 | **24.94±0.70** | **25.07±0.78** | **25.31±0.65** | 23.41±0.55 |
| | PGD-20 | **56.18±0.61** | **56.05±0.69** | **51.9±0.64** | **50.61±0.65** | **50.07±0.73** | **23.72±0.53** | **21.64±0.70** | **21.36±0.70** | **24.39±0.68** | **23.75±0.63** |
| | C&W | **56.42±0.63** | **56.21±0.67** | **54.14±0.66** | **54.38±0.72** | **55.32±0.75** | **22.37±0.50** | **24.23±0.71** | **25.48±0.77** | **26.39±0.67** | **28.69±0.52** |
| Ours-JELF | AA | **46.54±0.66** | **47.35±0.70** | **48.89±0.63** | **49.72±0.63** | **48.18±0.71** | **20.39±0.56** | **20.53±0.69** | **22.47±0.79** | **22.94±0.66** | **22.85±0.54** |
| | PGD-20 | **47.24±0.60** | **48.59±0.68** | **49.64±0.65** | **50.25±0.64** | **50.01±0.72** | **18.09±0.52** | **19.71±0.71** | **20.98±0.69** | **21.85±0.69** | **22.48±0.64** |
| | C&W | **47.61±0.62** | **48.12±0.66** | **49.01±0.67** | **48.19±0.71** | **48.66±0.74** | **18.99±0.49** | **20.01±0.72** | **21.54±0.78** | **22.03±0.68** | **21.99±0.51** |

Table 9: We assess CIFAR-10 and CIFAR-100 performance under various adversarial attacks and dataset pruning ratios. CCSFEM" uses forgetting, EL2N, and AUM scores with CCS to compute the mean accuracy. Ours-JE" applies the joint-entropy score with CCS sampling, Ours-LF" uses Learnable Frequency Pruning on the total dataset, and Ours-JELF" combines Learnable Frequency Pruning (preserving 50% of frequency components) with joint-entropy based coreset selection using CCS sampling, we run every experiment five times and get their mean and standard deviation.

| Pruning Algorithm | Attack | Pruning Rate (%) | | | | |
|---|---|---|---|---|---|---|
| | | 90 | 80 | 70 | 60 | 50 |
| Random | AA | 11.27±0.45 | 17.55±0.33 | 16.85±0.42 | 18.33±0.36 | 17.53±0.41 |
| | PGD-20 | 12.27±0.44 | 16.55±0.35 | 15.85±0.38 | 17.33±0.43 | 16.53±0.42 |
| | C&W | 12.39±0.36 | 16.45±0.47 | 14.83±0.35 | 16.53±0.39 | 16.77±0.38 |
| Entropy | AA | 17.65±0.41 | 17.27±0.37 | 26.92±0.49 | 19.28±0.38 | 16.74±0.45 |
| | PGD-20 | 16.68±0.43 | 16.87±0.39 | 16.92±0.42 | 17.28±0.51 | 18.74±0.37 |
| | C&W | 16.44±0.38 | 16.78±0.52 | 16.21±0.35 | 17.17±0.46 | 18.83±0.44 |
| CCSFEM | AA | 33.86±0.43 | 36.98±0.38 | 37.02±0.50 | 36.18±0.39 | 37.28±0.47 |
| | PGD-20 | 34.97±0.34 | 36.11±0.55 | 35.99±0.43 | 35.76±0.44 | 37.91±0.36 |
| | C&W | 34.99±0.51 | 36.33±0.44 | 36.02±0.39 | 35.96±0.53 | 38.05±0.37 |
| Ours-JE | AA | 36.16±0.44 | 40.32±0.35 | 38.97±0.56 | 37.38±0.42 | 36.65±0.39 |
| | PGD-20 | 35.16±0.37 | 35.32±0.54 | 37.07±0.41 | 36.88±0.34 | 38.95±0.50 |
| | C&W | 35.06±0.55 | 35.72±0.43 | 37.37±0.38 | 36.96±0.47 | 39.05±0.39 |
| Ours-LF | AA | 51.70±0.39 | 54.19±0.53 | 49.46±0.44 | 47.14±0.36 | 47.04±0.54 |
| | PGD-20 | 52.18±0.52 | 52.05±0.41 | 47.90±0.45 | 46.61±0.56 | 46.07±0.35 |
| | C&W | 52.42±0.45 | 52.21±0.50 | 50.14±0.33 | 50.38±0.43 | 51.32±0.55 |
| Ours-JELF | AA | 42.54±0.49 | 43.35±0.36 | 44.89±0.54 | 45.72±0.37 | 44.18±0.47 |
| | PGD-20 | 43.24±0.38 | 44.59±0.51 | 45.64±0.42 | 46.25±0.55 | 46.01±0.34 |
| | C&W | 43.61±0.46 | 44.12±0.36 | 45.01±0.53 | 44.19±0.43 | 44.66±0.56 |

(a) ShuffleNet

| Pruning Algorithm | Attack | Pruning Rate (%) | | | | |
|---|---|---|---|---|---|---|
| | | 90 | 80 | 70 | 60 | 50 |
| Random | AA | 13.77±0.41 | 19.85±0.36 | 19.15±0.42 | 20.83±0.37 | 19.93±0.44 |
| | PGD-20 | 14.77±0.45 | 18.85±0.34 | 18.15±0.38 | 19.83±0.43 | 18.93±0.43 |
| | C&W | 14.89±0.35 | 18.75±0.48 | 17.13±0.37 | 18.93±0.38 | 19.17±0.37 |
| Entropy | AA | 20.15±0.42 | 19.77±0.37 | 29.42±0.50 | 21.78±0.40 | 19.24±0.46 |
| | PGD-20 | 19.18±0.44 | 19.37±0.38 | 19.42±0.43 | 19.78±0.52 | 21.24±0.36 |
| | C&W | 18.94±0.40 | 19.28±0.53 | 18.71±0.34 | 19.67±0.47 | 21.33±0.43 |
| CCSFEM | AA | 36.36±0.44 | 39.48±0.37 | 39.52±0.51 | 38.68±0.38 | 39.78±0.48 |
| | PGD-20 | 37.47±0.33 | 38.61±0.56 | 38.49±0.42 | 38.26±0.45 | 40.41±0.35 |
| | C&W | 37.49±0.52 | 38.83±0.43 | 38.52±0.38 | 38.46±0.54 | 40.55±0.38 |
| Ours-JE | AA | 38.66±0.45 | 42.82±0.34 | 41.47±0.57 | 39.88±0.43 | 39.15±0.38 |
| | PGD-20 | 37.66±0.36 | 37.82±0.55 | 39.57±0.42 | 39.38±0.33 | 41.45±0.51 |
| | C&W | 37.56±0.56 | 38.22±0.42 | 39.87±0.37 | 39.46±0.48 | 41.55±0.38 |
| Ours-LF | AA | 54.20±0.38 | 56.69±0.54 | 51.96±0.43 | 49.64±0.35 | 49.54±0.55 |
| | PGD-20 | 54.68±0.53 | 54.55±0.40 | 50.40±0.46 | 49.11±0.57 | 48.57±0.34 |
| | C&W | 54.92±0.44 | 54.71±0.51 | 52.64±0.32 | 52.88±0.44 | 53.82±0.56 |
| Ours-JELF | AA | 45.04±0.50 | 45.85±0.37 | 47.39±0.55 | 48.22±0.36 | 46.68±0.48 |
| | PGD-20 | 45.74±0.37 | 47.09±0.52 | 48.14±0.43 | 48.75±0.56 | 48.51±0.33 |
| | C&W | 46.11±0.47 | 46.62±0.35 | 47.51±0.54 | 46.69±0.42 | 47.16±0.57 |

(b) MobileNet-v2

| Pruning Algorithm | Attack | Pruning Rate (%) | | | | |
|---|---|---|---|---|---|---|
| | | 90 | 80 | 70 | 60 | 50 |
| Random | AA | 14.89±0.43 | 21.23±0.35 | 20.45±0.44 | 21.98±0.38 | 21.03±0.42 |
| | PGD-20 | 15.77±0.46 | 20.05±0.37 | 19.35±0.40 | 20.83±0.45 | 19.98±0.44 |
| | C&W | 15.89±0.38 | 19.95±0.45 | 17.93±0.37 | 19.98±0.41 | 20.27±0.40 |
| Entropy | AA | 21.15±0.42 | 20.77±0.39 | 30.42±0.47 | 22.78±0.40 | 20.24±0.43 |
| | PGD-20 | 20.18±0.45 | 20.37±0.41 | 20.42±0.44 | 20.78±0.49 | 22.24±0.39 |
| | C&W | 19.94±0.40 | 20.28±0.50 | 19.71±0.37 | 20.67±0.44 | 22.33±0.42 |
| CCSFEM | AA | 37.36±0.45 | 40.48±0.40 | 40.52±0.48 | 39.68±0.41 | 40.78±0.45 |
| | PGD-20 | 38.47±0.36 | 39.05±0.53 | 39.49±0.45 | 39.26±0.46 | 41.41±0.38 |
| | C&W | 38.49±0.49 | 39.83±0.46 | 39.52±0.41 | 39.46±0.51 | 41.55±0.39 |
| Ours-JE | AA | 39.66±0.46 | 43.82±0.37 | 42.47±0.54 | 40.88±0.44 | 40.15±0.41 |
| | PGD-20 | 38.66±0.39 | 38.82±0.52 | 41.07±0.43 | 40.38±0.36 | 42.45±0.48 |
| | C&W | 38.56±0.53 | 39.22±0.45 | 40.87±0.40 | 40.46±0.45 | 42.55±0.41 |
| Ours-LF | AA | 55.20±0.41 | 57.69±0.51 | 52.96±0.46 | 50.64±0.38 | 50.54±0.52 |
| | PGD-20 | 55.68±0.50 | 55.55±0.43 | 51.40±0.47 | 50.11±0.54 | 49.57±0.37 |
| | C&W | 55.92±0.47 | 55.71±0.48 | 53.64±0.35 | 53.88±0.45 | 54.82±0.53 |
| Ours-JELF | AA | 46.04±0.47 | 46.89±0.38 | 48.39±0.52 | 49.22±0.39 | 47.68±0.45 |
| | PGD-20 | 46.74±0.40 | 48.09±0.49 | 49.14±0.44 | 49.75±0.53 | 49.51±0.36 |
| | C&W | 47.11±0.44 | 47.62±0.38 | 48.51±0.51 | 47.69±0.51 | 48.16±0.43 |

(c) EfficientNet-B0

Table 10: Performance comparison of different pruning algorithms under various attacks on CIFAR-10 with different network architectures. Each experiment is repeated five times to obtain the mean and standard deviation.

| Learning Rate | 0.1 | 0.05 | 0.01 | 0.005 | 0.001 | 0.0001 |
|---|---|---|---|---|---|---|
| **Accuracy (%)** | 48.18 | 47.11 | 47.09 | 46.88 | 46.75 | 47.08 |

(a) Learning Rate vs. Accuracy

| Iterations | 40000 | 50000 | 30000 | 45000 | 15000 | 20000 |
|---|---|---|---|---|---|---|
| **Accuracy (%)** | 48.18 | 47.02 | 46.99 | 47.18 | 46.15 | 47.11 |

(b) Iterations vs. Accuracy

Table 11: Ablation study on hyper-parameters when using "Ours-JELF" under Autoattack and pruning ratio 50% on CIFAR-10.

# H VISUALLIZATION OF OUR CORESET SELECTION

As illustrated in Figure 8, two key observations emerge: 1) The visual fidelity remains remarkably preserved even after 50% frequency component pruning, and their differences are hard to check in visualization, demonstrating the effectiveness of our frequency pruning strategy in maintaining essential image characteristics. 2) Images selected for the coreset exhibit notably distinct structural features compared to their non-selected counterparts.

Figure 9 illustrates that while the exact loss landscapes vary across 5 independent runs, the bold lines are averages, light-colored lines are for other cases, the relative smoothness characteristics between different methods remain consistent, validating the reliability of our comparative analysis.

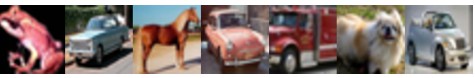

(a) Original images selected into coreset when pruning ratio = 90%.

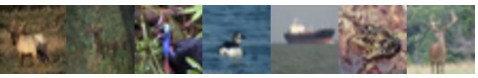

(b) Original images not be selected into coreset when pruning ratio = 90%.

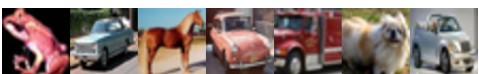

(c) Images with 50% frequency pruning ratio selected into coreset when pruning ratio = 90%.

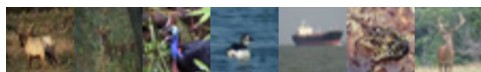

(d) Images with 50% frequency pruning ratio not be selected into coreset when pruning ratio = 90%.

Figure 8: Visuallization of CIFAR-10 trainset.

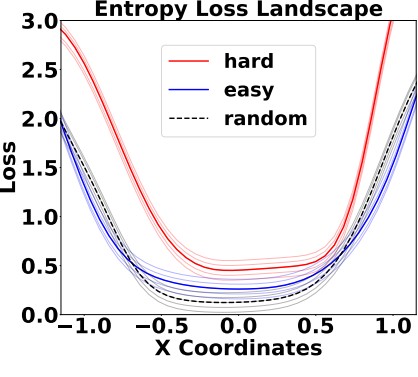

(a) Entropy Loss Landscape.

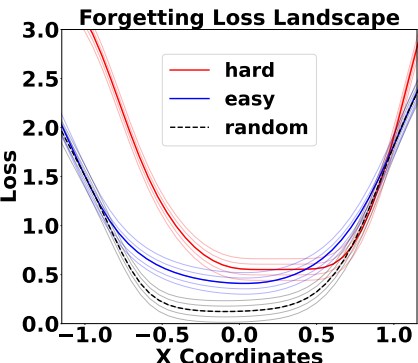

(b) Forgetting Loss Landscape.

Figure 9: Plot the line with 5 runs, the lighter color indicates the result in five parts, and the darker line indicates the average value.

# I  ADDITIONAL EXPERIMENT RESULTS ON ADVERSARIAL ROBUSTNESS

Table 12a, Table 12b and Table 12c shows more results when the model was attacked by adversarial samples, we can find that our algorithms have a better performance on improve the adversarial robustness of the model compare with traditional dataset pruning algorithms.

| Pruning Algorithms (Attack) | Pruning Ratio | | | | | | |
|---|---|---|---|---|---|---|---|
| | 95% | 90% | 80% | 70% | 60% | 50% | 30% |
| Random (AA) | 11.59 | 12.27 | 16.55 | 16.88 | 17.03 | 17.57 | 16.66 |
| Entropy (AA) | 10.50 | 11.56 | 14.33 | 17.98 | 15.33 | 17.71 | 17.81 |
| CCSFEM (AA) | 11.01 | 15.11 | 16.85 | 18.05 | 18.19 | 17.92 | 16.17 |
| ours-JE (AA) | 12.92 | 16.15 | 17.12 | 21.37 | 20.09 | 18.65 | 17.99 |
| ours-LF (AA) | **13.37** | **20.99** | **24.94** | **25.07** | **25.31** | **23.41** | **23.89** |
| ours-JELF (AA) | **14.10** | **20.39** | **20.53** | **22.47** | **22.94** | **22.85** | **22.15** |

(a) CIFAR-100 results.

| Pruning Algorithms (Attack) | Pruning Ratio | | | | | | |
|---|---|---|---|---|---|---|---|
| | 95% | 90% | 80% | 70% | 60% | 50% | 30% |
| Random (AA) | 18.59 | 15.27 | 21.55 | 20.85 | 22.33 | 21.53 | 19.66 |
| Entropy (AA) | 18.50 | 21.65 | 21.27 | 30.92 | 23.28 | 20.74 | 21.83 |
| CCSFEM (AA) | 20.82 | 37.86 | 40.98 | 41.02 | 40.18 | 41.28 | 41.12 |
| ours-JE (AA) | 22.92 | 40.16 | 44.32 | 42.97 | 41.38 | 40.65 | 41.70 |
| ours-LF (AA) | **40.70** | **55.70** | **58.19** | **53.46** | **51.14** | **51.04** | **51.37** |
| ours-JELF (AA) | **37.10** | **46.54** | **47.35** | **48.89** | **49.72** | **48.18** | **47.46** |

(b) CIFAR-10 results.

| Pruning Algorithms (Attack) | Pruning Ratio | | | | | | |
|---|---|---|---|---|---|---|---|
| | 95% | 90% | 80% | 70% | 60% | 50% | 30% |
| Random (AA) | 10.59 | 15.87 | 20.51 | 20.15 | 18.39 | 18.58 | 19.46 |
| Entropy (AA) | 11.82 | 16.87 | 18.06 | 21.02 | 18.11 | 17.28 | 20.72 |
| CCSFEM (AA) | 12.82 | 15.86 | 16.96 | 18.02 | 17.11 | 19.28 | 19.72 |
| ours-JE (AA) | 14.92 | 19.16 | 21.51 | 22.17 | 21.18 | 20.96 | 21.78 |
| ours-LF (AA) | **23.50** | **25.60** | **27.19** | **26.46** | **27.44** | **27.14** | **26.31** |
| ours-JELF (AA) | **20.10** | **23.54** | **26.15** | **25.03** | **25.22** | **24.95** | **24.89** |

(c) ImageNet-1K results.

Table 12: Comparison of pruning algorithms under different adversarial attacks for (a) CIFAR-100, (b) CIFAR-10, and (c) ImageNet-1K datasets. "Ours-JE" refers to coreset selection using the joint-entropy score with CCS sample strategy, "Ours-LF" applies Learnable Frequency Pruning, and "Ours-JELF" combines Learnable Frequency Pruning (preserving 50% of frequency components) with coreset selection using the joint-entropy score and CCS sample strategy. Each subfigure illustrates the robustness of the methods across different pruning ratios.

# J  ADDITIONAL EXPERIMENT RESULTS ON CLEAN DATASET

Table 13a and Table 13b show that our algorithms are better than SOTA dataset pruning algorithms which show that our algorithms also have a better performance on a clean dataset which shows that our dataset pruning also have potential to improve the performance of dataset pruning without adversarial attack.

| Pruning Rate | 30% | 50% | 70% | 80% | 90% |
|---|---|---|---|---|---|
| Random | 94.33 | 93.4 | 90.94 | 87.98 | 79.04 |
| Entropy | 94.44 | 92.11 | 85.67 | 79.08 | 66.52 |
| Forgetting | 95.36 | 95.29 | 90.56 | 62.74 | 34.03 |
| EL2N | 95.44 | 94.61 | 87.48 | 70.32 | 22.33 |
| AUM | 95.07 | 95.26 | 91.36 | 57.84 | 28.06 |
| CCSFEM | 95.17 | 94.67 | 92.74 | 90.55 | 86.15 |
| Ours-LF | **95.35** | **94.67** | **94.33** | **93.21** | **89.82** |
| Ours-JE | **95.15** | **94.07** | **92.03** | **90.98** | **85.86** |
| Ours-JELF | **95.11** | **94.02** | **91.93** | **90.18** | **84.86** |
| Ours-FEMLF | **95.19** | **95.07** | **93.23** | **91.98** | **87.06** |

(a) Comparison on CIFAR-10 without adversarial attack. The accuracy on the whole dataset is 95.41%.

| Pruning Rate | 30% | 50% | 70% | 80% | 90% |
|---|---|---|---|---|---|
| Random | 74.59 | 71.07 | 65.3 | 57.36 | 44.76 |
| Entropy | 72.26 | 63.26 | 50.49 | 41.83 | 28.96 |
| Forgetting | 76.91 | 68.6 | 38.06 | 24.23 | 15.93 |
| EL2N | 76.25 | 65.90 | 34.42 | 15.51 | 8.36 |
| AUM | 76.93 | 67.42 | 30.64 | 16.38 | 8.77 |
| CCSFEM | 76.33 | 73.44 | 68.30 | 63.01 | 54.39 |
| Ours-LF | **78.38** | **77.24** | **74.97** | **71.85** | **61.94** |
| Ours-JE | **75.15** | **72.07** | **68.03** | **60.98** | **53.86** |
| Ours-JELF | **75.11** | **74.02** | **68.13** | **59.19** | **54.16** |
| Ours-FEMLF | **77.33** | **75.45** | **67.30** | **63.81** | **55.39** |

(b) Comparison on CIFAR-100 without adversarial attack. The accuracy on the whole dataset is 78.21%.

Table 13: Comparison of different pruning methods across various pruning rates on CIFAR-10 and CIFAR-100 without adversarial attack. "Ours-LF" applies Learnable Frequency Pruning, and "Ours-FEMLF" combines Learnable Frequency Pruning with CCSFEM coreset selection algorithm. Accuracy on the full dataset is shown for reference in each subtable.

## K ANOTHER EXPERIMENT RESULTS ON USING ADVERSARIAL TRAINING ON PRUNED DATASET

In this section, we present additional results evaluating adversarial training on pruned datasets. Table 14a and Table 14b compare different adversarial training methods under various attack scenarios. In Table 14a, we include comparisons with AWP (Wu et al., 2020) and TRADES (Zhang et al., 2019), showing that our algorithm outperforms these state-of-the-art methods in dataset pruning scenarios. Table 14c further demonstrates that our method achieves superior results on CIFAR-100, outperforming other adversarial training algorithms even on more complex datasets.

| | Original Adversarial Training | | Sample Adversarial Training | | Pre-trained Adversarial Training | |
|---|---|---|---|---|---|---|
| | 50% coreset | Original Dataset | 50% coreset | Original Dataset | 50% coreset | Original Dataset |
| **TDFAT** | Acc:46.41 Time:59.58s | Acc:56.05 Time:119.69s | Acc:26.05 Time:18.53s | Acc:39.72 Time:39.69s | Acc:36.32 Time:10.35s | Acc:41.66 Time:17.66s |
| **CURC** | Acc:45.92 Time:74.12s | Acc:54.02 Time:149.52s | Acc:30.21 Time:17.87s | Acc:35.15 Time:38.52s | Acc:39.88 Time:9.41s | Acc:42.18 Time:18.71s |
| **RATTE** | Acc:43.91 Time:64.33s | Acc:53.98 Time:131.35s | Acc:32.05 Time:17.23s | Acc:34.92 Time:38.35s | Acc:35.15 Time:10.01s | Acc:39.73 Time:17.67s |
| AWP | Acc:42.11 Time:78.12s | Acc:55.89 Time:155.59s | Acc:28.21 Time:18.89s | Acc:25.15 Time:37.31s | Acc:36.88 Time:9.61s | Acc:33.18 Time:18.55s |
| TRADES ($1/\lambda = 6$) | Acc:42.01 Time:62.13s | Acc:52.95 Time:130.05s | Acc:31.05 Time:17.18s | Acc:32.62 Time:38.27s | Acc:33.11 Time:10.05s | Acc:35.73 Time:16.67s |
| **FATSC** | Acc:14.99 Time:59.82s | Acc:34.21 Time:122.71s | Acc:22.82 Time:17.12s | Acc:25.61 Time:39.71s | Acc:15.21 Time:9.82s | Acc:19.79 Time:18.98s |
| **ours-LF** | Acc:50.01 Time:9.51s | Acc:55.32 Time:17.64s | Acc:50.01 Time:9.51s | Acc:50.07 Time:17.64s | Acc:50.01 Time:9.69s | Acc:50.07 Time:17.66s |

(a) Comparison under PGD-20 (On CIFAR-10).

| | Original Adversarial Training | | Sample Adversarial Training | | Pre-trained Adversarial Training | |
|---|---|---|---|---|---|---|
| | 50% coreset | Original Dataset | 50% coreset | Original Dataset | 50% coreset | Original Dataset |
| **TDFAT** | Acc:44.41 Time:58.68s | Acc:48.33 Time:118.59s | Acc:27.05 Time:18.53s | Acc:40.52 Time:38.69s | Acc:35.82 Time:10.35s | Acc:42.96 Time:17.59s |
| **CURC** | Acc:45.92 Time:75.02s | Acc:52.48 Time:148.62s | Acc:32.32 Time:17.87s | Acc:36.28 Time:38.52s | Acc:37.98 Time:9.71s | Acc:43.23 Time:18.85s |
| **RATTE** | Acc:45.09 Time:65.13s | Acc:52.12 Time:130.75s | Acc:34.65 Time:17.23s | Acc:36.91 Time:38.35s | Acc:38.19 Time:10.01s | Acc:40.83 Time:17.87s |
| **FATSC** | Acc:19.79 Time:58.12s | Acc:28.19 Time:121.88s | Acc:26.87 Time:17.12s | Acc:27.68 Time:39.71s | Acc:20.51 Time:9.92s | Acc:22.77 Time:17.96s |
| **ours-LF** | Acc:48.66 Time:10.51s | Acc:50.07 Time:17.14s | Acc:48.66 Time:9.51s | Acc:50.07 Time:17.64s | Acc:48.66 Time:9.49s | Acc:50.07 Time:17.87s |

(b) Comparison under C&W (On CIFAR-10).

| | Original Adversarial Training | | Sample Adversarial Training | | Pre-trained Adversarial Training | |
|---|---|---|---|---|---|---|
| | 50% coreset | Original Dataset | 50% coreset | Original Dataset | 50% coreset | Original Dataset |
| **TDFAT** | Acc:21.41 Time:59.58s | Acc:25.69 Time:120.79s | Acc:12.09 Time:18.53s | Acc:15.72 Time:39.69s | Acc:12.01 Time:10.15s | Acc:12.36 Time:17.66s |
| **CURC** | Acc:22.92 Time:74.32s | Acc:24.48 Time:151.52s | Acc:12.33 Time:17.87s | Acc:14.15 Time:38.52s | Acc:12.91 Time:9.31s | Acc:13.08 Time:17.96s |
| **RATTE** | Acc:21.97 Time:64.33s | Acc:22.59 Time:131.35s | Acc:13.05 Time:17.27s | Acc:15.15 Time:38.39s | Acc:14.33 Time:10.29s | Acc:15.03 Time:17.88s |
| **FATSC** | Acc:14.24 Time:59.89s | Acc:17.39 Time:122.72s | Acc:16.82 Time:17.18s | Acc:17.78 Time:39.77s | Acc:11.02 Time:9.35s | Acc:13.93 Time:18.77s |
| **ours-LF** | Acc:22.85 Time:9.53s | Acc:23.14 Time:16.94s | Acc:22.85 Time:9.88s | Acc:23.14 Time:17.14s | Acc:22.85 Time:9.19s | Acc:23.14 Time:18.64s |

(c) Comparison under AutoAttack (On CIFAR-100).

Table 14: We compare recent adversarial training algorithms with our Learnable Frequency Pruning method under different adversarial attacks: PGD-20, C&W, and AutoAttack. "Original Adversarial Training" applies standard AT on the entire dataset, while "Sample Adversarial Training" applies adversarial perturbations to a random subset of images each epoch, leaving the rest unchanged to match our method's training cost. Finally, " pre-trained Adversarial Training" uses a pre-trained ResNet-18 model with high adversarial robustness to generate adversarial perturbations without further optimization during training, ensuring no additional Time. We train datasets of the same size for an equal number of epochs under identical conditions.

