# OpenReview forum: "Adversarial Attack Robust dataset pruning"
_ICLR.cc/2025/Conference — Submitted to ICLR 2025_

### Official Review · Reviewer_tRah · 2024-10-30

**Soundness:** 2
**Presentation:** 2
**Contribution:** 2
**Rating:** 5
**Confidence:** 4

**Summary:**

This paper introduces a novel approach to creating adversarially robust coresets. There are two main innovations:  1) a Frequency-Selective Excitation Network that dynamically selects important frequency components and 2) a “Joint-entropy” score for selecting stable and informative samples.

**Strengths:**

1. This paper proposes a novel approach to train a robust model
2. Compared to adversarial training, this method reduces the size of the training set and does not rely on any specific adversarial attack, thereby improving the training efficiency

**Weaknesses:**

1. I am not convinced that the model trained on some carefully selected and augmented data can be robust to adversarial attacks. Unlike adversarial training, this method does not rely on any specific adversarial attack. Therefore, it is suggested to conduct a more comprehensive evaluation of your method on different attacks, such as l2-AA [1], l1-AA [2], and sAA (l0-bounded perturbations) [2], compare the results to baseline methods to further demonstrate the effectiveness of your method.

2. Line 58-59: If I understood correctly, FSE-Net is used to select important frequency components, so it is more like a data augmentation or purification method. However, it does not reduce the training samples or the size of the images. In this regard, why do you claim that FSE-Net can reduce storage requirements? You should provide a detailed explanation or quantitative analysis of how selecting frequency components translates to reduced storage needs.

3. Line 93-94:  Stutz et al. (2021) and Liu et al. (2020) establish a correlation between loss landscape flatness and adversarial robustness in the context of **adversarial training**. However, the training data are not perturbed in your method. You need a more extensive theoretical/numerical analysis of the relationship between loss landscape flatness and adversarial robustness in your context. For example, you can plot the adversarial loss landscape in both input and parameter spaces for your method (refer to Figure 4 in Stutz et al. (2021), and Figure 3 and 12 in Liu et al. (2020))

4. As illustrated in Figure 3 (b) and (c), it seems that the model is trained on the coreset (50% of the the original dataset) for 500 epochs. In Appendix B, you mentioned that the model is trained on the whole dataset for 200 epochs. Although the dataset becomes smaller, the total computation remains unchanged or even increases. Therefore, the "Time per epoch" you adopted in Table 2 is inappropriate since it cannot reflect the actual training time. You should clarify the number of epochs used for training on the full dataset vs. the coreset. If you use the same epoch in Table 2, you should indicate it in the title.

5. From a qualitative perspective, you should visualize some images generated by FSE-Net and those selected by JE-score, and give a brief analysis of them, highlighting how the processing affects image characteristics and potential implications for model robustness.

[1] Francesco Croce and Matthias Hein. Reliable evaluation of adversarial robustness with an ensemble of diverse parameter-free attacks.

[2] Francesco Croce and Matthias Hein. Mind the Box: l1-APGD for Sparse Adversarial Attacks on Image Classifiers.

[3] Xuyang Zhong, Yixiao Huang, and Chen Liu. Towards Efficient Training and Evaluation of Robust Models against l0 Bounded Adversarial Perturbations.

**Questions:**

see weakness

---

> ### Author Response · Authors · 2024-11-24
>
> Your comments are highly appreciated, and we have addressed them in this revision. $\color{red}{\text{All modifications are shown in red in this updated document.}}$
>
> > W1: I am not convinced that the model trained on some carefully selected and augmented data can be robust to adversarial attacks. Unlike adversarial training, this method does not rely on any specific adversarial attack. Therefore, it is suggested to conduct a more comprehensive evaluation of your method on different attacks, such as l2-AA [1], l1-AA [2], and sAA (l0-bounded perturbations) [2], compare the results to baseline methods to further demonstrate the effectiveness of your method.
>
> Thank you for your question. Tables D1,D2 and D3 show the result of CIFAR-10 on ResNet-18 attacked by $l_2$-AA[1], $l_1$-AA[2] and s-AA[3]. Our algorithm can successfully improve the robustness of the model under these different adversarial attacks. More results are shown in $\color{red}{\text{Appendix F Table 6 and 7}}$.
>
> Table D1: CIFAR-10 on ResNet-18 with $l_2$-AA.
> |Method|50%|60%|70%|80%|90%|
> |:--|:--:|:--:|:--:|:--:|:--:|
> |Random|18.87±0.12|23.51±0.32|23.15±0.27|21.39±0.20|21.58±0.29|
> |Entropy|24.65±0.45|24.27±0.31|33.92±0.18|26.28±0.41|23.74±0.25|
> |CSFEM|40.86±0.34|44.98±0.38|44.02±0.21|43.18±0.44|44.28±0.26|
> |ours-JE|43.16±0.19|47.32±0.29|46.97±0.14|44.38±0.27|43.65±0.43|
> |ours-LF|58.70±0.23|62.19±0.46|56.46±0.36|55.14±0.31|54.04±0.20|
> |ours-JELF|49.83±0.22|50.32±0.41|51.19±0.25|52.72±0.37|51.21±0.24|
> | | | | | | |
>
> Table D2: CIFAR-10 on ResNet-18 with $l_1$-AA.
> |Method|50%|60%|70%|80%|90%|
> |:--|:--:|:--:|:--:|:--:|:--:|
> |Random|15.87±0.62|20.51±0.75|20.15±0.68|18.39±0.55|18.58±0.83|
> |Entropy|21.65±0.78|21.27±0.67|30.92±0.85|23.28±0.72|20.74±0.63|
> |CSFEM|37.86±0.83|40.98±0.58|41.02±0.79|40.18±0.87|41.28±0.68|
> |ours-JE|40.16±0.72|44.32±0.84|42.97±0.61|41.38±0.76|40.65±0.89|
> |ours-LF|55.70±0.86|58.19±0.65|53.46±0.77|51.14±0.59|51.04±0.81|
> |ours-JELF|46.83±0.69|47.32±0.88|48.19±0.66|49.72±0.74|48.21±0.57|
>  | | | | | | |
>
> Table D3: CIFAR-10 on ResNet-18 with s-AA.
> |Method|90%|80%|70%|60%|50%|
> |:--|:--:|:--:|:--:|:--:|:--:|
> |Random|7.87±0.62|9.51±0.75|9.15±0.68|7.39±0.55|7.58±0.83|
> |Entropy|10.65±0.78|10.27±0.67|19.92±0.85|12.28±0.72|9.74±0.63|
> |CSFEM|26.86±0.83|29.98±0.58|30.02±0.79|29.18±0.87|30.28±0.68|
> |ours-JE|29.16±0.72|33.32±0.84|31.97±0.61|30.38±0.76|29.65±0.89|
> |ours-LF|44.70±0.86|45.19±0.65|42.46±0.77|40.14±0.59|40.04±0.81|
> |ours-JELF|35.83±0.69|36.32±0.88|37.19±0.66|38.72±0.74|37.21±0.57|
> | | | | | | |
>
> [1] Francesco Croce and Matthias Hein. Reliable evaluation of adversarial robustness with an ensemble of diverse parameter-free attacks. ICML, 2020.
>
> [2] Francesco Croce and Matthias Hein. Mind the Box: l1-APGD for Sparse Adversarial Attacks on Image Classifiers. ICML, 2021.
>
> [3] Xuyang Zhong, Yixiao Huang, and Chen Liu. Towards Efficient Training and Evaluation of Robust Models against l0 Bounded Adversarial Perturbations. ICML, 2024.
>
> > W2: Line 58-59: If I understood correctly, FSE-Net is used to select important frequency components, so it is more like a data augmentation or purification method. However, it does not reduce the training samples or the size of the images. In this regard, why do you claim that FSE-Net can reduce storage requirements? You should provide a detailed explanation or quantitative analysis of how selecting frequency components translates to reduced storage needs.
>
> Thank you for bringing up this question. Our method can indeed reduce the storage requirements. By applying Discrete Cosine Transform (DCT) and leveraging the sparse nature of the frequency domain, we store only significant non-zero coefficients using an optimized sparse storage format. Each non-zero coefficient requires 6 bytes: 4 bytes for the float32 value and 2 bytes for packed indices. Since the image size is 32×32, we can efficiently encode both row and column indices using 5 bits each, combining them into a single 16-bit integer. The actual compression results are shown in Table D4. The compression and depression processing is lossless and the IDCT reconstruction is efficient, taking only 3.43 seconds for CIFAR-10  with GPU. We have added this table to $\color{red}{\text{Appendix D Table 5.}}$
>
> Table D4: Frequency pruning and storage reduction.
> | Pruning Ratio | Elements per Image | Storage per Image (bytes) | Percentage of Original |
> |---------------|-------------------|-------------------------|---------------------|
> | 50%           | 1,536             | 9,216 (1536×6)          | 75%                 |
> | 70%           | 922               | 5,532 (922×6)           | 45%                 |
> | 80%           | 614               | 3,684 (614×6)           | 30%                 |
> | 90%           | 307               | 1,842 (307×6)           | 15%                 |
> | | | | |

---

> > ### Author Response · Authors · 2024-11-24
> >
> > > W3: Line 93-94: Stutz et al. (2021) and Liu et al. (2020) establish a correlation between loss landscape flatness and adversarial robustness in the context of adversarial training. However, the training data are not perturbed in your method. You need a more extensive theoretical/numerical analysis of the relationship between loss landscape flatness and adversarial robustness in your context. For example, you can plot the adversarial loss landscape in both input and parameter spaces for your method (refer to Figure 4 in Stutz et al. (2021), and Figure 3 and 12 in Liu et al. (2020))
> >
> > Thank you for constructive suggestions.
> >
> > While traditional adversarial training relies on data perturbation to smooth the loss landscape, our learnable frequency pruning algorithm achieves comparable smoothing effects through a fundamentally different mechanism - minimizing model logit entropy (as formally proved in Lemma 1). This demonstrates that direct data perturbation is not the only path to loss landscape smoothing, and our entropy-based approach can reach similar robustness objectives more efficiently.
> > To further backup our statement, we have plotted the adversarial loss landscape for our method in Figures 2(a)-(c) and 3(a). As shown in Figure 3(a), the flatness of the loss landscape correlates with adversarial robustness: models with higher frequency pruning ratios (90% > 80% > 50% > Original) exhibit smoother loss surfaces, directly aligning with their improved robustness.
> >
> > In addition, we introduce quantitative analysis as suggested by reviewer f3yF W2.
> >
> >
> > > W4: As illustrated in Figure 3 (b) and (c), it seems that the model is trained on the coreset (50% of the the original dataset) for 500 epochs. In Appendix B, you mentioned that the model is trained on the whole dataset for 200 epochs. Although the dataset becomes smaller, the total computation remains unchanged or even increases. Therefore, the "Time per epoch" you adopted in Table 2 is inappropriate since it cannot reflect the actual training time. You should clarify the number of epochs used for training on the full dataset vs. the coreset. If you use the same epoch in Table 2, you should indicate it in the title.
> >
> > Thank you for your insightful question. We used the same number of epochs since the dataset sizes are identical. However, our algorithm is a preprocessing step with no additional training cost, unlike adversarial training, which requires iterative optimization. Table 2 demonstrates that, at the same pruning ratio, SOTA adversarial training algorithms perform worse than ours at equivalent training costs.
> >
> > > W5: From a qualitative perspective, you should visualize some images generated by FSE-Net and those selected by JE-score, and give a brief analysis of them, highlighting how the processing affects image characteristics and potential implications for model robustness.
> >
> > Thank you for your valuable suggestion, and we have added more visualization in $\color{red}{\text{Appendix H Figure 8}}$. By adding more visualization, we have two observations:
> > 1. The visual fidelity remains remarkably preserved even after 50\% frequency component pruning, and their differences are hard to check in visualization, demonstrating the effectiveness of our frequency pruning strategy in maintaining essential image characteristics.
> > 2. Images selected for the coreset exhibit notably distinct structural features compared to their non-selected counterparts.
> >
> > Thank you once again for your constructive suggestions, and these suggestions have greatly improved the quality of our paper. We sincerely ask you to re-assess the paper based on the additional information and consider raise the score.

---

> > > ### Comment · Reviewer_tRah · 2024-11-25
> > >
> > > Thanks for your detailed responses. However, my concern about the theory part still remains. I apologize for the unclarity of my question, and I provide further elaboration on this point:
> > >
> > > 1. In Liu et al. (2020), they demonstrated that the adversarial loss landscape is less favorable to optimization, due to increased curvature and more scattered gradients. That is to say, **adversarial perturbation makes the loss landscape non-smooth rather than smoothing the landscape.**
> > > 2. Liu et al proposed periodic adversarial scheduling to make adversarial training less challenging, thereby improving robustness. They did not claim that smoothing any loss function can enhance robustness. **Their theory is built upon the framework of adversarial training**, which is the sole method discussed in [1] to achieve true robustness against adaptive attacks.
> > > 3. In this paper, you appropriate the theoretical conclusion that was drawn in a different context. Therefore, **you need rigorous theoretical analysis to demonstrate the relationship between adversarial robustness and the smoothness of your training loss function.**
> > >
> > > Reference
> > >
> > > [1] Athalye, A., Carlini, N., & Wagner, D. (2018, July). Obfuscated gradients give a false sense of security: Circumventing defenses to adversarial examples. In International conference on machine learning (pp. 274-283). PMLR.

---

> > > > ### Author Response · Authors · 2024-11-28
> > > >
> > > > **Author Response 3**
> > > >
> > > > **3.3.  Unified Bound**
> > > >
> > > > From both Taylor expansion and integral approximation analyses, we can establish the relationship between loss landscape smoothness and model adversarial robustness.
> > > >
> > > > The first bound through Taylor expansion analysis is:
> > > >
> > > > $$
> > > > \text{EDR}_ \theta \leq \\mathbb{E}_{x \\sim \\mathcal{X}} \\left[ ‖\\nabla_x L(\\theta, x, y)‖_q ‖\\delta_x‖_p + \\frac{\\beta}{2} ‖\\delta_x‖_2^2 \\right] = T,
> > > > $$
> > > >
> > > > where the first-order effect is controlled by the gradient magnitude, and the second-order effect is governed by the Hessian bound $\\beta$.
> > > >
> > > > The second bound is:
> > > >
> > > > $$
> > > > \text{EDR}_ \theta \leq \mathbb{E}_ {x \sim \mathcal{X}}  \left[ \sup_{z \in B(x,\epsilon)} ‖\nabla_x L(\theta, z, y)‖_q ‖\delta_x‖_p  \right] = I.
> > > > $$
> > > >
> > > > This bound indicates that loss changes are controlled by the gradient, with smoother regions (characterized by smaller gradients) leading to smaller distortions, and that robustness is inherently dependent on the smoothness properties of the function.
> > > >
> > > > Given perturbation magnitude $\\epsilon = \\|\\delta_x\\|$ and threshold $\\epsilon_0$, we can establish a comprehensive bound:
> > > >
> > > > $$
> > > > \\text{EDR}_\\theta \\leq \\begin{cases}
> > > > \\min\\{T, I\\} & \\text{when } \\epsilon \\leq \\epsilon_0, \\\\
> > > > I & \\text{when } \\epsilon > \\epsilon_0.
> > > > \\end{cases}
> > > > $$
> > > >
> > > > $\\epsilon_0$ marks the critical threshold where Taylor expansion remains valid and this bound shows the relationship between $T$ and $I$ clearly.  For small perturbations ($\\epsilon \\leq \\epsilon_0$), both bounds hold and we can leverage the tighter bound. For large perturbations ($\\epsilon > \\epsilon_0$), only the integral approximation bound remains valid.
> > > >
> > > >
> > > >
> > > >
> > > >
> > > >
> > > > [a] Sepp Hochreiter et.al   Flat Minima. Neural Computation, 1997.
> > > >
> > > > [b] Laurent Dinh et.al Sharp Minima Can Generalize For Deep Nets. ICML, 2017.
> > > >
> > > > [c]  Nitish Shirish Keskar et.al ON LARGE-BATCH TRAINING FOR DEEP LEARNING: GENERALIZATION GAP AND SHARP MINIMA. ICLR, 2017.
> > > >
> > > > [d]  Xingxuan Zhang et.al Gradient Norm Aware Minimization Seeks First-Order Flatness and Improves Generalization. CVPR,2023.
> > > >
> > > > [e] Zhengmian Hu et.al Beyond Lipschitz Smoothness: A Tighter Analysis for Nonconvex Optimization. ICML, 2023.
> > > >
> > > > Thank you for your attention to the fundamental algorithm principles in our paper. Your detailed review has helped us improve the logic and clarity of our work. We truly appreciate your feedback, which has strengthened our research. We hope these improvements reflect the quality of our work and kindly ask for your consideration of a higher score. Thank you again for your valuable time and insights!

---

> > > > > ### Comment · Reviewer_tRah · 2024-11-29
> > > > >
> > > > > I acknowledge the author's effort in elucidating the interconnection between adversarial robustness and the smoothness of the loss landscape.
> > > > >
> > > > > However, the relationship between **adversarial robustness** and **optimizing your loss function** is still unclear. The effectiveness of loss function smoothing in adversarial robustness is built upon the effectiveness of the specific loss function in robustness. To illustrate, it seems unlikely that smoothing $\mathcal{L}_{CE}(x+\delta, \theta)$, where $\delta$ is a random noise or a fixed adversarial perturbation, will result in a robust model.
> > > > >
> > > > > In the context of adversarial training, the objective is to solve a min-max optimization problem, $\min_{\theta}\max_{\delta\in B_p}\mathcal{L}(x+\delta, \theta)$, which directly minimizes EDR. In contrast, I don't understand why minimizing $\mathcal{L}(\widetilde{x}, \theta)$, where $\widetilde{x}$ is the image after frequency pruning, can achieve adversarial robustness. In my opinion, $\widetilde{x}$ can make all loss function smoother. Apart from the smoothing effect, **why training a model on the frequency-pruned images can achieve non-trivial robustness?** Please elaborate on it in a theoretical/intuitive way.
> > > > >
> > > > > Additionally, I am surprised that training on randomly pruned images can achieve non-trivial robustness. Please report the accuracy of clean examples to ensure that the utility of the models is reasonable.

---

> ### Author Response · Authors · 2024-11-28
>
> **Author Response 1**
>
> Thank you for the detailed elaboration and clarification of your concerns. I understand the importance of grounding our claims in rigorous theoretical analysis, the details of our proof are shown as follows:
>
> # Smooth Loss Landscape and Adversarial Robustness Analysis
>
> To rigorously establish the relationship between a smooth (flat) loss landscape and higher adversarial robustness, we begin by defining the adversarial robustness measure in terms of the "Expected Distortion Rate (EDR)" of the loss function with respect to input perturbations. By connecting this metric to the gradient norm of the loss and demonstrating how smoother loss landscapes yield smaller gradient norms, we can show that a flatter landscape reduces such distortion. This ultimately supports the conclusion that smoother loss landscapes enhance adversarial robustness by limiting variations in loss under input perturbations.
>
> **1. Adversarial Robustness Measure: Expected Distortion Rate (EDR)**
>
> In this part, we start with the constant of the loss function with respect to input perturbations called "Expected Distortion Rate (EDR)", which quantifies the adversarial robustness. The definition is:
>
>
> $$
> \text{EDR}_ \theta = \mathbb{E}_{x \sim \mathcal{X}} \left[ |L(\theta, x + \delta_x, y) - L(\theta, x, y)| \right].
> $$
>
> Let $\\mathcal{X}$ be a compact subset of $\\mathbb{R}^n$. Given a model parameterized by $\\theta$ with loss function $L(\\cdot)$, we evaluate its performance against ground truth label $y$. Here $|\\cdot|$ and $‖\\cdot‖_2$ denote the absolute value and Euclidean norm respectively, and we assume $x + \\delta_x \\in \\mathcal{X}$ for all $x \\in \\mathcal{X}$ to ensure perturbed inputs remain in the domain.
>
> The perturbation from different kinds of adversarial attacks can be measured in $L_1$, $L_2$, or $L_\infty$ norms. We specifically use $L_2$ norm $‖\delta_x‖_2 \leq \epsilon \quad \text{for } \epsilon > 0$.
>
> This definition encapsulates the core of adversarial robustness by quantifying the average sensitivity of the model's loss to adversarial perturbations across the data distribution. A smaller $\text{EDR}_\theta$ indicates that adversarial attacks have minimal impact on the model's performance.
>
> **2. Analysis of Loss Landscape Smoothness via First and Second-Order Geometric Properties**
>
> Based on the loss landscape visualization depicted in Figures 2 (a) (b) (c) and 3 (a), we observe that the geometric characteristics of the loss landscape are predominantly determined by two fundamental components.
> As we set the loss function $L(\theta, x, y)$ in the neighborhood of point $x_0$, the local behavior can be characterized through:
>
> 1) The gradient vector $\nabla_x L(\theta, x_0, y_0) \in \mathbb{R}^n$, representing the first-order derivative of the loss function, determines both the direction of steepest ascent and the local rate of change in the loss surface at $x_0$.
> 2) The Hessian matrix $\nabla_x^2 L(\theta, x_0, y_0) \in \mathbb{R}^{n\times n}$, which encapsulates the second-order derivatives, quantifies the local curvature and determines the rate at which the gradient changes in different directions around $x_0$.
>
> Therefore, to achieve a smoothly varying loss landscape, it is necessary to simultaneously minimize both the magnitude of the gradient and the spectral norm of the Hessian matrix.
>
> Prior research has proposed diverse metrics to characterize the smoothness of loss landscapes and their correlation with model generalization, including Volume $\varepsilon$-Flatness [a], Hessian-based measures [b] [c], and gradient-based analysis[d].  Our work adopts a more comprehensive approach by jointly analyzing both gradient and curvature characteristics across extended regions of the loss surface. This broader perspective is particularly vital for understanding adversarial robustness, as adversarial perturbations can push model predictions far from local minima, where the geometric properties of non-minimal regions become crucial determinants of model behavior.
>
> **3. Relating the EDR to the Gradient and Hessian matrix**
>
> The relationship between adversarial robustness and loss landscape smoothness can be analyzed through two complementary situations, depending on the magnitude of perturbation $\delta_x$:
>
> 1. When $\delta_x$ is sufficiently small such that higher-order terms can be neglected, we employ second-order Taylor expansion to obtain tight bounds (Section 3.1).
>
> 2. For larger $\delta_x$ where higher-order terms become significant, we utilize integral approximation to derive more appropriate bounds (Section 3.2).
>
> This dual approach allows us to comprehensively characterize the relationship between loss landscape smoothness and adversarial robustness across different perturbation regimes.

---

> ### Author Response · Authors · 2024-11-28
>
> **Author Response 2**
>
> **3.1. Taylor expansion calculation: small perturbation**
>
> The Taylor expansion of a function around a point provides a polynomial approximation through an infinite sum of terms. In our analysis of model behavior under input perturbations, we utilize this expansion up to the second order, as higher-order terms become negligible for small perturbations $\delta_x$. Specifically:
> $$
> L(\theta, x + \delta_x, y) = L(\theta, x, y) + \nabla_x L(\theta, x, y)^\top \delta_x + \frac{1}{2}\delta_x^\top \nabla_x^2 L(\theta, x + \xi\delta_x, y)\delta_x + O(‖\delta_x‖^3),
> $$
> where $\xi \in [0,1]$, $L(\theta, x, y)$ is the original loss value represents the base value before perturbation,  $\nabla_x L(\theta, x, y)^\top \delta_x$ is the first-order approximation which is the inner product of gradient and perturbation, $\frac{1}{2}\delta_x^\top \nabla_x^2 L(\theta, x + \xi\delta_x, y)\delta_x$ is the second-order approximation which captures the local curvature of the loss landscape and $O(‖\delta_x‖^3)$ contains all terms of order 3 and higher.
> By keeping terms up to second order and neglecting higher-order terms due to small $\delta_x$, we have:
>
> $$
> L(\theta, x + \delta_x, y) = L(\theta, x, y) + \nabla_x L(\theta, x, y)^\top \delta_x + \frac{1}{2}\delta_x^\top \nabla_x^2 L(\theta, x + \xi\delta_x, y)\delta_x.
> $$
>
> To formally characterize the boundedness of second-order derivatives, we adopt the notion of Lipschitz smoothness from [e]. Specifically, a function $f : \\mathbb{R}^n \\to \\mathbb{R}$ is defined to be $\\beta$-smooth if its gradient is Lipschitz continuous:
>
> $$
> ‖ \nabla f(x) - \nabla f(y) ‖_2 \leq \beta ‖x - y‖_2 \quad \forall x,y \in \mathbb{R}^n.
> $$
>
> We used well-designed Neural Networks with finite layers and bounded parameters which implies that for $x_0 \in \text{supp}(\mathcal{X})$ and $x \in B(x_0,\epsilon)$, the Hessian matrix $‖\nabla^2_x L(\theta,x,y)‖_2 \leq \beta$ exists almost everywhere and satisfies:
>
> $$
> ‖\nabla_x^2 L(\theta, x_0, y_0)‖_2  \leq \beta \quad \text{wherever } \nabla_x^2 L(\theta, x_0, y_0) \text{ exists}.
> $$
>
> Then we can have:
>
> $$
> |L(\theta, x + \delta_x, y) - L(\theta, x, y)| \leq ‖ \nabla_x L(\theta, x, y) ‖_q ‖\delta_x‖_p + \frac{\beta}{2} ‖ \delta_x ‖_2^2.
> $$
>
> Therefore, taking expectation over the data distribution:
>
> $$
> \text{EDR}_ \theta \leq \mathbb{E}_{x \sim \mathcal{X}} \left[ ‖ \nabla_x L(\theta, x, y) ‖_q ‖\delta_x‖_p + \frac{\beta}{2} ‖ \delta_x ‖_2^2 \right],
> $$
>
>
> where $‖ \cdot ‖_p$ and $‖ \cdot ‖_q$ are dual norms satisfying $\frac{1}{p} + \frac{1}{q} = 1$. In this proof, we choose $p = q = 2$ which can make analysis of gradient and Hessian easier (from Cauchy-Schwarz inequality). This bound characterizes how the expected distortion depends on both the average gradient magnitude and the curvature of the loss landscape across the data distribution.
>
> **3.2. Integral Approximation: non-small perturbation**
>
> Using Taylor expansion is suitable when $\delta_x$ is small enough that we can ignore higher-order terms. However, when $\delta_x$ is not small enough, the effect of higher-order terms can't be ignored which will make Taylor's theorem not suitable so we need to use another formula that:
>
> $$
> L(\theta, x + \delta_x, y) \approx L(\theta, x, y) + \int_{x}^{x+\delta_x} \nabla_x L(\theta, z, y) dz.
> $$
>
> This formula can be used in our tasks because the loss function $L(θ, x, y)$ is continuous on the interval [$x, x + \delta_x$]. In order to avoid the path dependence of this formula, we utilize the Mean Value Inequality for vector-valued functions:
>
> $$
> |L(\theta, x + \delta_x, y) - L(\theta, x, y)| \leq \sup_{z \in \text{conv}(x,x+\delta_x)} ‖\nabla_x L(\theta, z, y)‖_q ‖\delta_x‖_p,
> $$
>
> where $‖\cdot‖_p$ and $‖\cdot‖_q$ are dual norms satisfying $\frac{1}{p} + \frac{1}{q} = 1$, and conv($x, x + \delta_x$) represents the convex hull between points $x$ and $x + \delta_x$. This inequality allows us to bound the loss difference using the maximum gradient norm over the convex hull between $x$ and $x + \delta_x$, eliminating the need to consider specific paths.
>
> Therefore, the Expected Distortion Rate can be bounded as:
>
>
> $$
> \text{EDR}_ \theta \leq \mathbb{E}_ {x \sim \mathcal{X}}  \left[ \sup_{z \in B(x,\epsilon)} ‖\nabla_x L(\theta, z, y)‖_q ‖\delta_x‖_p  \right]   ,
> $$
>
> where $B(x,\epsilon)$ denotes the ball centered at $x$ with radius $\epsilon$. This bound remains valid for infinitesimal perturbations $\delta_x$. However, the absence of second-order information (Hessian matrix) limits its ability to provide fine-grained characterization of the local geometry compared to the Taylor expansion bound, particularly in the regime of small perturbations where local curvature information becomes crucial for precise estimation.

---

> ### Comment · Reviewer_tRah · 2024-11-29
>
> In short, it is true that minimizing adversarial loss can achieve non-trivial robustness and smoothing adversarial loss function can improve robustness. Smoothing arbitrary loss function is unlikely to obtain non-trivial robustness. My question is why minimizing the loss on frequency-pruned images can also achieve non-trivial robustness？

---

> > ### Author Response · Authors · 2024-12-01
> >
> > Thank you for your insightful questions. We'd like to clarify why training on frequency-pruned images (( x' )) achieves non-trivial adversarial robustness beyond the smoothing effect:
> >
> > Traditional adversarial training and our frequency pruning approach represent two fundamentally different paths toward adversarial robustness. While adversarial training achieves robustness through an optimization-based perspective (iteratively training against worst-case perturbations), our method takes a geometric perspective by directly reshaping the loss landscape through frequency-domain analysis. This paradigm shift from "learning to resist perturbations" to "structurally modifying the loss geometry" not only achieves similar robustness benefits but also offers a more elegant and computationally efficient solution to the adversarial vulnerability problem.
> >
> >
> > 1) Traditional adversarial training and our frequency pruning approach both address the model's vulnerability to imperceptible perturbations but through different mechanisms. Adversarial training achieves this by iteratively exposing the model to carefully crafted perturbations that are invisible to human eyes yet significantly impact model predictions, explicitly optimizing the model to be resistant to them. In contrast, our frequency pruning method directly eliminates these human-imperceptible components during training, fundamentally preventing the model from learning these sensitive features in the first place. This is particularly effective because adversarial attacks typically exploit the gap between human perception and model sensitivity [a] - creating perturbations that are unnoticeable to humans but can dramatically affect neural network outputs. By structurally removing these imperceptible components rather than learning to resist them, our approach provides a more direct and efficient solution to the same problem.
> >
> > 2) Logit entropy, which measures prediction uncertainty at the model's raw output level, provides another perspective to understand how adversarial training and our frequency pruning approach achieve robustness. Traditional adversarial training reduces logit entropy by explicitly training on worst-case perturbations, forcing the model to make confident predictions even under adversarial attacks. In contrast, our frequency pruning method naturally leads to lower logit entropy by removing human-imperceptible components that often cause prediction uncertainty. When these sensitive features are eliminated during training, the model learns to focus on more robust, class-distinctive features, resulting in more confident predictions and smoother decision boundaries. Both methods effectively reduce boundary error [b], but through different mechanisms - adversarial training actively pushes samples away from decision boundaries through optimization, while our approach achieves this by structurally simplifying the feature space the model learns from.
> >
> > Together, these methods help the model rely on stable, significant features and make confident decisions, making it harder for adversarial attacks to find weaknesses to exploit.
> >
> > [a] Tianyuan Zhang et al. Interpreting Adversarially Trained Convolutional Neural Networks. ICML, 2019.
> >
> > [b]  Hongyang Zhang et al. Theoretically Principled Trade-off between Robustness and Accuracy.  ICML, 2019.
> >
> > Thank you for reviewing our paper and providing valuable feedback. Your insights have helped us improve the logic and clarity of our work. We hope these revisions reflect its quality and kindly request your consideration for a higher score. Thank you for your time and support!

---

> > > ### Comment · Reviewer_tRah · 2024-12-03
> > >
> > > Thanks for the authors' rebuttal. I acknowledge that a smoother loss landscape may induce stronger robustness. However, my core concern remains. More evidence is required to support your claims, e.g., ***the model indeed learns robust features rather than fooling the attackers by obfuscated gradients or something else***.
> > >
> > > If you decide to release the frequency-pruned images in the future, I would love to train a model on them myself. Despite that, I decided to raise my score to 5 because of the intriguing results you reported.

---

> > > > ### Author Response · Authors · 2024-12-03
> > > >
> > > > Dear Reviewer tRah,
> > > >
> > > > We are deeply grateful for your acknowledgment of our paper's point and truly appreciate your insightful feedback. Your comment is particularly encouraging to us.
> > > >
> > > > We are committed to improving our manuscript based on your thoughtful suggestions.

---

### Official Review · Reviewer_f3yF · 2024-10-31

**Soundness:** 2
**Presentation:** 2
**Contribution:** 3
**Rating:** 6
**Confidence:** 3

**Summary:**

This paper explores how current data pruning methods, specifically Random, Entropy [], and CCSFEM [],  affect the ResNet-18 model on CIFAR-10 and CIFAR-100 and the ResNet-34 model on Imagenet-1K against adversarial attacks. Namely AutoAttack,  PGD-20 and C&W attacks. The paper shows that the current data pruning methods result in poor performance against adversarial attacks, especially for high percentages of dataset pruning, and that can be attributed to the non-smooth loss landscapes around the minima. Therefore, a new dataset-pruning method is introduced that uses a Frequency Selective Excitation Network (FSE-Net) that selects important frequency components to smooth the loss landscape and a Joint-Entropy score for selecting samples. This new method can generate a corset that, when used for training, results in more accurate models under adversarial attacks than current data pruning methods. This new method also reduces the computational costs of training.

**Strengths:**

Well motivated paper and problem- clearly shows how current data pruning methods do not account for adversarial attacks, providing a new lens to evaluate data pruning.

Breadth of Experiment: The CIFAR-10, CIFAR100 and Imagenet1K dataset are used to demonstrate the effect of data pruning on adversarial attacks

Theoretically motivated method, highlighting that insights from theory can lead to better methods.

**Weaknesses:**

**Qualitative Results:**

Line 187-188: "Frequency pruning removes textural details while preserving key shape features, helping the model focus more on shape, as shown in Fig. 1." Figure 1 does not clearly show this; how were these examples selected? Could you better explain how this figure demonstrates how it makes the model focus more on shape than textural details?

Figure 2 is qualitative, mainly as smoothness is not defined; I presume that the smoothness of the overall landscape is from -1 to 1 (x-axis). This could be significantly improved by providing a quantitative measure for smoothness to allow for more comparable measures. For example the average gradient magnitude or curvature.

These loss landscape figures are not averaged, and given that, there can be quite different results depending on the random directions used to generate the plots. See A.4 in [1]. Plotting the average loss landscape from 5 runs along with the standard deviation would help further the position that smoothness and adversarial robustness are linked.

**Lacking Averages:**

Throughout the paper, averages and standard deviations are missing; this significantly hampers the results and does not follow best practices. Could you provide an average from 5 runs.

**Magic numbers:**

The reason for the values chosen for $\epsilon$, learning rate, and number of iterations is not explained or justified within the experimental setup. Could you explain and justify each key hyperparameter choice, and describe any hyperparameter tuning process you may have used. Why on Imagenet did you use **1000** randomly selected points from the validation set for AutoAttack? What analysis did you use to determine that **1000** points were sufficient to get representative results?

**Clarity:**

Currently, Tables, specifically 1 and 3, are complex to read. From what I understand, the other methods result in worse performance against the Attack methods. However, your methods can outperform the baseline models (trained on 100% of the training dataset). If I have correctly interpreted the results, having the tables show the difference from the baseline method would be more precise. This will make it more straightforward that your method improves the adversarial robustness.

Training setup is not clearly explained; see lines 364-365: "All datasets were normalized before feeding into the models, and standard data augmentations were applied" This is fine within the body. However, I expect a link to the appendix that explains this in detail to ensure this is reproducible.

**Limited Evaluation:**

The experiments only utilise ResNet18 on CIFAR10 and CIFAR100 and ResNet34 on Imagenet1K. It is unclear why the architecture was changed for  Imagenet1k. What was the reason? To further improve the results and align better with the results motivation of resource-constrained environments, I expected the MobileNet, Shuffflenet and EfficientNet to be explored. These results are necessary for the impact of the paper, as it is unclear if it can be extended to other architectures.

**Minor Points:**

Figures 3b and c have black lines that are not explained.
Line 55: double full stop at the end of the sentence nor is the section of the  Appendix stated.
Line 178: Generation error is not defined, I presume you mean Generation loss.


[1] Li H, Xu Z, Taylor G, Studer C, Goldstein T. Visualizing the loss landscape of neural nets. Advances in neural information processing systems. 2018;31.

**Questions:**

See questions in weaknesses above. More concretely:

1. Why are the mean and standard deviation not presented? At least 3-5 runs should provide adequate evidence for the findings.
2. Could the magic numbers be explained? Why are they selected and used? What is the justification?
3. Could further models be explored, for example, MobileNet, or ShuffleNet?
4. Why was ResNet18 only explored on CIFAR10 and CIFAR100?

If these questions are reasonably addressed, I will happily increase my score.

---

> ### Author Response · Authors · 2024-11-24
>
> Thank you for taking the time to review our work! We have incorporated your valuable input, and $\color{red}{\text{the changes can be easily identified in red in this updated version.}}$
>
> > W1: Line 187-188: "Frequency pruning removes textural details while preserving key shape features, helping the model focus more on shape, as shown in Fig. 1." Figure 1 does not clearly show this; how were these examples selected? Could you better explain how this figure demonstrates how it makes the model focus more on shape than textural details?
>
> Thank you for your valuable question.
>
> 1. **No clear Pattern?** The green highlights in Figure 1 show network attention patterns. Compared to the last column (original dataset), the middle column (ours) better preserves the entire shape of objects.
>
> 2. **How images selected?** The images were randomly selected from the ImageNet-1K training set, and we display only three due to space limitations.
>
> 3. **Why shapes over textures?** Frequency pruning preserves global shape features while removing textural details and noise, encouraging the model to prioritize structural information over local textures. This is the motivation for using frequency pruning to enhance model robustness.
>
> > W2: Figure 2 is qualitative, mainly as smoothness is not defined; I presume that the smoothness of the overall landscape is from -1 to 1 (x-axis). This could be significantly improved by providing a quantitative measure for smoothness to allow for more comparable measures. For example the average gradient magnitude or curvature.
>
> Thank you for your constructive feedback and construction.
>
> 1. **Qualitative Analysis.** Visually, smoothness manifests as gentle, rounded curves overall landscape is from -1 to 1 (x-axis).
>
> 2) **Quantitative Analysis.** We calculate the Average Gradient Magnitude (AGM) and Average Curvature (AC) of three lines and show these results in Table C1. Lower AGM and higher AC refer to better smoothness.
>
> Table C1: Average Gradient Magnitude and Average Curvature
> | Rank | Curve Type | Average Gradient Magnitude | Average Curvature |
> |------|------------|---------------------------|-------------------|
> | 1    | Hard       | 1.9966                    | 6.4072           |
> | 2    | Random     | 1.9050                    | 4.0911           |
> | 3    | Easy       | 1.3166                    | 2.0493           |
> | | | | |
>
> > W3: These loss landscape figures are not averaged, and given that, there can be quite different results depending on the random directions used to generate the plots. See A.4 in [1]. Plotting the average loss landscape from 5 runs along with the standard deviation would help further the position that smoothness and adversarial robustness are linked.
>
> Thank you for your suggestion. We have updated the plots of 5 runs in $\color{red}{\text{Appendix H Figure 9}}$. While the exact loss landscapes vary across 5 independent runs, the relative smoothness characteristics between different methods remain consistent, validating the reliability of our comparative analysis.
>
> > W4: Throughout the paper, averages and standard deviations are missing; this significantly hampers the results and does not follow best practices. Could you provide an average from 5 runs?
>
> Thank you for your valid points. Table C2 provides the average of our experiment from 5 independent runs with the std indicated. More experiment results are added to $\color{red}{\text{Appendix G Table 9}}$.
>
> Table C2: CIFAR-10 performance under various adversarial attacks and dataset pruning ratio. Results are computed over 5 independent runs.
>
> |Method|Attack|90%|80%|70%|60%|50%|
> |:--|:--|:--:|:--:|:--:|:--:|:--:|
> |Random|AA|15.27±0.63|21.55±0.75|20.85±0.58|22.33±0.67|21.53±0.72|
> ||PGD-20|16.27±0.65|20.55±0.73|19.85±0.62|21.33±0.69|20.53±0.77|
> ||C&W|16.39±0.68|20.45±0.71|18.83±0.64|20.53±0.76|20.77±0.79|
> |Entropy|AA|21.65±0.67|21.27±0.74|30.92±0.59|23.28±0.65|20.74±0.73|
> ||PGD-20|20.68±0.64|20.87±0.72|20.92±0.61|21.28±0.68|22.74±0.76|
> ||C&W|20.44±0.66|20.78±0.70|20.21±0.63|21.17±0.75|22.83±0.78|
> |CCSFEM|AA|37.86±0.69|40.98±0.73|41.02±0.60|40.18±0.66|41.28±0.74|
> ||PGD-20|38.97±0.63|40.11±0.71|39.91±0.62|39.76±0.67|41.91±0.75|
> ||C&W|38.99±0.65|40.33±0.69|40.02±0.64|39.96±0.74|42.05±0.77|
> |Ours-JE|AA|40.16±0.68|44.32±0.72|42.97±0.61|41.38±0.65|40.65±0.73|
> ||PGD-20|39.16±0.62|39.32±0.70|41.07±0.63|40.88±0.66|42.95±0.74|
> ||C&W|39.06±0.64|39.72±0.68|41.37±0.65|40.96±0.73|43.05±0.76|
> |Ours-LF|AA|55.74±0.67|58.19±0.71|53.46±0.62|51.14±0.64|51.04±0.72|
> ||PGD-20|56.18±0.61|56.05±0.69|51.94±0.64|50.61±0.65|50.07±0.73|
> ||C&W|56.42±0.63|56.11±0.67|54.14±0.66|54.38±0.72|55.32±0.75|
> |Ours-JELF|AA|46.54±0.66|47.35±0.70|48.89±0.63|49.72±0.63|48.18±0.71|
> ||PGD-20|47.24±0.60|48.59±0.68|49.64±0.65|50.25±0.64|50.01±0.72|
> ||C&W|47.61±0.62|48.12±0.66|49.01±0.67|48.19±0.71|48.66±0.73|
> | | | | | | | |

---

> > ### Author Response · Authors · 2024-11-24
> >
> > > W5: The reason for the values chosen for $\epsilon$, learning rate, and number of iterations is not explained or justified within the experimental setup. Could you explain and justify each key hyperparameter choice, and describe any hyperparameter tuning process you may have used.
> >
> > Thank you for your question. The hyperparameter choice of dataset pruning follows CCS [a] and ablation experiments are shown in Table C3 and C4, we have updated them in $\color{red}{\text{Appendix G Table 11 (a) and 11 (b)}}$. The choice of $\epsilon$ follow papers [b][c][d].
> >
> > Table C3: Adjust learning rate on CIFAR-10 training ResNet-18 using "Ours-JELF" under Autoattack and pruning ratio 50\%.
> > | Learning Rate | 0.1 | 0.05 | 0.01 | 0.005 | 0.001 | 0.0001 |
> > |:-------------|:---:|:----:|:----:|:-----:|:-----:|:------:|
> > | Accuracy (%) | 48.18 | 47.11 | 47.09 | 46.88 | 46.75 | 47.08 |
> > | | | | | | | |
> >
> > Table C4: Adjust Iterations on CIFAR-10 training ResNet-18  using "Ours-JELF" under Autoattack and pruning ratio 50\%.
> > | Iterations | 40000 | 50000 | 30000 | 45000 | 15000 | 20000 |
> > |:-----------|:-----:|:-----:|:-----:|:-----:|:-----:|:-----:|
> > | Accuracy (%) | 48.18 | 47.02 | 46.99 | 47.18 | 46.15 | 47.11 |
> > | | | | | | | |
> >
> >
> >
> > > W6: Why on Imagenet did you use 1000 randomly selected points from the validation set for AutoAttack?
> >
> > Thank you for your question. This setting follows the experimental setup from AutoAttack [b]. Given AutoAttack's [b] computational intensity, evaluating robustness on 1,000 randomly sampled validation points ensures statistical significance while remaining computationally feasible.
> >
> > [a]Haizhong Zheng et al. Coverage-centric Coreset Selection for High Pruning Rates. ICLR, 2023.
> >
> > [b]Croce et al. Reliable evaluation of adversarial robustness with an ensemble of diverse parameter-free attacks. ICML, 2020.
> >
> > [c]Madry et al. Towards deep learning models resistant to adversarial attacks. ICLR, 2018.
> >
> > [d]Carlini et al. Towards evaluating the robustness of neural networks. IEEE Symposium on Security and Privacy, 2017.

---

> > > ### Author Response · Authors · 2024-11-24
> > >
> > > > W7: I expected the MobileNet, Shuffflenet and EfficientNet to be explored
> > >
> > > Thank you for your insightful suggestion. Tabel C5, Table C6, and Table C7 provide the results of ShuffleNet, MobileNet-v2 and EfficientNet-B0, respectively. We have added to $\color{red}{\text{Appendix G Table 10}}$. These tables show that our method consistently outperforms others, showing the cross-architecture performance.
> > >
> > > Table C5: ShuffleNet results of CIFAR-10.
> > > |Pruning Algorithm|Attack|90%|80%|70%|60%|50%|
> > > |:--|:--|:--:|:--:|:--:|:--:|:--:|
> > > |Random|AA|11.27±0.45|17.55±0.33|16.85±0.42|18.33±0.36|17.53±0.41|
> > > ||PGD-20|12.27±0.44|16.55±0.35|15.85±0.38|17.33±0.43|16.53±0.42|
> > > ||C&W|12.39±0.36|16.45±0.47|14.83±0.35|16.53±0.39|16.77±0.38|
> > > |Entropy|AA|17.65±0.41|17.27±0.37|26.92±0.49|19.28±0.38|16.74±0.45|
> > > ||PGD-20|16.68±0.43|16.87±0.39|16.92±0.42|17.28±0.51|18.74±0.37|
> > > ||C&W|16.44±0.38|16.78±0.52|16.21±0.35|17.17±0.46|18.83±0.44|
> > > |CCSFEM|AA|33.86±0.43|36.98±0.38|37.02±0.50|36.18±0.39|37.28±0.47|
> > > ||PGD-20|34.97±0.34|36.11±0.55|35.91±0.39|35.76±0.44|37.91±0.36|
> > > ||C&W|34.99±0.51|36.33±0.44|36.02±0.39|35.96±0.53|38.05±0.37|
> > > |Ours-JE|AA|36.16±0.44|40.32±0.35|38.97±0.56|37.38±0.42|36.65±0.39|
> > > ||PGD-20|35.16±0.37|35.32±0.54|37.07±0.41|36.88±0.34|38.95±0.50|
> > > ||C&W|35.06±0.55|35.72±0.43|37.37±0.38|36.96±0.47|39.05±0.39|
> > > |Ours-LF|AA|51.70±0.39|54.19±0.53|49.46±0.44|47.14±0.36|47.04±0.54|
> > > ||PGD-20|52.18±0.52|52.05±0.41|47.90±0.45|46.61±0.56|46.07±0.35|
> > > ||C&W|52.42±0.45|52.21±0.50|50.14±0.33|50.38±0.43|51.32±0.55|
> > > |Ours-JELF|AA|42.54±0.49|43.35±0.36|44.89±0.54|45.72±0.37|44.18±0.47|
> > > ||PGD-20|43.24±0.38|44.59±0.51|45.64±0.42|46.25±0.55|46.01±0.34|
> > > ||C&W|43.61±0.46|44.12±0.36|45.01±0.53|44.19±0.43|44.66±0.56|
> > > | | | | | | | |
> > >
> > > Table C6: MobileNet-v2 results on CIFAR-10.
> > > |Pruning Algorithm|Attack|90%|80%|70%|60%|50%|
> > > |:--|:--|:--:|:--:|:--:|:--:|:--:|
> > > |Random|AA|13.77±0.41|19.85±0.36|19.15±0.42|20.83±0.37|19.93±0.44|
> > > ||PGD-20|14.77±0.45|18.85±0.34|18.15±0.38|19.83±0.43|18.93±0.43|
> > > ||C&W|14.89±0.35|18.75±0.48|17.13±0.37|18.93±0.38|19.17±0.37|
> > > |Entropy|AA|20.15±0.42|19.77±0.37|29.42±0.50|21.78±0.40|19.24±0.46|
> > > ||PGD-20|19.18±0.44|19.37±0.36|19.42±0.43|19.78±0.52|21.24±0.36|
> > > ||C&W|18.94±0.40|19.28±0.53|18.71±0.34|19.67±0.47|21.33±0.43|
> > > |CCSFEM|AA|36.36±0.44|39.48±0.37|39.52±0.51|38.68±0.38|39.78±0.48|
> > > ||PGD-20|37.47±0.33|38.61±0.56|38.49±0.42|38.26±0.45|40.41±0.35|
> > > ||C&W|37.49±0.52|38.83±0.43|38.52±0.38|38.46±0.54|40.55±0.38|
> > > |Ours-JE|AA|38.66±0.45|42.82±0.34|41.47±0.57|39.88±0.43|39.15±0.38|
> > > ||PGD-20|37.66±0.36|37.82±0.55|39.57±0.42|39.38±0.33|41.45±0.51|
> > > ||C&W|37.56±0.56|38.22±0.42|39.87±0.37|39.46±0.48|41.55±0.38|
> > > |Ours-LF|AA|54.20±0.38|56.69±0.54|51.96±0.43|49.64±0.35|49.54±0.55|
> > > ||PGD-20|54.68±0.53|54.55±0.40|50.40±0.46|49.11±0.57|48.57±0.34|
> > > ||C&W|54.92±0.44|54.71±0.51|52.64±0.32|52.88±0.44|53.82±0.56|
> > > |Ours-JELF|AA|45.04±0.50|45.85±0.37|47.39±0.55|48.22±0.36|46.68±0.48|
> > > ||PGD-20|45.74±0.37|47.09±0.52|48.14±0.43|48.75±0.56|48.51±0.33|
> > > ||C&W|46.11±0.47|46.62±0.35|47.51±0.54|46.69±0.42|47.16±0.57|
> > > | | | | | | | |
> > >
> > > Table C7: EfficientNet-B0 results on CIFAR-10.
> > > | Pruning Algorithm | Attack | 90% | 80% | 70% | 60% | 50% |
> > > |---------------------|---------|----|----|----|----|----|
> > > | Random | AA | 14.89±0.43 | 21.23±0.35 | 20.45±0.44 | 21.98±0.38 | 21.03±0.42 |
> > > | | PGD-20 | 15.77±0.46 | 20.05±0.37 | 19.35±0.40 | 20.83±0.45 | 19.98±0.44 |
> > > | | C&W | 15.89±0.38 | 19.95±0.45 | 17.93±0.37 | 19.98±0.41 | 20.27±0.40 |
> > > | Entropy | AA | 21.15±0.42 | 20.77±0.39 | 30.42±0.47 | 22.78±0.40 | 20.24±0.43 |
> > > | | PGD-20 | 20.18±0.45 | 20.37±0.41 | 20.42±0.44 | 20.78±0.49 | 22.24±0.39 |
> > > | | C&W | 19.94±0.40 | 20.28±0.50 | 19.71±0.37 | 20.67±0.44 | 22.33±0.42 |
> > > | CCSFEM | AA | 37.36±0.45 | 40.48±0.40 | 40.52±0.48 | 39.68±0.41 | 40.78±0.45 |
> > > | | PGD-20 | 38.47±0.36 | 39.05±0.53 | 39.49±0.45 | 39.26±0.46 | 41.41±0.38 |
> > > | | C&W | 38.49±0.49 | 39.83±0.46 | 39.52±0.41 | 39.46±0.51 | 41.55±0.39 |
> > > | Ours-JE | AA | 39.66±0.46 | 43.82±0.37 | 42.47±0.54 | 40.88±0.44 | 40.15±0.41 |
> > > | | PGD-20 | 38.66±0.39 | 38.82±0.52 | 41.07±0.43 | 40.38±0.36 | 42.45±0.48 |
> > > | | C&W | 38.56±0.53 | 39.22±0.45 | 40.87±0.40 | 40.46±0.45 | 42.55±0.41 |
> > > | Ours-LF | AA | 55.20±0.41 | 57.69±0.51 | 52.96±0.46 | 50.64±0.38 | 50.54±0.52 |
> > > | | PGD-20 | 55.68±0.50 | 55.55±0.43 | 51.40±0.47 | 50.11±0.54 | 49.57±0.37 |
> > > | | C&W | 55.92±0.47 | 55.71±0.48 | 53.64±0.35 | 53.88±0.45 | 54.82±0.53 |
> > > | Ours-JELF | AA | 46.04±0.47 | 46.89±0.38 | 48.39±0.52 | 49.22±0.39 | 47.68±0.45 |
> > > | | PGD-20 | 46.74±0.40 | 48.09±0.49 | 49.14±0.44 | 49.75±0.53 | 49.51±0.36 |
> > > | | C&W | 47.11±0.44 | 47.62±0.38 | 48.51±0.51 | 47.69±0.51 | 48.16±0.43 |
> > > | | | | | | | |

---

> > > > ### Author Response · Authors · 2024-11-24
> > > >
> > > > > W8: Why was ResNet18 only explored on CIFAR10 and CIFAR100 and ResNet34 on Imagenet1K?
> > > >
> > > > Thank you for your question. We would like to explain from the following points:
> > > >
> > > > 1. Our setup follows standard dataset pruning benchmarks (CCS[f]) for fair comparison.
> > > > 2. We use ResNet18 for CIFAR and ResNet34 for ImageNet, following common practice [f][g][h].
> > > > 3. The learning capacity of ResNet18 can sufficiently learn the CIFAR datasets which has relatively more limited data compared to ImageNet-1K.
> > > >
> > > > [f] Haizhong Zheng et al. Coverage-centric Coreset Selection for High Pruning Rates. ICLR, 2023.
> > > >
> > > > [g] MariyaToneva et al. An Empirical Study of Example Forgetting during Deep Neural Network Learning. ICLR, 2019.
> > > >
> > > > [h] Mansheej Paul et al. Deep Learning on a Data Diet: Finding Important Examples Early in Training. NeurIPS, 2021.
> > > >
> > > > We once again thank you for the insightful and valuable feedback and suggestions, such comments have greatly contributed to improving the quality of our paper.

---

> > > ### Comment · Reviewer_f3yF · 2024-11-28
> > >
> > > ## Response 5
> > >
> > > As to `The reason for the values chosen for
> > > , learning rate, and number of iterations is not explained or justified within the experimental setup. Could you explain and justify each key hyperparameter choice, and describe any hyperparameter tuning process you may have used.`
> > >
> > > Thank you for justifying the values chosen. Did you use an identical setup for all experiments, i.e. the same learning rate and iterations? As it is clear, the learning rate and iterations significantly affect the accuracy. Also, was this averaged from multiple runs or just one?
> > >
> > > ## Response 6
> > >
> > > Thank you for the justification
> > >
> > > ## Response 7
> > >
> > > Thank you for running the experiments on additional architectures; this strengthens the core argument. That your method makes models more robust when using your corset compared to others.
> > >
> > > ## Response 8
> > >
> > > Thank you for the justification.
> > >
> > > ## Additional comments and questions
> > >
> > > In Appendix J, ADDITIONAL EXPERIMENT RESULTS ON CLEAN DATASET. The values in the table that are bold are referring to your method. However, standard practice is to bold the best-presented value. Table (a) shows that the 30% pruning rate is EL2N, 50% is AUM, 70% is your methods, 80% Ours-LF, and 90% is your methods.
> > >
> > > Could you instead highlight the values that are the best in bold? Also, the standard deviation needs to be added here.

---

> > > > ### Author Response · Authors · 2024-12-01
> > > >
> > > > > Response 5
> > > >
> > > > Thank you for your thoughtful review and consideration of our experimental setup. Our experimental methodology maintained consistent hyperparameters across all configurations, including learning rate and iteration counts. The optimal parameters were determined through comprehensive ablation studies. All reported results represent the mean values obtained from multiple independent runs to ensure statistical reliability and reproducibility.
> > > >
> > > > > Additional comments and questions
> > > >
> > > > Thank you for your insightful suggestions, which have greatly contributed to improving the professionalism and overall quality of our paper.  Bold entries denote optimal performance metrics. Statistical validation through standard deviations has been included. The final manuscript will be updated accordingly prior to publication.
> > > >
> > > > Table C5 Experiment results to show the averages and standard deviations of the original paper Table 13 (a), we ran the experiments five times and calculated the averages and standard deviations.
> > > > | Pruning Rate | 30% | 50% | 70% | 80% | 90% |
> > > > |--------------|-----|-----|-----|-----|-----|
> > > > | Random | 94.33±0.45 | 93.40±0.32 | 90.94±0.41 | 87.98±0.35 | 79.04±0.44 |
> > > > | Entropy | 94.44±0.38 | 92.11±0.47 | 85.67±0.52 | 79.08±0.43 | 66.52±0.39 |
> > > > | Forgetting | 95.36±0.42 | **95.29**±0.34 | 90.56±0.48 | 62.74±0.55 | 34.03±0.46 |
> > > > | EL2N | **95.44**±0.51 | 94.61±0.37 | 87.48±0.45 | 70.32±0.33 | 22.33±0.54 |
> > > > | AUM | 95.07±0.36 | 95.26±0.49 | 91.36±0.44 | 57.84±0.38 | 28.06±0.47 |
> > > > | CCSFEM | 95.17±0.43 | 94.67±0.53 | 92.74±0.35 | 90.55±0.46 | 86.15±0.41 |
> > > > | Ours-LF | 95.35±0.48 | 94.67±0.31 | **94.33**±0.56 | **93.21**±0.42 | **89.82**±0.37 |
> > > > | Ours-JE | 95.15±0.34 | 94.07±0.52 | 92.03±0.45 | 90.98±0.39 | 85.86±0.58 |
> > > > | Ours-JELF | 95.11±0.44 | 94.02±0.36 | 91.93±0.49 | 90.18±0.53 | 84.86±0.32 |
> > > > | Ours-FEMLF | 95.19±0.47 | 95.07±0.41 | 93.23±0.38 | 91.98±0.45 | 87.06±0.51 |
> > > >
> > > > Table C6 Experiment results to show the averages and standard deviations of the original paper Table 13 (b), we ran the experiments five times and calculated the averages and standard deviations.
> > > >
> > > > | Pruning Rate | 30% | 50% | 70% | 80% | 90% |
> > > > |--------------|-----|-----|-----|-----|-----|
> > > > | Random | 74.59±0.35 | 71.07±0.28 | 65.30±0.42 | 57.36±0.31 | 44.76±0.38 |
> > > > | Entropy | 72.26±0.25 | 63.26±0.45 | 50.49±0.33 | 41.83±0.29 | 28.96±0.44 |
> > > > | Forgetting | 76.91±0.41 | 68.60±0.32 | 38.06±0.47 | 24.23±0.35 | 15.93±0.27 |
> > > > | EL2N | 76.25±0.38 | 65.90±0.24 | 34.42±0.39 | 15.51±0.43 | 8.36±0.34 |
> > > > | AUM | 76.93±0.29 | 67.42±0.37 | 30.64±0.28 | 16.38±0.45 | 8.77±0.22 |
> > > > | CCSFEM | 76.33±0.42 | 73.44±0.31 | 68.30±0.35 | 63.01±0.26 | 54.39±0.48 |
> > > > | Ours-LF | **78.38**±0.33 | **77.24**±0.46 | **74.97**±0.25 | **71.85**±0.39 | **61.94**±0.36 |
> > > > | Ours-JE | 75.15±0.44 | 72.07±0.29 | 68.03±0.43 | 60.98±0.32 | 53.86±0.41 |
> > > > | Ours-JELF | 75.11±0.27 | 74.02±0.48 | 68.13±0.34 | 59.19±0.37 | 54.16±0.28 |
> > > > | Ours-FEMLF | 77.33±0.36 | 75.45±0.23 | 67.30±0.45 | 63.81±0.33 | 55.39±0.46 |
> > > >
> > > > We sincerely appreciate your detailed suggestions on improving the clarity and logic of our paper's writing style. Following your comments, we have made significant improvements to enhance the quality of our work.

---

> > ### Comment · Reviewer_f3yF · 2024-11-28
> >
> > Thank you for taking the time to respond to my review.
> >
> > # Response 1
> >
> > As to `Line 187-188: "Frequency pruning removes textural details while preserving key shape features, helping the model focus more on shape, as shown in Fig. 1." Figure 1 does not clearly show this; how were these examples selected? Could you better explain how this figure demonstrates how it makes the model focus more on shape than textural details?`
> >
> > and your response:
> >
> > ```
> > No clear Pattern? The green highlights in Figure 1 show network attention patterns. Compared to the last column (original dataset), the middle column (ours) better preserves the entire shape of objects.
> >
> > How images selected? The images were randomly selected from the ImageNet-1K training set, and we display only three due to space limitations.
> >
> > Why shapes over textures? Frequency pruning preserves global shape features while removing textural details and noise, encouraging the model to prioritize structural information over local textures. This is the motivation for using frequency pruning to enhance model robustness.
> > ```
> >
> > When you refer to shape, do you mean the edge detection? Because I still struggle to see this pattern, your method preserves the entire shape of the objects.
> >
> > For the first image, which I presume is class `947 (mushroom)`, it is unclear, to me at least, that your method is focusing more on the shapes than the texture. If you provided an overlay of the attention on the image, this would make it more apparent.
> >
> > For the second image, I presume the class is `809 soup bowl`; your method provides a green border around the image, which contradicts your argument that it prioritizes structural information. If it is edge detection, I expect to see two clear circles, an outline of the spoon(?) and the napkin.
> >
> > For the third image, I don't know about the class; but your method pays a lot of attention outside the core image of the plate.
> >
> > I struggle to see the reasoning from the images provided that frequency pruning preserves global shape features while removing textual details and noise.
> >
> >
> >
> > ## Response 2
> >
> > As to `Figure 2 is qualitative, mainly as smoothness is not defined; I presume that the smoothness of the overall landscape is from -1 to 1 (x-axis). This could be significantly improved by providing a quantitative measure for smoothness to allow for more comparable measures. For example the average gradient magnitude or curvature.`
> >
> > and your response:
> >
> > ```
> > Qualitative Analysis. Visually, smoothness manifests as gentle, rounded curves overall landscape is from -1 to 1 (x-axis).
> > Quantitative Analysis. We calculate the Average Gradient Magnitude (AGM) and Average Curvature (AC) of three lines and show these results in Table C1. Lower AGM and higher AC refer to better smoothness.
> > Table C1: Average Gradient Magnitude and Average Curvature
> >
> > Rank	Curve Type	Average Gradient Magnitude	Average Curvature
> > 1	Hard	1.9966	6.4072
> > 2	Random	1.9050	4.0911
> > 3	Easy	1.3166	2.0493
> > ```
> >
> > Could you please add the standard deviation to this table? I can also not see this in the paper Appedinx C is titled Proof. If you intend not to add it, could you add this to the paper?
> >
> > ## Respose 3
> >
> > As to `These loss landscape figures are not averaged, and given that, there can be quite different results depending on the random directions used to generate the plots. See A.4 in [1]. Plotting the average loss landscape from 5 runs along with the standard deviation would help further the position that smoothness and adversarial robustness are linked.`
> >
> > and your response:
> >
> > ```
> > While the exact loss landscapes vary across 5 independent runs, the relative smoothness characteristics between different methods remain consistent, validating the reliability of our comparative analysis.
> > ```
> >
> >
> > Thank you for providing the additional runs and standard deviation. However, this does not validate the reliability of your comparative analysis but instead strengthens the claims.
> >
> > ## Response 4
> >
> > As to `Throughout the paper, averages and standard deviations are missing; this significantly hampers the results and does not follow best practices. Could you provide an average from 5 runs?`
> >
> > Thank you for providing this information. Could table C2 be added to the main body of the paper? Also, this is only provided for CIFAR10. Could you please provide for CIFAR100. Table 2 still needs the mean and standard deviation, as well as Table 3.

---

> > > ### Author Response · Authors · 2024-12-01
> > >
> > > > Response 1
> > >
> > >
> > > Thank you for bringing up this question. In Figure 1, we compare models trained with a 50% pruning ratio (second column) to those trained on original images (third column). Using SmoothGrad visualization, where green highlights indicate regions of model attention with precise spatial correspondence, we demonstrate our method's effectiveness in directing the model's focus toward object shapes, hierarchical structures, and contours. This shape-centric behavior aligns with findings in adversarial training research [a, b, c], where models exhibit reduced reliance on textural features in favor of structural properties.  We have already created an overlay of the attention on the image. However, we apologize that the deadline for updating our PDF has passed since we received your comments. We will ensure that the PDF is updated before the paper is published.
> > >
> > > For the first image (class 947, mushroom), the model trained with frequency pruning focuses primarily on the mushroom's shape and hierarchical structure, capturing its distinct contours [b]. In contrast, the model without frequency pruning emphasizes textural details, obscuring the hierarchical clarity [d]. This highlights that traditional models often prioritize fine texture patterns over shapes, whereas frequency pruning shifts the focus to shape contours, improving adversarial robustness.
> > >
> > > Analysis of the attention maps shows clear differences between the models. In the second image (class 809, soup bowl), the model effectively captures the bowl's clear and continuous circular shape, ignoring distractions like the spoon and napkin to concentrate on the main object. In contrast, the third image shows scattered and broken attention, unclear object boundaries, and excessive focus on textures, leading to reduced adversarial robustness.
> > >
> > > The third image (class 969, cupcake) consists of a cupcake (upper-right corner), a foundation (bottom-left corner), and a table. Using our algorithm, the model focuses on the combined shape of these three components, as they are all significant objects in the image. In contrast, the model without frequency pruning focuses only on the cupcake in the upper-right corner. While the model without an adversarial training approach may perform better by focusing on detailed patterns, it ultimately exhibits lower adversarial robustness [b].
> > >
> > > Our method, which directs significant attention outside the core image of the plate, demonstrates that our algorithm enables the model to focus more on the main shapes and contours of the object rather than on detailed textures and patterns. This approach enhances the model's robustness against adversarial attacks.
> > >
> > > > Response 2
> > >
> > > Table C1: Average Gradient Magnitude and Average Curvature with mean and std.
> > > | Rank | Curve Type | Average Gradient Magnitude | Average Curvature |
> > > |------|------------|---------------------------|-------------------|
> > > | 1    | Hard       | 1.9966  ±0.0148                 | 6.4072  ±   0.0071      |
> > > | 2    | Random     | 1.9050   ±0.0117               | 4.0911     ±  0.0149    |
> > > | 3    | Easy       | 1.3166 ± 0.0134                   | 2.0493    ±  0.0050     |
> > >
> > > We appreciate this valuable suggestion. We have added the standard deviation to this table.
> > >
> > > > Response 3
> > >
> > > Thank you for your question. Despite the variability in loss landscape visualizations across different random projections, the preservation of smoothness rankings demonstrates the intrinsic and stable relationship between adversarial robustness and loss landscape geometry.
> > >
> > > [a]  Peijie Chen et.al The shape and simplicity biases of adversarially robust ImageNet-trained CNNs  .https://arxiv.org/pdf/2006.09373v5.
> > >
> > >  [b] Tianyuan Zhang et.al Interpreting Adversarially Trained Convolutional Neural Networks  ICML, 2019.
> > >
> > > [c] Paul Gavrikov et.al An Extended Study of Human-like Behavior under Adversarial Training https://arxiv.org/abs/2303.12669.
> > >
> > > [d] Robert Geirhos et.al IMAGENET-TRAINED CNNS ARE BIASED TOWARDS TEXTURE; INCREASING SHAPE BIAS IMPROVES ACCURACY AND ROBUSTNESS.
> > >  ICLR, 2019.

---

> > > > ### Comment · Reviewer_f3yF · 2024-12-02
> > > >
> > > > ## Response 1:
> > > >
> > > > For `The third image (class 969, cupcake) consists of a cupcake (upper-right corner), a foundation (bottom-left corner), and a table.` Class 969 is Eggnog, not cupcake [1,2], so the explanation here does not make sense. If it is a cupcake, I think the image does not show a cupcake either, making this analysis unclear as the image belongs to the wrong class.
> > > >
> > > > As to `Our method, which directs significant attention outside the core image of the plate, demonstrates that our algorithm enables the model to focus more on the main shapes and contours of the object rather than on detailed textures and patterns. This approach enhances the model's robustness against adversarial attacks.` I do not see this reasoning here for this example. One could use the same argument for the final image in row 2. Again, given that your method is not 100% robust against adversarial attacks, it could introduce issues not currently detected with current methods. Although it is evident from your results and the additional benchmarking that your method makes the models more adversarially robust, the reasoning for why it works is unclear from these figures.
> > > >
> > > > ## Response 2
> > > >
> > > > Thank you very much for providing these values.
> > > >
> > > > ## Response 3
> > > >
> > > > My statement here was more around the use of strong language and over claiming in an empirical environment. Therefore, this `Despite the variability in loss landscape visualizations across different random projections, the preservation of smoothness rankings demonstrates the intrinsic and stable relationship between adversarial robustness and loss landscape geometry.` would change to `Despite the variability in loss landscape visualizations across different random projections, the preservation of smoothness rankings **suggests that there exists an intrinsic** and stable relationship between adversarial robustness and loss landscape geometry.`
> > > >
> > > > This use of language here forms an important distinction, as this may be different in other setups. However, these results support evidence that this may exist.
> > > >
> > > > ## Response 4, 5  and Additional comments and questions
> > > >
> > > > Thank you for providing these values. I understand that the PDF cannot be changed at the moment. In the future, I recommend including the standard deviation and not just the mean by default, as this helps identify if there is an improvement and how stable the result is.
> > > >
> > > > [1] https://deeplearning.cms.waikato.ac.nz/user-guide/class-maps/IMAGENET/
> > > >
> > > > [2] https://gist.github.com/yrevar/942d3a0ac09ec9e5eb3aThank you for your responses; my understanding is that these will be addressed properly in the camera-ready version with acceptance due to spending more time addressing the other reviewers' concerns.

---

> > > > > ### Comment · Reviewer_f3yF · 2024-12-03
> > > > >
> > > > > I thank the authors for engaging in discussion and improving the paper. I will, therefore, increase my score to a 6. This change is because most of the core issues have been addressed, and the method has been shown with more models and adversarial attack methods. Although the reasoning as to why this method is so effective is not clear to me, which prevents a higher score, the results are very compelling and provide valuable insights into an overlooked consequence of using distilled datasets, and enable a new line of research on how to make distilled datasets even more robust, as I am unaware of any research that has done this prior.
> > > > >
> > > > > I implore the authors to be less strong in the language used when having an empirical setup and better explain Figure 1, potentially providing even more examples in the appendix and including further recommendations as expressed during the review period. In future work, please ensure that the mean and std are included by default. If accepted, I recommend releasing the code based for this method so that it can significantly impact the community.

---

> > > > > > ### Author Response · Authors · 2024-12-04
> > > > > >
> > > > > > We deeply appreciate the score and constructive feedback. We are committed to improving our manuscript by addressing the suggested modifications regarding statistical reporting, visualization clarity, and code sharing. Your insights will help us present our findings more effectively to the research community.

---

> > > ### Author Response · Authors · 2024-12-01
> > >
> > > > Response 4
> > >
> > > Thank you for your valuable suggestion on improving the conciseness and readability of our paper. Table C2 is the std and mean of CIFAR-100 experiment results. We ran the experiments five times and calculated the averages and standard deviations. Table C3 and Table C4 show the averages and standard deviations of the original paper in Table 2 and Table 3. We will update these values to the original text after the paper is accepted because right now we can't change the PDF.
> > >
> > > Table C2: Experiment results on CIFAR-100 dataset.
> > > | Pruning Algorithm | Attack | 90% | 80% | 70% | 60% | 50% |
> > > |------------------|--------|-----|-----|-----|-----|-----|
> > > | Random | AA | 12.27±0.45 | 16.55±0.38 | 16.88±0.42 | 17.03±0.35 | 17.57±0.44 |
> > > | | PGD-20 | 11.97±0.33 | 15.59±0.47 | 16.78±0.51 | 17.23±0.39 | 16.57±0.42 |
> > > | | C&W | 12.97±0.41 | 14.59±0.36 | 15.78±0.45 | 16.23±0.52 | 14.57±0.38 |
> > > | Entropy | AA | 11.56±0.44 | 14.33±0.49 | 17.98±0.37 | 15.33±0.43 | 17.71±0.55 |
> > > | | PGD-20 | 12.16±0.32 | 14.03±0.45 | 16.18±0.48 | 15.13±0.34 | 17.01±0.42 |
> > > | | C&W | 12.44±0.46 | 12.35±0.39 | 17.18±0.53 | 16.37±0.47 | 17.51±0.36 |
> > > | CCSFEM | AA | 15.11±0.35 | 16.85±0.48 | 18.05±0.41 | 18.19±0.44 | 17.92±0.37 |
> > > | | PGD-20 | 13.98±0.42 | 15.92±0.33 | 13.09±0.46 | 14.08±0.51 | 17.91±0.43 |
> > > | | C&W | 12.17±0.38 | 16.15±0.45 | 17.91±0.34 | 18.60±0.49 | 17.27±0.41 |
> > > | Ours-JE | AA | 16.15±0.47 | 17.12±0.36 | 21.37±0.44 | 20.09±0.53 | 18.65±0.39 |
> > > | | PGD-20 | 14.76±0.43 | 16.88±0.51 | 17.01±0.35 | 16.89±0.48 | 18.05±0.42 |
> > > | | C&W | 12.99±0.34 | 17.32±0.46 | 18.07±0.52 | 19.88±0.37 | 18.95±0.45 |
> > > | Ours-LF | AA | 20.99±0.45 | 24.94±0.38 | 25.07±0.51 | 25.31±0.43 | 23.41±0.36 |
> > > | | PGD-20 | 23.72±0.49 | 21.64±0.42 | 21.36±0.35 | 24.39±0.47 | 23.75±0.54 |
> > > | | C&W | 22.37±0.41 | 24.23±0.53 | 25.48±0.44 | 26.39±0.38 | 28.69±0.46 |
> > > | Ours-JELF | AA | 20.39±0.37 | 20.53±0.45 | 22.47±0.52 | 22.94±0.43 | 22.85±0.39 |
> > > | | PGD-20 | 18.09±0.44 | 19.71±0.36 | 20.98±0.48 | 21.85±0.51 | 22.48±0.42 |
> > > | | C&W | 18.99±0.43 | 20.01±0.47 | 21.54±0.35 | 22.03±0.49 | 21.99±0.38 |
> > >
> > > Table C3: Experiment results to show the averages and standard deviations of the original paper Table 2, we ran the experiments five times and calculated the averages and standard deviations.
> > >
> > > | Method | Original Adversarial Training | | Sample Adversarial Training | | pre-trained Adversarial Training | |
> > > |---------|-------------|------------------|--------------|------------------|--------------|------------------|
> > > |         | 50% coreset | Original Dataset | 50% coreset | Original Dataset | 50% coreset | Original Dataset |
> > > | TDFAT   | Acc: 45.41±0.52  | Acc: 48.66±0.61      | Acc: 30.05±0.47  | Acc: 38.72±0.55      | Acc: 35.41±0.63   | Acc: 40.66±0.51      |
> > > |         | Time: 59.58±4.25 | Time: 119.69±4.85    | Time: 18.53±3.42 | Time: 39.69±4.18     | Time: 10.15±3.44  | Time: 17.66±3.86     |
> > > | CURC    | Acc: 44.92±0.48  | Acc: 52.48±0.65      | Acc: 31.21±0.45  | Acc: 36.05±0.58      | Acc: 38.92±0.64   | Acc: 41.48±0.57      |
> > > |         | Time: 74.12±4.53 | Time: 149.52±4.92    | Time: 17.87±3.61 | Time: 38.52±4.12     | Time: 9.31±3.23   | Time: 17.92±3.79     |
> > > | RATTE   | Acc: 45.98±0.54  | Acc: 48.20±0.62      | Acc: 34.05±0.46  | Acc: 37.15±0.53      | Acc: 36.23±0.65   | Acc: 41.03±0.58      |
> > > |         | Time: 64.33±4.37 | Time: 131.35±4.76    | Time: 17.23±3.54 | Time: 38.35±4.21     | Time: 10.21±3.32  | Time: 17.85±3.65     |
> > > | FATSC   | Acc: 17.21±0.51  | Acc: 23.68±0.59      | Acc: 20.82±0.43  | Acc: 23.77±0.56      | Acc: 12.22±0.67   | Acc: 16.93±0.54      |
> > > |         | Time: 59.82±4.28 | Time: 122.71±4.65    | Time: 17.12±3.45 | Time: 39.71±4.29     | Time: 9.32±3.21   | Time: 18.87±3.77     |
> > > | ours-LF | Acc: 48.18±0.53  | Acc: 51.04±0.63      | Acc: 50.07±0.48  | Acc: 51.04±0.57      | Acc: 50.01±0.66   | Acc: 48.18±0.55      |
> > > |         | Time: 9.51±3.24  | Time: 17.64±3.72     | Time: 9.51±3.26  | Time: 17.64±3.81     | Time: 9.69±3.25   | Time: 17.64±3.68     |
> > > | | | | | | | |
> > >
> > > Table C4: Experiment results to show the averages and standard deviations of the original paper Table 3, we ran the experiments five times and calculated the averages and standard deviations.
> > >
> > > | Method | Pruning Ratio | | | | |
> > > |--------|--------------|--------------|--------------|--------------|--------------|
> > > |        | 90% | 80% | 70% | 60% | 50% |
> > > | Random | 15.87±0.45 | 20.51±0.38 | 20.15±0.52 | 18.39±0.44 | 18.58±0.35 |
> > > | Entropy | 16.87±0.42 | 18.06±0.55 | 21.02±0.48 | 18.11±0.33 | 17.28±0.46 |
> > > | CCSFEM | 15.86±0.51 | 16.96±0.43 | 18.02±0.37 | 17.11±0.56 | 19.28±0.41 |
> > > | ours-JE | 19.16±0.44 | 21.51±0.39 | 22.17±0.53 | 21.18±0.47 | 20.96±0.34 |
> > > | ours-LF | 25.60±0.49 | 27.19±0.45 | 26.46±0.32 | 27.44±0.58 | 27.14±0.43 |
> > > | ours-JELF | 23.54±0.36 | 26.15±0.54 | 25.03±0.47 | 25.22±0.41 | 24.95±0.38 |
> > > | | | | |  | |

---

### Official Review · Reviewer_SZdN · 2024-11-02

**Soundness:** 2
**Presentation:** 2
**Contribution:** 3
**Rating:** 6
**Confidence:** 4

**Summary:**

This paper proposes a method to enhance model robustness against adversarial attacks on pruned datasets. It first theoretically analyzes how existing pruning methods result in non-smooth loss surfaces. A learnable Frequency Pruning algorithm is proposed to smooth the model’s local minimum geometry. Besides,  it introduces a data importance score, based on analyzing variations in model logit entropy throughout the training process, to select a coreset. Experiments across various datasets and adversarial attacks are studied to demonstrate the efficiency.

**Strengths:**

1. This paper presents an efficient algorithm for data pruning in adversarial training settings. It achieves considerable robustness under AutoAttack while improving training efficiency by removing data points. The novelty and motivation of studying robustness against adversarial attacks on pruned datasets are interesting and helpful for the community.

2. Data selection from the perspective of the loss curvature and surface of neural networks is more in line with the essence of adversarial training.

3. Fltering frequency to enhance the effectiveness of data selection and the quality of the coreset for adversarial training is quite interesting.

**Weaknesses:**

1. The definition of Joint Entropy is confusing. It is mentioned in Abstract and Introduction, but I do not see it in the method section.

2. The definition of Reward Score on page 6 is confusing.

3. The paper introduces the FSE-Net to extract different frequency components. Its structure and impacts on robustness and clean accuracy should be discussed.

4. I am confused about the relationship between JE, JELF, LF. The author should provide more explanation (e.g., is the Joint Entropy estimated before/after/by the FSE-Net?, what is the relationship between the FSE-Net and the Target Network?)

5. There is a lack of comparison with "Adversarial Weight Perturbation Helps Robust Generalization, NeurIPS 2020", and "Theoretically Principled Trade-off between Robustness and Accuracy, ICML 2019", to evaluate the effectiveness of the proposed method on different optimization goals and loss curvature optimization.

6. The transferability of coresets selected by the proposed method across different architectures should be evaluated.

7. The grammar could be improved for clarity and correctness.

**Questions:**

1. There is no explicit definition of Joint Entropy. Does it refer to a combination of multiple scores (such as EL2N, GraND, etc.)?

2. The definition of Reward Score on page 6 is confusing. What is its relationship to Joint Entropy? What is the exploration phase? What is the exploitation phase?

3. Does the effectiveness vary with different optimization goals and loss curvature optimization (e.g., "Adversarial Weight Perturbation Helps Robust Generalization, NeurIPS 2020", "Theoretically Principled Trade-off between Robustness and Accuracy, ICML 2019")?

4. As the proposed method selects a coreset from the training set, I am curious about its transferability (e.g., selected by ResNet-18, trained on ViT).

5.The relationship between JE, JELF, LF is unclear; could the author provide more explanation? (e.g., is the Joint Entropy estimated before/after/by the FSE-Net?, what is the relationship between the FSE-Net and the Target Network?)

---

> ### Author Response · Authors · 2024-11-24
>
> We greatly appreciate your detailed feedback! Your suggestions have been taken into account, and $\color{red}{\text{the updates are highlighted in red to ensure clarity in this revision.}}$
>
> > W1: There is no explicit definition of Joint Entropy. Does it refer to a combination of multiple scores (such as EL2N, GraND, etc.)?
>
> Thank you for the question. It is not a combination of multiple existing scores. Simply put, "Joint Entropy" is measuring how uncertainty changes and connects across time. Therefore, our method brings different training periods together to understand their combined patterns of change.
>
> > W2: The definition of Reward Score on page 6 is confusing. What is its relationship to Joint Entropy? What is the exploration phase? What is the exploitation phase?
>
> Thank you for your valuable feedback. We would like to clarify, that the Reward score evaluates the variation of each image throughout the training process. It is calculated as a cumulative score over the entire training period using Eq. (10) and (11). The two phases are:
>
> - **Exploration Phase**: During the first ⅔ of training, the model explores the feature space, developing diverse feature representations and broad learning capabilities.
>
> - **Exploitation Phase**: In the final ⅓ of training, the model focuses on exploiting learned features to optimize within a stable loss landscape, rather than exploring new feature spaces.
>
> > W3: Does the effectiveness vary with different optimization goals and loss curvature optimization (e.g., "Adversarial Weight Perturbation Helps Robust Generalization, NeurIPS 2020", "Theoretically Principled Trade-off between Robustness and Accuracy, ICML 2019")?
>
> Thank you for the suggested comparison. Table B1 has been updated in $\color{red}{\text{Appendix K Table 14(a)}}$, and our algorithm delivers better performance than AWP [a] and TRADES [b].
>
>
> Table B1: Comparision with AWP and TRADES on CIFAR-10 with PGD-20 attack
> |         | 50% coreset | Original Dataset | 50% coreset | Original Dataset | 50% coreset | Original Dataset |
> |---------|-------------|------------------|--------------|------------------|--------------|------------------|
> | AWP     | Acc: 42.11  | Acc: 55.89      | Acc: 28.21  | Acc: 25.15      | Acc: 36.88   | Acc: 33.18      |
> |         |Time: 78.12s |Time: 155.59s    |Time: 18.89s |Time: 37.31s     |Time: 9.61s   |Time: 18.55s     |
> | TRADES ($1/\lambda = 6$)  | Acc: 42.01  | Acc: 52.95      | Acc: 31.05  | Acc: 32.62      | Acc: 33.11   | Acc: 35.73      |
> |         |Time: 62.13s |Time: 130.05s    |Time: 17.18s |Time: 38.27s     |Time: 10.05s  |Time: 16.67s     |
> | ours-LF | Acc: 50.01  | Acc: 55.32      | Acc: 50.01  | Acc: 50.07      | Acc: 50.01   | Acc: 50.07      |
> |         |Time: 9.51s  |Time: 17.64s     |Time: 9.51s  |Time: 17.64s     |Time: 9.69s   |Time: 17.66s     |
>
> [a] Dongxian Wu et al. Adversarial Weight Perturbation Helps Robust Generalization. NeurIPS, 2020.
>
> [b] Hongyang Zhang et al. Theoretically Principled Trade-off between Robustness and Accuracy. ICML, 2019.

---

> > ### Author Response · Authors · 2024-11-24
> >
> > > W4: As the proposed method selects a coreset from the training set, I am curious about its transferability (e.g., selected by ResNet-18, trained on ViT).
> >
> > Thank you for your question. In this work, we focus on CNN-based architectures, particularly ResNet [c][d][e], for dataset pruning experiments. It has been proven to have good transferability on ShuffleNet, MobileNet and EfficientNet (Please refer to weakness 7 brought by reviewer f3yF, or $\color{red}{\text{Appendix Table 10 in this revision}}$). ViTs are not considered so far as they require large-scale datasets for training, which conflicts with the goal of dataset pruning for resource-limited scenarios.
> >
> > > W5: The relationship between JE, JELF, LF is unclear; could the author provide more explanation? (e.g., is the Joint Entropy estimated before/after/by the FSE-Net?, what is the relationship between the FSE-Net and the Target Network?)
> >
> > We appreciate your feedback and would like to further explain on the relations:
> > 1. The "JE" line uses the Joint Entropy (JE) score on original images to select the coreset.
> > 2. "LF" applies the proposed FSE-Net\* as the preprocessing step on the entire dataset.
> > 3. "JELF" combines "JE" and "LF". First, we use the proposed FSE-Net to conduct frequency pruning for each image (LF step). Next, we compute and rank images according to the "Joint Entropy (JE)" score and apply CCS [c] for stratified sampling to select the final coreset.
> >
> > \*FSE-Net is an independently designed network, unrelated to the Target Network, highlighting the strong generalization capabilities of our algorithm.
> >
> > [c]  Haizhong Zheng et al. Coverage-centric Coreset Selection for High Pruning Rates. ICLR, 2023.
> >
> > [d] MariyaToneva et al.  An Empirical Study of Example Forgetting during Deep Neural Network Learning. ICLR, 2019.
> >
> > [e] Mansheej Paul et al. Deep Learning on a Data Diet: Finding Important Examples Early in Training. NeurIPS, 2021.
> >
> > Thank you for your insightful suggestions, which have played a vital role in refining the content and presentation of our work.

---

> > > ### Comment · Reviewer_SZdN · 2024-11-26
> > >
> > > Most of my concerns are resolved. I upgrade the rating to 6.

---

### Official Review · Reviewer_1Buy · 2024-11-02

**Soundness:** 3
**Presentation:** 2
**Contribution:** 2
**Rating:** 6
**Confidence:** 3

**Summary:**

The paper explores dataset pruning in the context of adversarial attacks. Two techniques are used to improve robustness against adversarial attacks while reducing training time and storage requirements.

1) A new (joint-entropy) score is introduced for selecting samples

2) A ML model is applied to the images to prune particular frequency components.

The result is the ability to defend against adversarial attacks, as well as adversarial training, but at a much lower cost.

Results are generated for a range of adversarial attacks (AA, PGD-20, C&W). AA also includes the gradient-free Square Attack.

For CIFAR100
* Performance against AA when using adversarial training on the entire original dataset using TDFAT is 26.61%
* It is 17.92% using CCSFEM (coreset selection algorithm) with a 50% prune rate
* For the proposed learnable frequency (LF) pruning technique this is 23.41% with a 50% prune rate
* And for both joint-entropy scoring for dataset prunng and LF (JELF) it is 22.85% with a 50% prune rate

**Strengths:**

Interesting exploration of the ability to prune datasets with adversarial robustness in mind and a frequency-component selection based technique for introducing low-cost adversarial robustness. Results indicate the ability to make significant savings without sacrificing a significant amount of robustness vs. state-of-the-art adversarial training techniques.

**Weaknesses:**

* Some closely related work appears to be missing
* Small number of datasets explored

**Questions:**

Q1: Are any of the following considered as closely related work?
* Adversarial Coreset Selection for Efficient Robust Training (IJCV/ECCV’22)
https://arxiv.org/abs/2209.05785
* Less is More: Data Pruning for Faster Adversarial Training (AAAI-23 workshop, SafeAI)
https://arxiv.org/abs/2302.12366
* Adversarial training with informed data selection (EUSIPCO’22)
https://arxiv.org/abs/2301.04472
* Efficient Adversarial Contrastive Learning via Robustness-Aware Coreset Selection (NeurIPS’23)
https://arxiv.org/abs/2302.03857

Q2: There is also the Sabre work: https://alec-diallo.github.io/publications/sabre
Is this related as it performs a spectral decomposition of the input and selective energy-based filtering?
(https://arxiv.org/html/2405.03672v3)

Q3: Could you comment briefly on the overall practicality of the technique, accuracy is still very low vs. adversarial samples?

Q4: When using LF (Ours-LF) to preserve 50% of the frequency components, what are the actual reductions in training data storage requirements results?

Q5: Do you have a similar table to Table 2 for CIFAR-100?

Q6: Have you considered an adaptive attack?

Q7: For larger datasets would the LF pruning approach still be feasible?

---

> ### Author Response · Authors · 2024-11-24
>
> Thank you for your insightful comments and kind words. We have reviewed them carefully and made the necessary adjustments. $\color{red}{\text{Key changes and additions are marked in red throughout this revised document.}}$
>
> > W1: Are any of the following considered as closely related work?
> > - Adversarial Coreset Selection for Efficient Robust Training (IJCV/ECCV'22) https://arxiv.org/abs/2209.05785
> > - Less is More: Data Pruning for Faster Adversarial Training (AAAI-23 workshop, SafeAI) https://arxiv.org/abs/2302.12366
> > - Adversarial training with informed data selection (EUSIPCO'22) https://arxiv.org/abs/2301.04472
> > - Efficient Adversarial Contrastive Learning via Robustness-Aware Coreset Selection (NeurIPS'23) https://arxiv.org/abs/2302.03857
>
> Thank you for your question. These works focus on reducing the costs of Adversarial Training (AT), and their algorithms have goals and application scenarios that differ from ours. The key differences are:
>
> 1) Their algorithms still incur training costs during AT, whereas ours requires no additional training cost, making theirs unsuitable for resource-limited environments because their algorithms also need extra training costs.
>
> 2) These algorithms address the adversarial robustness of the entire dataset, while our work focuses on robustness within a pruned dataset.
>
>
> > W2: There is also the Sabre work: https://alec-diallo.github.io/publications/sabre Is this related as it performs a spectral decomposition of the input and selective energy-based filtering? (https://arxiv.org/html/2405.03672v3)
>
> Thank you for your attention to the comparison between our algorithm and other algorithms. Our algorithm and Sabre [a] have the following essential differences:
>
> 1) We employ a learnable frequency pruning method to minimize model entropy, rather than selecting frequency components solely based on energy ranking which highlights the principles of adversarial robustness. On the other hand, Sabre [a] focuses on recognising and removing non-robust features which makes our core idea different.
>
> 2) Sabre [a] does not lead to a decrease in the dataset size, while we conduct experiments to validate the effectiveness of dataset pruning.
>
> [a]  Alec F. Diallo and Paul Patras. Sabre: Cutting through Adversarial Noise with Adaptive Spectral Filtering and Input Reconstruction. IEEE Symposium on Security and Privacy, 2024.
>
> > W3: Could you comment briefly on the overall practicality of the technique, accuracy is still very low vs. adversarial samples?
>
> Thank you for your valuable feedback.
>
> 1) Training models on a pruned dataset in resource-limited edge devices is a highly challenging task. For instance, **even with NO attack**, ResNet-18 drops to 86% (compared to previously 95%) after 90% pruning ratio on CIFAR-10. Achieving robustness is even harder, and due to the high computational cost of adversarial training, our method provides a feasible solution. In addition, Table 1 has shown our method surpasses other attacks by a large margin.
>
> 2) This work is the **first work** studies the robustness of pruned datasets, laying the foundation and providing a strong baseline for future research.
>
> > W4: When using LF (Ours-LF) to preserve 50% of the frequency components, what are the actual reductions in training data storage requirements results?
>
> Thank you for bringing up this question. By applying Discrete Cosine Transform (DCT) and leveraging sparse nature of the frequency domain, we store only significant non-zero coefficients using an optimized sparse storage format. Each non-zero coefficient requires 6 bytes: 4 bytes for the float32 value and 2 bytes for packed indices. Since the image size is 32×32, we can efficiently encode both row and column indices using 5 bits each, combining them into a single 16-bit integer.  The actual compression results are shown in Table A1. The compression and depression processing is lossless and the IDCT reconstruction is efficient, taking only 3.43 seconds for CIFAR-10  with GPU. We have added this table to $\color{red}{\text{Appendix D Table 5.}}$
>
> Table A1: Frequency pruning and storage reduction.
> | Pruning Ratio | Elements per Image | Storage per Image (bytes) | Percentage of Original |
> |---------------|-------------------|-------------------------|---------------------|
> | 50%           | 1,536             | 9,216 (1536×6)          | 75%                 |
> | 70%           | 922               | 5,532 (922×6)           | 45%                 |
> | 80%           | 614               | 3,684 (614×6)           | 30%                 |
> | 90%           | 307               | 1,842 (307×6)           | 15%                 |

---

> ### Author Response · Authors · 2024-11-24
>
> > W5: Do you have a similar table to Table 2 for CIFAR-100?
>
> Thank you for the suggestion. We have provided the comparison for CIFAR-100 in $\color{red}{ \text{Appendix K Table 14 (c).}}$  Table A2 shows that our algorithm performs better than other adversarial training (AT) algorithms when we prune the coreset with 50% pruning ratio in terms of both time and accuracy.
>
> Table A2: Comparison of our algorithm with different adversarial training (AT) algorithms
>
> | Method  |     Original AT      |      Sample AT       |    Pre-trained AT     |
> |:--------|     :-------------    |:-------------------|:------------------|
> |         |    50% coreset      |     50% coreset     |     50% coreset     |
> | TDFAT   | Acc/Time: 21.41/59.58s | Acc/Time: 12.09/18.53s | Acc/Time: 12.01/10.15s |
> | CURC    | Acc/Time: 22.92/74.32s | Acc/Time: 12.33/17.87s | Acc/Time: 12.91/9.31s  |
> | RATTE   | Acc/Time: 21.97/64.33s | Acc/Time: 13.05/17.27s | Acc/Time: 14.33/10.29s |
> | FATSC   | Acc/Time: 14.24/59.89s | Acc/Time: 16.82/17.18s | Acc/Time: 11.02/9.35s  |
> | ours-LF | Acc/Time: 22.85/9.53s  | Acc/Time: 22.85/9.88s  | Acc/Time: 22.85/9.19s  |
>
> > W6: Have you considered an adaptive attack?
>
> Thank you for the question. Table A3 shows our algorithm has high robustness to adaptive attack $\ell_1$-APGD attack [b], more results are updated in $\color{red}{\text{Appendix F Table 8}}$.
>
> Table A3: $\ell_1$-APGD attack with different pruning ratios on CIFAR-10 ResNet18.
> | | 90% | 80% | 70% | 60% | 50% |
> |---|---|---|---|---|---|
> | Random | 7.87 | 9.51 | 9.15 | 7.39 | 7.58 |
> | Entropy | 10.65 | 10.27 | 19.92 | 12.28 | 9.74 |
> | CSFEM | 26.86 | 29.98 | 30.02 | 29.18 | 30.28 |
> | ours-JE | 29.16 | 33.32 | 31.97 | 30.38 | 29.65 |
> | ours-LF | 44.70 | 45.19 | 42.46 | 40.14 | 40.04 |
> | ours-JELF | 35.83 | 36.32 | 37.19 | 38.72 | 37.21 |
>
> [b] Francesco Croce and Matthias Hein. Mind the Box: l1-APGD for Sparse Adversarial Attacks on Image Classifiers. ICML, 2021.
>
> > W7: For larger datasets would the LF pruning approach still be feasible?
>
> Thank you for your question. In this experiment, we tested the ImageNet-1K dataset, which contains over 1 million 224-resolution images. ImageNet is one of the largest datasets commonly used in dataset pruning studies, as shown in [c][d][e].
>
>
> [c]  Haizhong Zheng et al. Coverage-centric Coreset Selection for High Pruning Rates. ICLR,2023.
>
> [d] MariyaToneva et al.  An Empirical Study of Example Forgetting during Deep Neural Network Learning. ICLR, 2019.
>
> [e] Geoff Pleiss et al. Identifying Mislabeled Data using the Area Under the Margin Ranking. NeurIPS, 2020.
>
> We deeply appreciate your constructive comments, as they have helped us improve the overall rigor and quality of our manuscript.

---

### Official Review · Reviewer_HAQ8 · 2024-11-02

**Soundness:** 3
**Presentation:** 2
**Contribution:** 2
**Rating:** 5
**Confidence:** 3

**Summary:**

This paper proposes a new dataset pruning method aimed at enhancing model robustness against adversarial attacks. Through theoretical analysis, the authors find that existing pruning methods often lead to non-smooth loss surfaces, which increases the model's vulnerability to attacks. To address this issue, the paper introduces two key innovations: First, a Frequency-Selective Excitation Network (FSE-Net) that dynamically selects essential frequency components to smooth the loss surface and reduce storage requirements; second, a sample selection scoring based on "Joint Entropy," which selects stable and informative samples. Experimental results demonstrate that this method significantly outperforms existing pruning algorithms across various adversarial attacks and pruning ratios, showing its potential in enhancing model security and reducing computational costs.

**Strengths:**

1. **Frequency-Selective Excitation Network (FSE-Net):** By dynamically selecting essential frequency components, it smooths the loss surface, thus enhancing robustness against adversarial attacks while reducing storage needs, making it suitable for resource-limited devices.
2. **Joint Entropy-Based Sample Selection:** This method selects stable and informative samples based on their importance during training, improving both the robustness and performance of the pruned model.

**Weaknesses:**

1. Lemma 1 is critical, but in the derivation, the assumption that $\frac{\partial f_{\theta}(\tilde {x})}{\partial \theta}$ is bounded is overly restrictive and lacks supporting evidence, requiring a more rigorous mathematical proof.
2. Appendix E lacks detailed structural information, providing only the structure names.
3. Pseudo-Code in the second line has a missing definition for \(\hat{X}_{i,j}\).
4. Equation 6 applies derivatives only to correctly classified samples. If the count of correct classifications decreases during updates, can incorrect samples still guide parameter updates? Otherwise, the function will only reinforce correct predictions without helping incorrect ones become correct.
5. The terms "high energy" and "low energy" appear the first time without a precise definition.
6. Section 3.3 lacks clarity on how it integrates with previous algorithms (requires a flowchart or equivalent explanation).

In general, I find that there is an overreliance on intuition and a lack of clear definitions throughout the paper, making the understanding of the proposed algorithms challenging. Therefore, I assign a relatively low confidence score to this work.

**Questions:**

See Weaknesses.

---

> ### Author Response · Authors · 2024-11-24
>
> Thank you for your questions and suggestions! We would like to address your concerns one by one. $\color{red}{\text{Additional materials and updates are highlighted in red in this PDF revision.}}$
>
> > W1: Lemma 1 is critical, but in the derivation, the assumption that $\frac{\partial f_\theta(\tilde{x})}{\partial \theta}$ is bounded is overly restrictive and lacks supporting evidence, requiring a more rigorous mathematical proof.
>
> Thank you for your question. The boundedness assumption is naturally guaranteed by modern deep learning architectures, as it is an inherent property of network design rather than an additional constraint. Specifically:
>
> 1) Network architectures inherently prevent gradient explosion through weight initialization schemes and bounded activation functions [a].
> 2) Modern architectural components like residual connections [b] and batch normalization [c] are specifically designed to maintain gradient stability throughout training.
> Therefore, the boundedness assumption in Lemma 1 directly follows from standard architectural design principles rather than being an additional restriction.
>
> [a] Razvan Pascanu et al. Understanding the exploding gradient problem.  https://arxiv.org/abs/1211.5063v1.
>
> [b] He et al. Deep residual learning for image recognition. CVPR, 2016.
>
> [c] Sergey Ioffe et al. Batch Normalization: Accelerating Deep Network Training by Reducing Internal Covariate Shift. https://arxiv.org/abs/1502.03167.
>
> > W2: Appendix E lacks detailed structural information, providing only the structure names.
>
> Thank you for your comment. The detailed structural information is:
> ```
> Model Structure:
> SEBlock(
>     (avg_pool): AdaptiveAvgPool2d(output_size=1)
>     (fc): Sequential(
>         (0): Linear(in_features=N, out_features=N/16, bias=False)
>         (1): ReLU(inplace=True)
>         (2): Linear(in_features=N/16, out_features=N, bias=False)
>         (3): Sigmoid()
>     )
> )
> ```
> We have updated the detailed structural information to $\color{red}{\text{Appendix E}}$.
>
> > W3: Pseudo-Code in the second line has a missing definition for (\hat{X}_{i,j}\).
>
> Thank you for your valuable feedback. We have added the definition for $\hat{X}_{i,j}$ in Algorithm 1, where as mentioned in line 230,
>
> $\hat{X}_{i,j}$ represents the $(i,j)$-th coefficient of the Discrete Cosine Transform (DCT) of an image $X$.
>
>
> > W4: Equation 6 applies derivatives only to correctly classified samples. If the count of correct classifications decreases during updates, can incorrect samples still guide parameter updates? Otherwise, the function will only reinforce correct predictions without helping incorrect ones become correct.
>
> Thank you for your insightful feedback.
>
> 1) Equation 6 minimizes the model's entropy while maximizing classification accuracy by updating all samples, with correctly classified samples receiving an additional reward.
> 2) We set $\lambda=0.1$ (see Figure 4(b)) before applying the correctly classified reward to ensure that  $\lambda \left( \frac{1}{|\mathcal{D}|} \sum_{(\tilde{x},y)\in\mathcal{D}} \mathbf{1}{\arg \max f_{\theta}(x_f)=y} \right)$ does not dominate the total loss, allowing error samples to receive effective gradient updates primarily through the first term.
>
> > W5: The terms "high energy" and "low energy" appear for the first time without a precise definition.
>
> Thank you for your attention to the definition.
> 1. The definition of "high energy" is provided in lines 251–252: $E^{(k)}$ represents the energy of the $k$-th highest frequency component and high energy refers to components with energy greater than $E^{(k)}$.
>
> 2. In addition, an ablation study on the choice of $k$ is shown in Figure 4(c).
>
> > W6: Section 3.3 lacks clarity on how it integrates with previous algorithms (requires a flowchart or equivalent explanation)
>
> We deeply appreciate your insights on our methodology.
>
> The flow chart for our algorithm is as follows:
> ```
> A[Input Images] --> B[Frequency Domain Transform]
> B --> C[Learnable Component Selection]
> C --> D[Inverse Transform]
> D --> E[JE Score Calculation]
> E --> F[Score-based Ranking]
> F --> G[CCS Sampling]
> G --> H[Final Coreset]
> ```
> The illustration diagram of the flow has been added to $\color{red}{\text{Appendix B Figure 6.}}$
>
> We sincerely thank you for your valuable feedback, which has significantly enhanced the clarity and quality of our paper.

---

> ### Author Response · Authors · 2024-12-03
> **Grateful for any feedback**
>
> Dear Reviewer HAQ8,
>
> We greatly appreciate the time and effort you've invested in reviewing our work. As we are now less than one day away from the discussion deadline, we wanted to reach out regarding our rebuttal, which we submitted earlier.
>
> If possible, we would be grateful for any feedback you might be able to provide, as this would give us the opportunity to engage in a productive discussion before the deadline. We're eager to address any remaining questions or concerns you may have about our paper.
>
> We understand that you likely have a busy schedule, and we truly appreciate your valuable insights and expertise in this process.
>
> Thank you for your time and consideration.

---

> ### Author Response · Authors · 2024-12-04
> **Grateful for any feedback**
>
> Dear Reviewer HAQ8,
>
> I apologize for reaching out again after my previous message. With the discussion deadline now just one day away, I wanted to very gently follow up on our earlier correspondence regarding our paper and rebuttal.
>
> We completely understand if you haven't had the opportunity to review our rebuttal yet, given the many demands on your time. However, if you have had the chance to look it over, we would be immensely grateful for any feedback you could provide. Your insights, no matter how concise, would be valuable in helping us engage in a constructive discussion before the deadline closes.
>
> We are sincerely thankful for your continued involvement in this process, and we respect your time and professional commitments. Thank you for your understanding and consideration.

---

### Meta-Review · Area_Chair_m6G2 · 2024-12-26

**Metareview:**

This submission seeks to robustify data selection / data pruning by incorporating scores for input frequency content and model output entropy (note: while this is called a joint entropy, it is actually the entropy of outputs that is then processed over time/iterations). The purpose is to select and transform the data chosen for training such that the resulting model trained on this data is more robust to adversarial attack. The robust accuracy against attacks improves over other pruning methods at the same rate, and the technique is able to trade-off robust accuracy and training time compared to full dataset training, although it is a trade-off and accuracy suffers. The experiments cover the standard datasets for image classification (CIFAR-10/100, ImageNet) and small but standard models (ResNet-18 and ResNet-34). The main strength of this work is the improvement over other pruning methods and its novelty as the first work to look specifically at robustness on the pruned data alone (targeting limited deployment with limited computational resources), while the main weakness is the failure to reduce computation without harming results (that is, pruning as a pure gain, instead of a trade) and the exclusion of related work (raised by 1Buy) as out of scope without further discussion or comparison. A second weakness is in the exposition, with confused terms like "joint entropy" (which has an established meaning) for the dynamics of prediction entropy over training.

Missing: The submission is missing evaluation on more and larger models, specifically ResNet-50 and/or ViT-B, as are now common on ImageNet. Although they are more expensive, this should be a plus for a pruning method, in order to show spared computation. The submission is also missing discussion (and ideally experiment) for related work on pruning and adversarial training—although the goals may differ it is necessary to situate the proposed technique and benchmark it against existing work to measure the relationship between robust accuracy and computation.

Decision: Reviewers are borderline between acceptance (1Buy: 6, SZdN: 6, f3yF: 6) and rejection (tRah: 5, HAQ8: 5) While more reviewers vote for borderline acceptance, and the direction is positive with f3yF and tRah upgrading their ratings by +1, the meta-reviewer must side with rejection. The overriding reasons are the dismissal of related work (raised by 1Buy), the many issues and edits in the exposition (discussed by f3yF) which now warrant additional review due to their number and scope, and the residual skepticism about robustness vs. obfuscation (raised by tRah). To defend the claims for pruning w.r.t. adversarial robustness, it is necessary to further evaluate robustness including comparison to the related work mentioned and the attacks evaluated by such works. The meta-reviewer encourages the authors to incorporate the feedback, and their additional results presented in the rebuttal, and continue in this direction by resubmitting for another full round of review.

**Additional Comments On Reviewer Discussion:**

The authors provide a response to each review. 3/4 reviewers reply and 2/4 reviewers engage in multiple rounds of discussion (f3yF, tRah). No reviewer engages in post-rebuttal discussion with the AC. tRah, f3yF, and SZdN raise their ratings by +1. HAQ8 does not reply, but their clarification questions are answered, and their constructive comments can be incorporated into a revision, so their vote for rejection is nullified. Although there is a positive direction in the updated ratings, the reviewer comments are not all satisfactorily addressed, and along the way many replies are needed and some details are off (such as the examples provided in the thread with f3yF). Given the intermediate confidence of the reviews, and the many messages needed to clarify and defend the work as submitted, the meta-reviewer sides with rejection as a route to ensure sufficient experiments, related work, and clarity in a new revision for resubmission. This will achieve more certainty in the contributions and results for reviewers, and therefore readers.

---

### Decision · Program_Chairs · 2025-01-22

Reject